# VideoGPA: Distilling Geometry Priors for 3D-Consistent Video Generation

**Hongyang Du** [1 2 *]   **Junjie Ye** [1 *]   **Xiaoyan Cong** [2 *]   **Runhao Li** [1]   **Jingcheng Ni** [2]
**Aman Agarwal** [2]   **Zeqi Zhou** [2]   **Zekun Li** [2]   **Randall Balestriero** [2 †]   **Yue Wang** [1 †]

https://hongyang-du.github.io/VideoGPA-Website

## Abstract

While recent video diffusion models (VDMs) produce visually impressive results, they fundamentally struggle to maintain 3D structural consistency, often resulting in object deformation or spatial drift. We hypothesize that these failures arise because standard denoising objectives lack explicit incentives for geometric coherence. To address this, we introduce **VideoGPA** (**Video G**eometric **P**reference **A**lignment), a data-efficient self-supervised framework that leverages a geometry foundation model to automatically derive dense preference signals that guide VDMs via Direct Preference Optimization (DPO). This approach effectively steers the generative distribution toward inherent 3D consistency without requiring human annotations. VideoGPA significantly enhances temporal stability, geometric plausibility, and motion coherence using minimal preference pairs, consistently outperforming state-of-the-art baselines in extensive experiments.

## 1. Introduction

The rapid evolution of Video Diffusion Models (VDMs) has revolutionized content creation, achieving remarkable generalizability and visual fidelity (Hong et al., 2023; Yang et al., 2025; Kong et al., 2025; Team Wan, 2025; NVIDIA, 2025; Team Seedance, 2025). Beyond artistic generation, the community is actively exploring the potential of VDMs as data engines for downstream tasks such as Embodied AI (Bruce et al., 2024b;a; Yang et al., 2024; Feng et al., 2025; Li et al., 2025a; Ye et al., 2026; Deng et al., 2026; Kim et al., 2026), Novel View Synthesis (NVS) (Liu et al., 2023; Yu

*Equal contribution †Equal advising [1]Physical Superintelligence Lab, University of Southern California, Los Angeles, CA, US [2]Department of Computer Science, Brown University, Providence, RI, US. Correspondence to: Hongyang Du <hongyang_du@brown.edu>, Junjie Ye <yejunjie@usc.edu>.

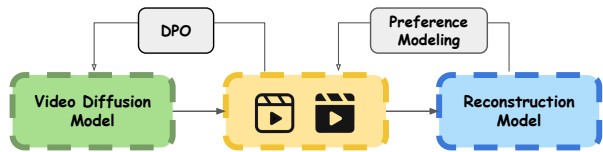

**(a) A Simple Pipeline Improves Video Generation Models**

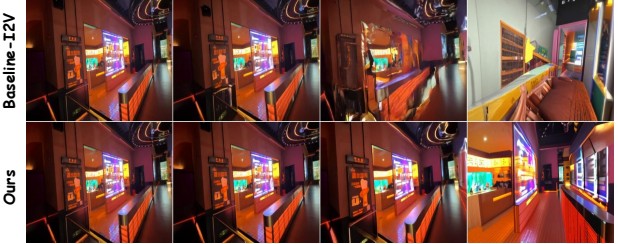

**(b) Image-To-Video Task Example**

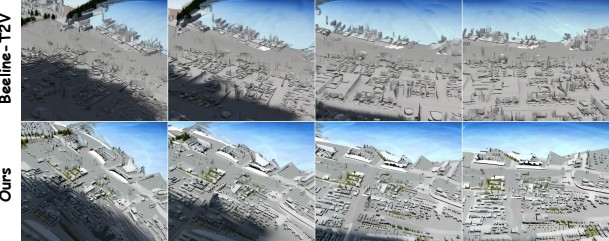

**(c) Text-To-Video Task Example**

*Figure 1.* **Overview of VideoGPA and representative results.** (a) VideoGPA aligns a pretrained video diffusion model through proposed reconstruction-guided preference optimization. (b) Image-to-video examples comparing the base model (Yang et al., 2025) and VideoGPA, showing improved geometric stability under camera motion. (c) Text-to-video examples demonstrating improved structural coherence and reduced geometric artifacts.

et al., 2025; Kwak et al., 2024; Voleti et al., 2024; You et al., 2025), and physics simulation (Sun et al., 2025; Cai et al., 2025; Qin et al., 2025; Liu et al., 2024; Chen et al., 2025). Across these applications, faithful 3D understanding plays a fundamental role. This raises a fundamental question: *Have these models truly learned the 3D laws of the world?* We posit that a 3D consistent VDM, which inherently respects geometric constraints, is critical to model the 3D world.

However, despite being pre-trained on billion-scale datasets, current VDMs still exhibit deficiencies in 3D consistency. As illustrated in Fig. 1, pre-trained VDMs fail to maintain structural consistency and temporal stability. Common

artifacts include object deformation, spatial drifting, and geometry collapse over time.

We attribute this paradox, where models have seen scalable 3D-consistent data but exhibit inconsistent behaviors, to the nature of denoising objective. Standard training incentivizes pixel-level statistical matching but lacks geometric regularization. Consequently, the model learns to hallucinate plausible textures without effectively injecting 3D consistency into latent space.

Recent advances in Geometry Foundation Models (GFMs) demonstrate strong geometric priors through their ability to infer dense 3D structure and camera motion from 2D observations (Wang et al., 2024a; 2025). We leverage these priors by distilling reconstruction-based 3D knowledge into video diffusion models, aligning generation toward physically consistent geometry without retraining from scratch or relying on human annotations.

In this work, we introduce **VideoGPA** (**Video G**eometric **P**reference **A**lignment), a data-efficient, self-supervised framework that equips pre-trained VDMs with 3D consistency. The key to our approach is a novel 3D consistency metric derived from the principle of re-projection consistency. Specifically, given a generated video, we utilize a GFM to render a 3D consistent video as a reference. The discrepancy between the input video and the rendered reference serves as a robust proxy for 3D consistency: if a video is geometrically valid, the GFM-derived 3D structure should accurately reconstruct the original input. By leveraging a geometry foundation model as a reward model backbone, we construct geometric preference pairs, distinguishing between samples with high and low structural integrity. These pairs guide the VDM via Direct Preference Optimization (DPO) (Rafailov et al., 2023), effectively steering the generative distribution toward the 3D-consistent manifold. Remarkably, we show that with only $\sim 2,500$ preference pairs and minimal post-training (LoRA fine-tuning (Hu et al., 2022) on ~1% of model parameters), VideoGPA substantially improves geometric coherence and temporal stability, while preserving the base model's visual quality and motion realism. Across both image-to-video and text-to-video settings, VideoGPA consistently improves over prior art on multiple geometric consistency and perceptual metrics.

## 2. Related Works

### 2.1. Video Generation Models

Recent video generation has achieved high visual fidelity by scaling Diffusion Transformer (DiT) architectures (Yang et al., 2025; Kong et al., 2025; Team Wan, 2025; Team Seedance, 2025; NVIDIA, 2025). Within this paradigm, Team Wan (2025) optimize for high-ratio temporal compression via a 3D causal VAE, while Kong et al. (2025)

introduce a hybrid dual-stream design to refine multimodal fusion. CogVideoX (Yang et al., 2025) serves as a representative of this class, utilizing expert transformer blocks to model spatiotemporal patches. Despite these achievements, these models are fundamentally optimized for pixel-level denoising rather than geometric structural coherence. Consequently, they struggle with 3D geometric consistency, often exhibiting object deformation or spatial drift under global camera maneuvers.

### 2.2. Video Diffusion Alignment

To address these structural bottlenecks, recent research adapts post-training alignment frameworks to refine the output manifold of diffusion models. These approaches generally follow the paradigms of supervised fine-tuning (SFT) and reinforcement learning (RL). SFT-based methods such as Force Prompting (Gillman et al., 2025) improve consistency by training on high-quality curated data, yet they often suffer from limited generalization. Alternatively, RL-based approaches like DDPO (Black et al., 2024), Flow-GRPO (Liu et al., 2025a), and DanceGRPO (Xue et al., 2025) frame denoising as a multistep decision process to optimize for aesthetic or motion rewards. Diffusion-DPO (Wallace et al., 2024) further simplifies this by providing a stable offline objective for preference learning without the complexity of iterative sampling. While Kupyn et al. (2025) explore geometric alignment via epipolar constraints and Liu et al. (2025b) utilize human feedback, we introduce a self-supervised paradigm that leverages feed-forward 3D reconstruction as a dense automated geometric signal.

### 2.3. Geometric Foundation Models

Recent advances in geometric foundation models offer potential for recovering dense geometric structure from sparse views without iterative optimization. This paradigm, established by DUSt3R (Wang et al., 2024a), utilizes transformer architectures to regress pointmaps directly from pixels, while subsequent models like MASt3R (Leroy et al., 2024) further refine local matching. More recently, Wang et al. (2025) and Wang et al. (2026) scale this approach to multi-view sequences by simultaneously predicting globally consistent camera poses and pointmaps. This capability provides a stable, dense, and differentiable 3D reference, bridging the gap between 2D appearance and 3D space. Such models offer huge potential as automated geometric supervisors for generative tasks. By distilling these structural priors into the diffusion process, we explore how reconstruction can help rectify the motion manifold of a generator to ensure geometric plausibility while maintaining the simplicity of the standard generative pipeline.

# 3. Methodology

VideoGPA introduces a review-and-correct framework that aligns video diffusion models with 3D geometric laws. As shown in Fig. 2, the process begins by using a geometry foundation model to extract the 3D structure and camera motion from generated videos. We then calculate a 3D consistency score by measuring the error in reconstructing the original frames from this 3D structure. Finally, with preference pairs constructed using these scores, we apply DPO to teach the model to favor geometrically consistent outputs. This lightweight approach allows the model to maintain rigid structures with minimal additional training.

## 3.1. Preliminaries

**Direct Preference Optimization** The DPO (Rafailov et al., 2023) framework is derived from the Bradley-Terry preference model. For a policy $\pi$, the probability that a preferred completion $x^w$ is chosen over a dispreferred completion $x^l$ given a context $c$ is:

$$P(x^w \succ x^l | c) = \sigma \left( r(x^w, c) - r(x^l, c) \right), \qquad (1)$$

where $r(x, c)$ is the reward function and $\sigma$ is the sigmoid function. DPO reparameterizes the reward using the log-likelihood ratio between policy $\pi_\theta$ and reference policy $\pi_{\text{ref}}$:

$$r(x, c) = \beta \log \frac{\pi_\theta(x|c)}{\pi_{\text{ref}}(x|c)} + \beta \log Z(c). \qquad (2)$$

Substituting this into the Bradley-Terry model yields the DPO objective:

$$\mathcal{L}_{\text{DPO}} = -\mathbb{E}_{(c,x^w,x^l)} \Big[ \log \sigma \Big( \beta \log \frac{\pi_\theta(x^w|c)}{\pi_{\text{ref}}(x^w|c)} \\ - \beta \log \frac{\pi_\theta(x^l|c)}{\pi_{\text{ref}}(x^l|c)} \Big) \Big]. \qquad (3)$$

**From Policy to Diffusion Log-likelihood** In diffusion models, the log-probability $\log p(x)$ is typically approximated via the Evidence Lower Bound (ELBO). For a denoising model $\epsilon_\theta$, the log-probability ratio can be expressed as the difference in the score-matching loss (Wallace et al., 2024). Specifically, for a given timestep $t$:

$$\log \frac{\pi_\theta(x|c)}{\pi_{\text{ref}}(x|c)} \propto -\mathbb{E}_{t,\epsilon} \Big[ \|\epsilon - \epsilon_\theta(x_t, t, c)\|^2 \\ - \|\epsilon - \epsilon_{\text{ref}}(x_t, t, c)\|^2 \Big], \qquad (4)$$

where $x_t = \sqrt{\bar{\alpha}_t} x_0 + \sqrt{1 - \bar{\alpha}_t} \epsilon$ represents the noisy latent at timestep $t$.

## 3.2. DPO for $v$-Prediction Video Diffusion

Recent advancements in video generation have been significantly driven by diffusion transformers (DiTs) employing

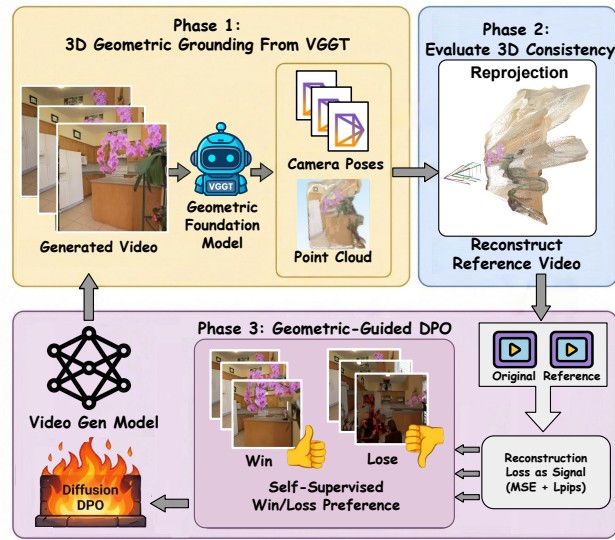

*Figure 2.* **Pipeline of VideoGPA.** A geometric foundation model probes generated videos to assess scene-level 3D consistency, which is used to form self-supervised preference pairs for post-training alignment via DPO.

the $v$-prediction parameterization (Salimans & Ho, 2022; Ho et al., 2022; Lipman et al., 2023). Trained on large-scale datasets, these models demonstrate remarkable potential in synthesizing high-fidelity dynamic content with stable training convergence. Building upon this foundation, we adapt the Diffusion-DPO (Wallace et al., 2024) framework to this parameterization to explicitly steer the model toward geometrically consistent manifolds.

For a general diffusion model trained with $v$-prediction, the target velocity $v_t$ at timestep $t$ is formally defined as:

$$v_t \equiv \dot{x}_t = \alpha_t \epsilon - \sigma_t x_0, \qquad (5)$$

where $\alpha_t$ and $\sigma_t$ are the noise schedule coefficients. The network $v_\theta$ is trained to minimize the mean squared error (MSE) relative to the velocity target $v_t$.

By substituting the velocity error into the DPO framework, we define the energy term $\mathcal{E}$ for a sample $x$ as:

$$\mathcal{E}(\theta, x, t) = \|v_t - v_\theta(x_t, t, c)\|^2. \qquad (6)$$

The log-probability ratio in the velocity space thus becomes:

$$\log \frac{\pi_\theta(x|c)}{\pi_{\text{ref}}(x|c)} \propto \mathbb{E}_{t,\epsilon} \left[ \mathcal{E}(\text{ref}, x, t) - \mathcal{E}(\theta, x, t) \right] \qquad (7)$$

The final DPO loss for our $v$-prediction video model is formulated as the negative log-likelihood of the reward margin between the winning and losing samples:

$$\mathcal{L}_{\text{DPO}} = -\mathbb{E} \Big[ \log \sigma \Big( \beta \Big( [\mathcal{E}(\text{ref}, x^w, t) - \mathcal{E}(\theta, x^w, t)] \\ - [\mathcal{E}(\text{ref}, x^l, t) - \mathcal{E}(\theta, x^l, t)] \Big) \Big) \Big]. \qquad (8)$$

During training, we sample shared noise $\epsilon$ and timestep $t$ for each preference pair $(x^w, x^l)$ to ensure a consistent optimization baseline. The model $v_\theta$ (parameterized via LoRA (Hu et al., 2022)) is updated to minimize the velocity prediction error for $x^w$ relative to $x^l$, effectively steering the latent video manifold toward higher geometric consistency.

### 3.3. Preference Modeling

DPO requires preference pairs to guide post-training alignment. In our setting, the objective favors video samples that exhibit stronger 3D geometric consistency. To this end, we derive a self-supervised geometric preference signal by analyzing generated videos with a feed-forward geometric foundation model (Wang et al., 2025). The resulting 3D consistency score enables automatic construction of preference pairs without human annotations or explicit structural priors. We next describe how this score is computed from reconstructed camera motion and scene geometry.

For each generated video, we uniformly sample $T$ frames to obtain an image sequence $\mathcal{I} = \{I_t\}_{t=1}^T$. Given this sequence, the geometric model $\Phi$ predicts a depth map $D_t$ and camera pose $(R_t, t_t)$ for each frame $I_t$,

$$(D_t, R_t, t_t) = \Phi_\theta(I_t), \qquad R_t \in SO(3),\ t_t \in \mathbb{R}^3, \quad (9)$$

along with camera intrinsics $K \in \mathbb{R}^{3 \times 3}$. Using the camera-to-world transform $E_t = [R_t | t_t] \in SE(3)$, each pixel $\tilde{\mathbf{u}} = [u, v, 1]^\top$ with depth $D_t(u, v) > 0$ can be projected to the world coordinate frame with:

$$\begin{aligned} \mathbf{x}_t^{\mathrm{cam}}(u, v) &= D_t(u, v)\, K^{-1} \tilde{\mathbf{u}}, \\ \mathbf{X}_t(u, v) &= R_t\, \mathbf{x}_t^{\mathrm{cam}}(u, v) + t_t, \end{aligned} \quad (10)$$

formulating a colored point cloud $\mathcal{P} = \{(\mathbf{X}_i, \mathbf{c}_i)\}_{i=1}^N$, where $\mathbf{c}_i$ is RGB color. By default, we set $T = 10$.

**3D Consistency Score**   We measure 3D geometric consistency by how well the recovered structure explains the original frames under reprojection. Geometrically coherent videos admit a consistent 3D explanation across viewpoints, while inconsistencies lead to elevated reprojection error.

Formally, each 3D point $\mathbf{X} \in \mathcal{P}$ is reprojected into frame $k$ using inverse camera pose $E_{t,\mathrm{w2c}} = [R_t^\top | -R_t^\top t_t]$, yielding

$$\mathbf{x}_t^{\mathrm{cam}} = R_t^\top (\mathbf{X} - t_t), \quad (u_t, v_t) = \pi(K\, \mathbf{x}_t^{\mathrm{cam}}), \quad (11)$$

where $\pi$ denotes perspective division. Using vectorized rendering with a painter's algorithm, we obtain a reprojected image $\hat{I}_t$ for each frame.

We quantify geometric consistency between reprojected image $\{\hat{I}_t\}_{t=1}^T$ and original frame $\{I_t\}_{t=1}^T$ using a standard reconstruction loss (Yao et al., 2018):

$$E_{\mathrm{Recon}} = \frac{1}{T} \sum_{t=1}^T \Big( \mathrm{MSE}(\hat{I}_t, I_t) + \lambda \mathrm{LPIPS}(\hat{I}_t, I_t) \Big). \quad (12)$$

Lower reconstruction error indicates stronger cross-view geometric consistency, while higher error reflects violations of 3D coherence. We use this reprojection-based error as a dense, self-supervised signal to construct preference pairs.

### 3.4. Post-Training Alignment

With the proposed preference modeling strategy, we curate a specialized training dataset for post-training alignment by exposing geometric differences among candidate generations. Specifically, for each conditioning input, we sample multiple videos from a pretrained video diffusion model using different random seeds. These samples share the same semantic content but may differ in geometric consistency, allowing us to isolate geometry as the primary factor for preference learning and construct preference pairs based on the 3D consistency score (Sec. 3.3). We consider both image-to-video and text-to-video settings to ensure the learned alignment generalizes across conditioning modalities.

**Image-to-Video (I2V)**   For I2V data, we use the initial frames from a subset of DL3DV-10K (Ling et al., 2024) as visual prompts. To encourage the model to generate samples with diverse camera trajectories where geometric inconsistencies are more likely to emerge, we design structured motion prompts composed of 2–3 randomly sampled camera motion primitives from a predefined vocabulary (*e.g.*, *pull back away from the scene*, *roll gently to one side*, *orbit around the scene*). The full list of motion primitives is provided in Appendix A. This scripted prompting strategy systematically varies camera motion while keeping scene content fixed, making geometric consistency the dominant distinguishing factor among candidate samples and reducing confounding effects from semantic variation.

**Text-to-Video (T2V)**   For T2V data, we use video captions generated by CogVLM2-Video (Hong et al., 2024) as textual prompts. Compared to the I2V setting, this setup naturally introduces higher semantic diversity and more open-ended scene dynamics, enabling us to evaluate whether the proposed alignment strategy remains effective under unconstrained language inputs and complex scene compositions.

For each prompt, we rank candidate samples according to their 3D consistency scores and form preference pairs by selecting samples with a sufficient margin in geometric consistency. To ensure a stable and informative training signal, we prune samples that are static, exhibit poor overall visual quality, or show negligible score differences, following the filtering strategy detailed in Appendix A.

*Table 1.* **Quantitative evaluation** on image-to-video (I2V) and text-to-video (T2V) generation. We report 3D reconstruction error, 3D geometric consistency metrics, and human-aligned VideoReward scores. Overall, VideoGPA consistently achieves the strongest geometric consistency while maintaining or improving perceptual quality across various base models.

| Method | 3D Reconstruction Error | | | 3D Consistency | | | VideoReward (Win Rate %) | | | |
|---|---|---|---|---|---|---|---|---|---|---|
| | PSNR ↑ | SSIM ↑ | LPIPS ↓ | MVCS ↑ | 3DCS ↓ | Epipolar ↓ | VQ | MQ | TA | OVL |
| *Image-to-Video (I2V) Base Model: CogVideoX-I2V-5B* | | | | | | | | | | |
| Baseline-I2V | **22.85** | **0.786** | 0.476 | 0.945 | 0.485 | 0.585 | - | - | - | - |
| SFT | 21.58 | 0.749 | 0.513 | 0.947 | 0.524 | 0.640 | 44.67 | 33.00 | 52.67 | 35.00 |
| Epipolar-DPO | 21.38 | 0.773 | 0.475 | 0.944 | 0.487 | 0.545 | 67.33 | 51.33 | 56.67 | 66.00 |
| **VideoGPA (Ours)** | 21.24 | 0.779 | **0.473** | **0.950** | **0.483** | **0.539** | **74.00** | **56.00** | **57.67** | **76.00** |
| *Text-to-Video (T2V) Base Model: CogVideoX-5B* | | | | | | | | | | |
| Baseline-T2V | 21.47 | 0.784 | 0.435 | 0.944 | 0.445 | 0.584 | - | - | - | - |
| SFT | 19.99 | 0.721 | 0.496 | 0.937 | 0.510 | 0.719 | 14.67 | 23.67 | 39.33 | 15.33 |
| Epipolar-DPO | **21.58** | 0.791 | 0.434 | **0.953** | 0.443 | 0.579 | 45.00 | 53.67 | 49.00 | 48.67 |
| **VideoGPA (Ours)** | 21.24 | **0.803** | **0.411** | 0.953 | **0.422** | **0.548** | **62.67** | **67.00** | 42.67 | **60.33** |
| *Text-to-Video (T2V) Base Model: CogVideoX1.5-5B* | | | | | | | | | | |
| Baseline-T2V15 | 16.79 | 0.538 | 0.522 | 0.980 | **0.548** | 0.685 | - | - | - | - |
| GeoVideo | 15.20 | 0.458 | 0.667 | 0.819 | 0.703 | 0.875 | 17.36 | 44.44 | 30.56 | 18.06 |
| **VideoGPA (Ours)** | 14.88 | 0.503 | **0.520** | **0.982** | 0.556 | **0.567** | **60.42** | **54.17** | **52.08** | **57.64** |

*Note: All reprojection based metrics are calculated using the Depth Anything V3 backbone to prevent circular evaluation.*

During evaluation, we adopt natural, descriptive narrations generated by CogVLM2-Video (Hong et al., 2024) as prompts for both I2V and T2V tasks. Although this differs from the structured motion prompts used during I2V training, we observe no evidence of overfitting to the scripted prompt format. Instead, models trained with our alignment strategy retain their ability to handle detailed natural language descriptions while exhibiting substantially improved 3D geometric consistency.

## 4. Experiments

### 4.1. Experimental Setup

#### 4.1.1. TRAINING DETAILS

We evaluate our method using CogVideoX (Yang et al., 2025), a representative large-scale video diffusion model, in both I2V and T2V settings. We fine-tune CogVideoX 5B models using LoRA, employing a rank $r = 64$ and a scaling factor $\alpha = 128$ within the PEFT framework (approximately 1% of model parameters) (Hu et al., 2022; Xu et al., 2026). Training is performed on 8×A100 GPUs for 10,000 steps using AdamW optimizer, with a peak learning rate of $5 \times 10^{-6}$, a cosine decay schedule, 500 warm-up steps, and batch size 16. Unless otherwise specified, subsets $8K$, $9K$, $10K$, and $11K$ from DL3DV-10K (Ling et al., 2024) are used for training, and subset $1K$ is used for evaluation.

#### 4.1.2. BASELINES

We compare our proposed VideoGPA against several representative baselines that reflect different strategies for improving geometric consistency in video generation:

*Base model* uses the pretrained video diffusion model (Yang et al., 2025) without post-training alignment as a reference for evaluating the effect of alignment.

*Supervised Fine-Tuning (SFT)* fine-tunes the base video diffusion model on curated video-text pairs. This baseline represents data-driven improvement without explicit geometric preference modeling.

*Epipolar-DPO* (Kupyn et al., 2025) applies DPO using epipolar errors as a preference signal. This method leverages sparse pair-wise geometric constraints to rank samples.

*GeoVideo* (Bai et al., 2025) incorporates explicit geometric consistency losses during supervised fine-tuning to encourage stable scene structure in large camera motion.

For fair comparison, Epipolar-DPO and SFT are trained using the same LoRA configuration and optimization schedule as our method. Detailed GPU usage is in Appendix B.

#### 4.1.3. EVALUATION PROTOCOL

We evaluate all methods along three complementary dimensions to capture geometric fidelity and perceptual quality:

**3D Reconstruction Error** We assess the pixel-level fidelity of the video-to-3D reconstruction process by reprojecting reconstructed geometry back to the image plane. Quantitative metrics include PSNR for intensity differences, SSIM for structural similarity, and LPIPS for perceptual similarity between reprojected frames and the original video.

**3D Consistency** We measure the geometric plausibility and cross-view consistency of generated videos using multi-

ple geometry-based metrics. Multi-View Consistency Score (MVCS) (Bai et al., 2025) evaluates cross-frame projection accuracy, our 3D Consistency Score (3DCS) measures global scene integration as defined in Eq. 12, and the Epipolar Sampson Error (Kupyn et al., 2025) quantifies point-to-epipolar-line distances, reflecting camera-to-scene geometric precision.

**Human-Aligned Video Quality** To evaluate alignment with human preferences, we use VideoReward (Liu et al., 2025b) to compute win rates against baselines. This includes Visual Quality (VQ), Motion Quality (MQ), Text Alignment (TA), and Overall (OVL) scores. Please note that static videos are filtered out during both training and evaluation to ensure reported metrics accurately reflect performance on the motion manifold rather than stationary content.

### 4.2. Quantitative Evaluation

**Image-to-Video (I2V)** We first evaluate I2V generation, where maintaining geometric consistency relative to the input frame is particularly challenging. As shown in Table 1, VideoGPA consistently improves both geometric fidelity and perceptual quality over all baselines. For the **CogVideoX-I2V-5B** baseline, VideoGPA achieves the best performance across all 3D consistency metrics, improving MVCS from $0.945$ to $0.950$, reducing 3DCS from $0.485$ to $0.483$, and decreasing Epipolar error from $0.585$ to $0.539$. These geometric gains translate to significantly stronger human-aligned performance, reaching an overall VideoReward (OVL) win rate of 76.0%, substantially outperforming Epipolar-DPO (66.0%) and SFT (35.0%).

**Text-to-Video (T2V)** Without a reference image, T2V introduces greater semantic diversity. As shown in Table 1 (middle), VideoGPA again delivers consistent improvements over the **CogVideoX-5B** base model and competing baselines. In terms of 3D reconstruction error, VideoGPA achieves the best SSIM (0.803) and LPIPS (0.411). It also outperforms Epipolar-DPO and SFT across all 3D consistency metrics, achieving the highest MVCS (0.953) and the lowest 3DCS (0.422) and epipolar error (0.548). Importantly, these geometric gains do not come at the cost of perceptual quality. VideoGPA substantially improves human-aligned metrics, achieving an overall VideoReward win rate of 60.33%, compared to 48.67% for Epipolar-DPO and 15.33% for SFT. These results indicate that VideoGPA generalizes beyond image-conditioned generation and remains effective under unconstrained text prompts.

We further compare VideoGPA with GeoVideo (Bai et al., 2025) in the T2V setting. Since GeoVideo is based on **CogVideoX1.5-5B**, we post-train the same base model with VideoGPA. Notably, VideoGPA is trained for only 1,500 steps, in contrast to GeoVideo, which is trained on ∼10,000

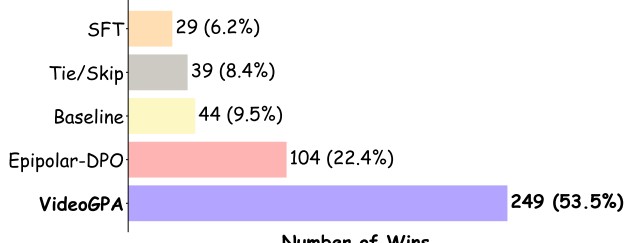

*Figure 3.* **Human preference study** on I2V generation. VideoGPA is most frequently preferred, indicating improved perceptual quality and 3D consistency.

DL3DV-10K videos with depth supervision. As shown in Table 1, GeoVideo improves reconstruction accuracy for large camera motions, but exhibits degraded perceptual quality, reflected by low VideoReward scores (OVL: 18.06%). In contrast, with this lightweight post-training, VideoGPA already maintains the base model's generative quality while achieving stronger overall geometric consistency (Epipolar: 0.567 v.s. 0.875; MVCS: 0.982 v.s. 0.819) and substantially higher human-aligned performance (OVL: 57.64%). These results suggest that geometry-guided preference alignment provides a more balanced improvement than explicit geometric supervision.

**Additional Quantitative Evaluation** We further verify that VideoGPA does not sacrifice scene complexity or motion dynamics, as detailed in Appendix D. Additional out-of-distribution (OOD) evaluation results on the WebVid (Bain et al., 2021) and Panda-70M (Chen et al., 2024) datasets are presented in Appendix F. We further evaluate Wan2.2 (Team Wan, 2025) to examine VideoGPA generalizability, reported in Appendix G. Finally, comprehensive visual comparisons and qualitative examples are provided in Appendix I.

### 4.3. Human Preference Study

While the quantitative results demonstrate consistent improvements across metrics, we further evaluate whether these gains are perceptible to humans through a blind user preference study. The study is performed in the I2V setting. A total of 25 participants are each assigned 20 randomly sampled video groups, where each group contains four videos generated by different methods using the same prompt and seed. The order of videos within each group is randomized. Participants are asked to select the best video based on overall visual quality and consistency, with the option to skip when samples are indistinguishable. As shown in Fig. 3, VideoGPA achieves the highest preference rate by a large margin, accounting for 53.5% of total wins. In contrast, the second-best method Epipolar-DPO receives 22.4% of the votes. These results indicate that the geometric improvements introduced by VideoGPA are consistently perceptible to humans and translate into higher preference.

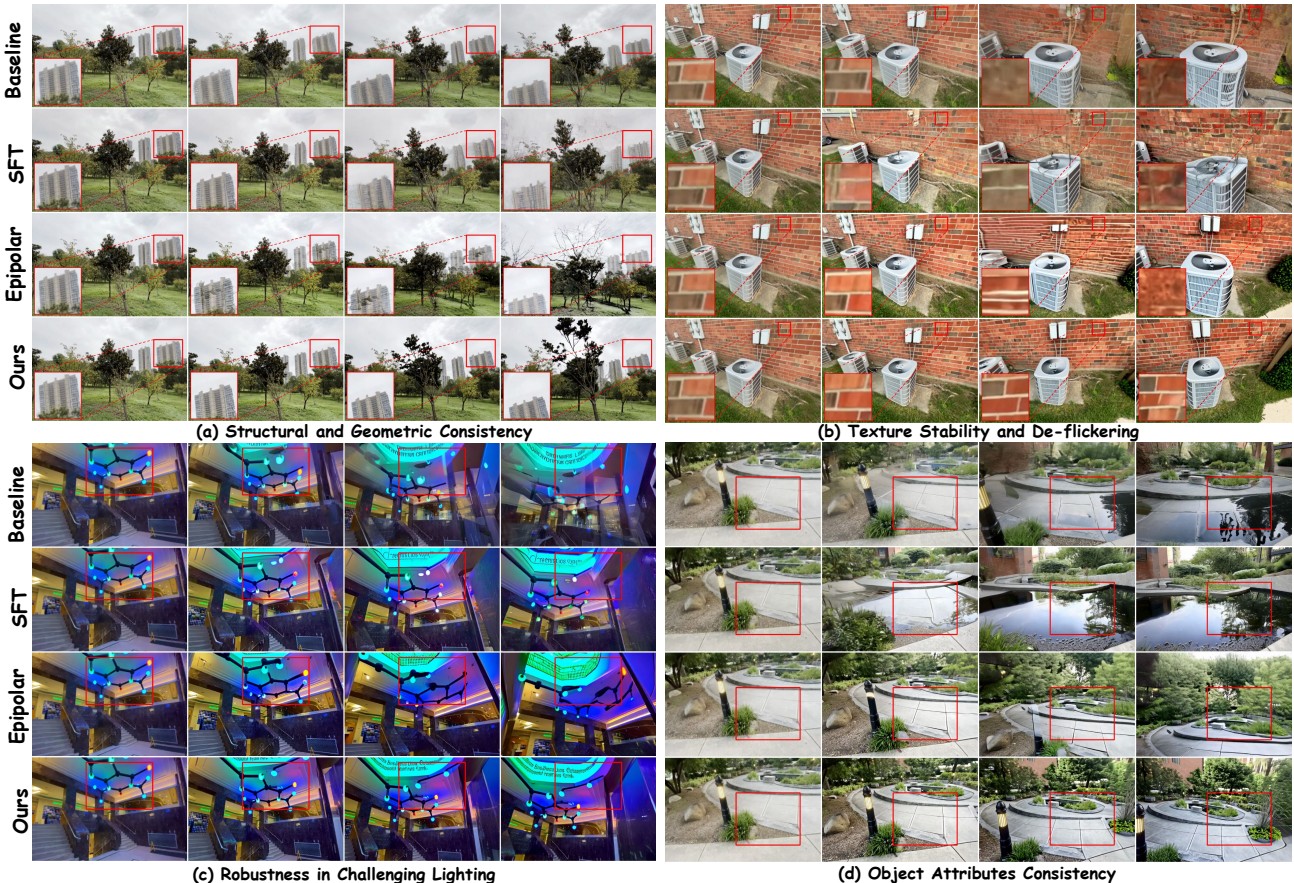

*Figure 4.* **Qualitative comparison** on I2V generation. We compare VideoGPA with the base model, SFT, and Epipolar-DPO. Highlighted regions illustrate improvements in (a) structural and geometric consistency, (b) texture stability and de-flickering, (c) robustness under challenging lighting, and (d) object attribute consistency.

## 4.4. Qualitative Analysis

We observe consistent qualitative improvements across several key aspects: i) *Structural and geometric consistency* is substantially improved, with VideoGPA suppressing object splitting and preserving rigid-body integrity under camera motion (Fig. 4a). ii) *Texture stability* is enhanced, significantly reducing flickering in high-frequency regions such as building facades and fan grilles (Fig. 4b). iii) *Robustness under challenging lighting* is improved, preventing scene degradation in low-light conditions and stabilizing specular reflections on reflective surfaces (Fig. 4c). iv) *Object attribute consistency* is better maintained, with the model preserving semantic identity across frames, including color constancy and material appearance (Fig. 4d). v) Although our approach does not explicitly optimize motion dynamics, it avoids degradation and often improves *dynamic motion coherence*, maintaining object integrity in scenes involving large camera motion or dynamic elements (Appendix H), which we further discuss in Sec. 5.2. A comprehensive set of additional visualizations and frame-by-frame comparisons is provided in Appendix I.

## 5. Discussion

### 5.1. Scene-Level Geometry *v.s.* Local Constraints

In this subsection, we analyze why scene-level geometric preference modeling, as used in VideoGPA, provides a more reliable alignment signal than local, pairwise geometric constraints. Epipolar-DPO (Kupyn et al., 2025) relies on frame-level epipolar relations, which are effective for minor geometric corrections but fragile under severe generative artifacts. In practice, local metrics can yield false positives, since degenerate outputs such as texture collapse or frozen regions may still satisfy sparse epipolar constraints or exhibit high local similarity, resulting in weak or misleading preference signals during alignment.

In contrast, VideoGPA evaluates geometric consistency at the scene-level by enforcing a global reprojection constraint. All frames must jointly admit a single, coherent 3D explanation. Otherwise, reconstruction error increases sharply. This global requirement prevents spatial drift from accumulating over time and provides a dense, unambiguous signal that penalizes collapsed or physically implausible gener-

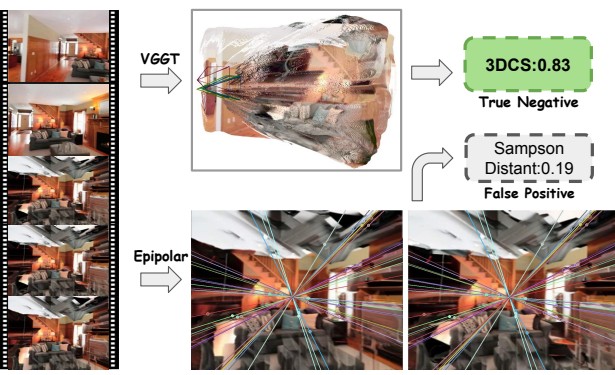

*Figure 5.* **Scene-level *v.s.* local geometry.** Comparison between local geometric metric and scene-level metric on a corrupted video. Local, pairwise constraints yield a false positive, while the scene-level metric correctly identifies geometric inconsistency.

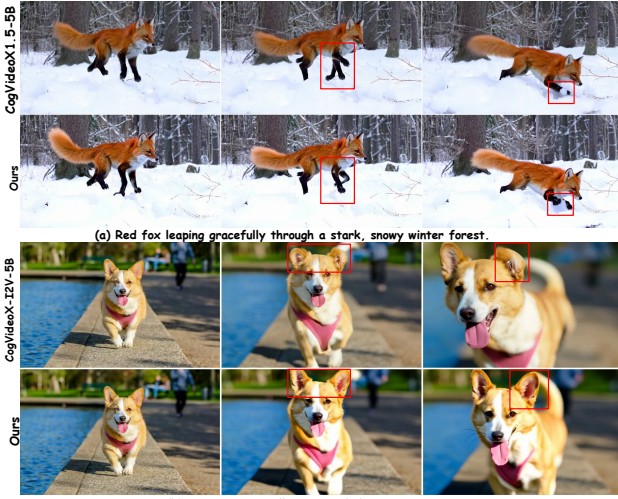

*Figure 6.* **Improved object motion coherence** in both T2V (top) and I2V (bottom) generation. VideoGPA better preserves object structure and motion continuity across frames.

ations. Consequently, preference optimization guided by scene-level geometry avoids rewarding locally consistent but globally invalid samples, leading to more stable alignment. Fig. 5 illustrates representative cases where epipolar-based metrics fail to penalize corrupted sequences that are correctly rejected by the proposed scene-level metric.

### 5.2. Geometry as a Regularizer for Motion Generation

Although VideoGPA explicitly targets geometric consistency in predominantly static scenes, we observe consistent improvements in dynamic motion coherence as demonstrated in Fig. 6 and reflected by higher Motion Quality (MQ) win rates in Table 1. We interpret this behavior through the lens of the *video motion manifold*. Video diffusion models strive to approximate the high-dimensional manifold of natural video distributions (Ho et al., 2020; Song et al., 2021). Geometric inconsistencies (e.g., warping backgrounds or inconsistent perspectives) represent a divergence from the physically plausible subspace of this manifold (Karras et al., 2022; Blattmann et al., 2023).

By enforcing strict geometric constraints, VideoGPA effectively acts as a *geometric regularizer*, projecting the generative process back onto a manifold subspace where 3D consistency holds. This geometric stability serves as a foundational *anchor* for dynamic generation. When the consistent background and camera trajectory adhere to projective geometry, the model can more effectively disentangle camera movement from object motion (Wang et al., 2024b; Guo et al., 2024). Consequently, the model's capacity is freed from hallucinating spatial corrections, allowing its inherent motion priors to focus on generating coherent and realistic object dynamics. In essence, by fixing the geometry of the *stage*, we enable the model to generate more coherent performances for the *actors*. Futher discussion in Appendix H.

## 6. Conclusion

We address geometric inconsistency in video diffusion models through post-training alignment guided by a scene-level 3D consistency score derived from geometry foundation models. By integrating this score into a preference optimization framework, we demonstrate that pretrained video diffusion models can be effectively aligned to produce videos with substantially improved 3D coherence and temporal stability, without introducing explicit structural priors or degrading general-purpose generation quality. Our results suggest that geometric failures in current video generators are largely attributable to objective misalignment rather than architectural limitations, and can be mitigated through lightweight post-training. A remaining limitation is the scalability of geometric reconstruction, whose runtime and memory costs grow with video length. We expect advances in lightweight geometric foundation models to address this.

## Acknowledgement

The USC Physical Superintelligence Lab acknowledges generous support from Toyota Research Institute, Dolby, Google DeepMind, Capital One, Nvidia, and Qualcomm. Junjie Ye is supported by a fellowship from Capital One. Yue Wang is also supported by a Powell Research Award.

## Impact Statement

This paper presents work whose goal is to advance the field of machine learning. There are many potential societal consequences of our work, none of which we feel must be specifically highlighted here.

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

# A. Additional Details on Preference Data Construction

This appendix provides additional implementation details for the data curation pipeline used in post-training alignment. The core strategy is shared across both image-to-video (I2V) and text-to-video (T2V) settings, with additional design choices introduced for I2V to elicit diverse camera motion.

For both I2V and T2V alignment, we generate candidate videos by sampling from pretrained video diffusion models using multiple random seeds per conditioning input. Specifically, we source approximately 3,000 conditioning inputs from subsets $8k$, $9k$, $10k$, and $11k$ of the DL3DV-10K (Ling et al., 2024) dataset. For each input, we generate three video samples with 3 different random seeds, resulting in roughly 9,000 candidate videos in total. Detailed statistics for each model variant and training configuration are reported in Table 3.

Candidate samples are ranked using the 3D consistency score described in Sec. 3.3, and preference pairs $(x_w, x_l)$ are constructed by selecting the best and worst samples within each group. To ensure the quality and discriminative power of the preference signal, we apply the same multi-stage filtering procedure for both I2V and T2V data.

- **Motion Salience Filtering**: To avoid trivial static generations, we first quantify camera motion magnitude from reconstructed poses. For consecutive camera poses $(R_i, t_i)$, we compute translation $\Delta t_i = \|t_{i+1} - t_i\|_2$ and rotation $\Delta \theta_i = \arccos\left(\frac{\text{Tr}(R_{i+1}R_i^\top)-1}{2}\right)$. Translation is normalized by its mean step $s_{\text{trans}} = \frac{1}{T-1}\sum_i \Delta t_i$, while rotation remains unscaled:

$$\bar{t} = \frac{1}{T-1}\sum_i \frac{\Delta t_i}{s_{\text{trans}}}, \qquad \bar{r} = \frac{1}{T-1}\sum_i \Delta \theta_i. \tag{13}$$

  The motion score $\alpha$ is defined as:

$$\alpha = \bar{t} + \lambda \cdot \bar{r} + \epsilon, \tag{14}$$

  where $\lambda = 0.1$. Samples with $\alpha < 0.001$ are discarded.

- **Geometric Margin Selection**: Within each generation group, we retain only preference pairs whose 3D consistency scores differ by more than 0.05, ensuring that the preference signal reflects a meaningful geometric distinction.

- **Difficulty Pruning**: Finally, we remove pairs in which the preferred sample still exhibits poor global geometry, defined as a consistency score greater than 0.8. This prevents the model from learning from low-quality geometric references.

After filtering, we obtain a compact set of preference pairs for each model variant, with statistics reported in Table 3. For fair comparison, we adopt the same preference data construction pipeline for Epipolar-DPO (Kupyn et al., 2025), including identical candidate generation and filtering steps. The two methods differ in the geometric signal used to evaluate candidate consistency, with Epipolar-DPO relying on local, pairwise epipolar constraints rather than the proposed scene-level 3D consistency formulation.

**Scripted Prompting for I2V.** In the I2V setting, we additionally introduce scripted camera-motion prompts to elicit diverse camera trajectories while keeping scene content fixed. Each prompt consists of a static-scene constraint followed by a multi-stage camera motion description composed of $N \in \{2, 3\}$ motion primitives. Motion primitives are drawn from three categories, translations, rotations, and complex paths, as listed in Table 2, and concatenated using natural temporal connectors such as "then" and "followed by". An example of a generated prompt is: *"slide sideways across the room, then pan toward the main subject, followed by a gentle roll to one side."* The scripted camera-motion data construction procedure used to generate preference pairs is summarized in Algorithm 1.

To prevent the model from introducing unintended dynamics, we prepend a fixed static-scene constraint to every prompt:

> *"A realistic continuation of the reference scene. Everything must remain completely static: no moving people, no shifting objects, and no dynamic elements. Only the camera is allowed to move."*

This constraint ensures that geometric consistency is the primary factor distinguishing candidate samples during alignment.

Please note that although this scripted setup explicitly restricts training samples to static scenes, we observe that the resulting alignment improves 3D consistency even in dynamic scenes generated by the model. As discussed in Sec. 5.2, this behavior suggests that enforcing scene-level geometric consistency provides a stabilizing inductive bias that transfers beyond the static training setting.

*Table 2.* The predefined motion primitives used for synthetic prompt generation. These actions are combined using temporal connectors to form diverse camera trajectories.

| Translations | Rotations | Complex Paths |
|---|---|---|
| push forward into the scene | pan across the room | orbit around the scene |
| pull back away from the scene | pan toward the main subject | arc around the center of the room |
| slide sideways across the room | scan across the shelves | circle around the main object |
| move laterally along the furniture line | tilt upward toward the ceiling | swing around the room |
| drift across the space | tilt downward toward the floor | pivot around the viewpoint |
| glide toward the room center | roll gently to one side | |
| shift through the foreground | look around the environment | |
| move diagonally through the space | | |

---

**Algorithm 1** Scripted Camera-Motion Preference Data Construction for I2V

---

**Require:** Set of first frames $\mathcal{I}$, motion primitive sets $\mathcal{M}_{T,R,C}$
**Ensure:** Metadata repository $\mathcal{D}$
1: **for** each image $I_i \in \mathcal{I}$ **do**
2:      $n \leftarrow \mathrm{random}(\{2,3\})$
3:      segments $\leftarrow$ sample $n$ pieces from $\{\mathcal{M}_T \cup \mathcal{M}_R \cup \mathcal{M}_C\}$
4:      $\mathcal{A}_{motion} \leftarrow \mathrm{JoinSegments}(\text{segments, "then", "followed by")}$
5:      $P_{text} \leftarrow \mathrm{StaticPrefix} + \text{"Camera motion: "} + \mathcal{A}_{motion}$
6:      {Prune pairs with static motion, low-quality winners, or negligible gaps.}
7:      $\mathcal{D}[ID_i] \leftarrow \{I_i, \mathcal{A}_{motion}, P_{text}\}$
8: **end for**
9: **return** $\mathcal{D}$

---

## B. Training Details and GPU Usage

All experiments follow a unified optimization protocol to ensure fair comparison across methods. The training is conducted on $8\times$ NVIDIA A100 GPUs using the AdamW optimizer with a peak learning rate of $5 \times 10^{-6}$. We employed a cosine decay schedule with 500 warm-up steps and a global batch size of 16. All post-training alignment uses LoRA (Hu et al., 2022) with rank $r = 64$ and $\alpha = 128$, affecting approximately 1% of the total model parameters.

**Data Statistics**    For preference-based alignment, we generate multiple candidate videos per conditioning input and curate preference pairs using the filtering strategy described in Sec. 3.4 and Appendix A. After filtering for motion salience and geometric margin, each model variant is trained on a compact set of $\sim 2500$ preference pairs. The same data construction pipeline is applied to VideoGPA and Epipolar-DPO to ensure fair comparison.

For the SFT baseline, we fine-tune on video-caption pairs derived from DL3DV-10K using captions generated by CogVLM2-Video, resulting in 20,356 training videos. This setup mirrors standard supervised fine-tuning without introducing additional geometric signals. A summary of dataset statistics, training steps, and wall-clock time is provided in Table 3.

*Table 3.* Statistics of curated datasets and training computational costs on $8\times$A100 GPUs.

| Model Variant | Initial Samples | Pref. Pairs | Steps | Total Time |
|---|---|---|---|---|
| CogVideoX-I2V-5B (VideoGPA) | 9,441 | 2,663 | 10,000 | |
| CogVideoX-I2V-5B (Epipolar-DPO) | 9,441 | 2,542 | 10,000 | 5d |
| CogVideoX-T2V-5B (VideoGPA) | 8,496 | 2,550 | 10,000 | |
| CogVideoX-T2V-5B (Epipolar-DPO) | 8,496 | 2,330 | 10,000 | |
| CogVideoX1.5-T2V-5B (VideoGPA) | 9,441 | 3,051 | 1,500* | 3d |
| SFT (T2V) | | - | 10,000 | |
| SFT (I2V) | 20,356 (Clips) | - | 10,000 | 2d |

*For CogVideoX1.5-5B, we utilized a shortened schedule consisting of 500 warm-up steps followed by 1,000 optimization steps.

## C. Scalability and VRAM Consumption

We analyze the scalability of the proposed 3D consistency score by varying the number of frames $T$ used for reconstruction. As summarized in Table 4, both runtime and memory consumption increase monotonically with sequence length. While throughput remains competitive for short clips, longer sequences incur higher computational cost and substantially increased VRAM usage.

In our experiments, we empirically select $T = 10$ frames for post-training alignment. This choice is motivated by the temporal length of the base video models, which generate 49 frames (CogVideoX) or 81 frames (CogVideoX1.5). Sampling 10 frames already provides sufficiently dense temporal coverage to capture scene-level geometry, and we do not observe significant gains in geometric consistency when increasing $T$ beyond this range, while incurring substantially higher computational and memory costs.

More broadly, post-training alignment for long video generation remains challenging, as larger temporal windows require proportionally higher memory and compute. Addressing this limitation will likely require advances in more efficient geometric foundation models or scalable reconstruction strategies, and we leave this direction for future work.

*Table 4.* 3D Consistency Score (VGGT): Performance and VRAM scalability analysis by frame count.

| Frames | Avg Time (s) ↓ | FPS ↑ | Peak VRAM (GB) |
|---|---|---|---|
| 5 | 0.3785 | 13.21 | 11.63 |
| 10 | 0.8695 | 11.50 | 13.86 |
| 20 | 2.1460 | 9.32 | 19.84 |
| 40 | 5.9529 | 6.72 | 32.58 |

**Latency and Throughput Comparison:** We assessed the runtime efficiency of 3D consistency score against standard epipolar-based geometric metrics, including Epipolar Sampson Distance using SIFT (CPU-based) and LIGHTGLUE (GPU-based). All experiments are conducted on a single NVIDIA RTX 6000 Ada GPU using sequences of $T = 10$ frames. As shown in Table 5, our dense consistency metric achieves comparable throughput to GPU-based epipolar methods (11.50 FPS vs. 12.10 FPS), despite operating on dense reconstructions rather than sparse correspondences. This demonstrates that reconstruction-based geometric signals can be computed efficiently and are practical for large-scale preference construction.

*Table 5.* **Runtime efficiency comparison** between 3D consistency score and epipolar-based metrics on 10-frame sequences.

| METHOD | TIME (S/VIDEO) ↓ | FPS ↑ |
|---|---|---|
| Epipolar (SIFT) | 3.91 | 2.56 |
| Epipolar (LIGHTGLUE) | **0.83** | **12.10** |
| **3D Consistency Score (VGGT)** | 0.86 | 11.50 |

## D. Analysis of Scene and Motion Fidelity

A common concern in preference fine-tuning is that the model might "collapse" toward simpler, more predictable generations to satisfy specific reward constraints, leading to a loss of texture or movement. To mitigate this, VideoGPA incorporates **KL regularization** within the Direct Preference Optimization (DPO) framework, preventing the model from deviating excessively from the original policy's distribution.

To provide rigorous quantitative evidence that geometric alignment does not come at the cost of visual richness, we evaluate four spatio-temporal metrics across both Image-to-Video (I2V) and Text-to-Video (T2V) settings:

- **Laplacian Variance (LV):** Evaluates texture sharpness and image clarity.

- **FFT High-Frequency Ratio (FHR):** Captures the preservation of fine-grained spatial details.

- **Edge Density (ED):** Indicates structural and compositional complexity.

- **Optical Flow Magnitude (OFM):** Quantifies motion intensity and dynamic range.

As demonstrated in Table 6, all four metrics are either strictly preserved or significantly improved after VideoGPA fine-tuning. Notably, in the I2V setting, we observe a substantial increase in Laplacian Variance (from 299.56 to 839.26), suggesting that our alignment process not only maintains but actually enhances the visual crispness of the generated frames. These results confirm that VideoGPA successfully achieves geometric consistency without suffering from scene complexity or motion degeneration.

*Table 6.* **Scene complexity analysis.** We report Laplacian Variance (LV), FFT High-frequency Ratio (FHR), Edge Density (ED), and Optical Flow Magnitude (OFM) to quantify scene detail and motion intensity. All metrics are preserved or improved after VideoGPA fine-tuning.

| Model | LV ↑ | FHR ↑ | ED ↑ | OFM ↑ |
|---|---|---|---|---|
| | Image-to-Video (I2V) | | | |
| Baseline-I2V | $299.56 \pm 217.91$ | $0.506 \pm 0.054$ | $0.117 \pm 0.053$ | $10.25 \pm 4.77$ |
| **VideoGPA (Ours)** | $\mathbf{839.26 \pm 687.60}$ | $\mathbf{0.558 \pm 0.058}$ | $\mathbf{0.136 \pm 0.059}$ | $\mathbf{11.17 \pm 4.75}$ |
| | Text-to-Video (T2V) | | | |
| Baseline-T2V | $711.13 \pm 374.30$ | $0.585 \pm 0.058$ | $0.113 \pm 0.049$ | $11.46 \pm 5.42$ |
| **VideoGPA (Ours)** | $\mathbf{917.12 \pm 512.53}$ | $\mathbf{0.594 \pm 0.056}$ | $\mathbf{0.115 \pm 0.049}$ | $\mathbf{11.55 \pm 5.61}$ |

# E. Ablation Study on Training Steps

We analyze the effect of post-training duration in the I2V setting by evaluating models fine-tuned for different numbers of optimization steps. For each checkpoint, we prompt the model to generate 150 video samples and compute the metrics reported in Table 7. To ensure consistency with the main experiments, results from the final 10,000-step model are used for comparison with other methods.

As shown in Table 7, the model achieves competitive geometric and perceptual performance as early as 1,000 training steps, with only marginal improvements observed at later checkpoints. Notably, evaluation is conducted using natural, descriptive prompts that differ from the scripted camera-motion prompts used during I2V training, making the validation setting more challenging. Therefore, the limited gains at later stages indicate early convergence of the alignment objective under distribution shift, rather than overfitting to the training prompt format.

*Table 7.* Ablation study on the number of post-training steps in I2V setting.

| Step | 3D Reconstruction Error | | | 3D Consistency | | |
|---|---|---|---|---|---|---|
| | PSNR ↑ | SSIM ↑ | LPIPS ↓ | MVCS ↑ | 3DCS ↓ | Epipolar ↓ |
| Baseline | 18.4336 | 0.6176 | 0.5371 | 0.9662 | 0.5587 | 0.5819 |
| Step-1000 | 19.5250 | 0.6648 | 0.5161 | **0.9829** | 0.5313 | 0.5264 |
| Step-5000 | 19.5244 | 0.6647 | 0.5160 | **0.9829** | 0.5312 | 0.5264 |
| Step-8000 | 19.5250 | 0.6648 | 0.5161 | **0.9829** | 0.5313 | 0.5264 |
| Step-10000 | **19.6853** | **0.6756** | **0.5123** | 0.9822 | **0.5270** | **0.5229** |

Note: Results are evaluated with VGGT backbone.

## E.1. Limitations of Supervised Fine-Tuning for Geometric Grounding

A common paradigm in preference learning is to initialize the policy with Supervised Fine-Tuning (SFT). However, our empirical evidence in Table 8 challenges this assumption for the task of geometric grounding. We compare VideoGPA (DPO-only) against two SFT baselines: (1) SFT on preference winners and (2) a large-scale SFT on 10,000 real-world videos followed by 1,000 steps of DPO.

Our results yield a striking observation: **SFT stages consistently harm the acquisition of geometric priors.** Specifically:

- **DPO vs. SFT on Identical Data:** When trained on the same 2,500 winner samples, SFT-1K performs significantly

worse than VideoGPA, with Epipolar error increasing from $0.509$ to $0.589$. This confirms that imitating "correct" samples is insufficient; the model requires the contrastive penalty provided by DPO to actively distinguish physically plausible motion from geometric artifacts.

- **Sequential Interference:** Even with an extensive SFT stage on 10,000 high-quality real-world videos ($\text{SFT}_{10K} \to \text{DPO}_{1K}$), the performance fails to match the DPO-only configuration. This suggests that SFT on real-world data provides redundant signals that the foundation model has already internalized during pre-training. More critically, the SFT stage appears to *prematurely constrain* the policy, creating an optimization barrier that hinders the subsequent DPO phase from discovering optimal 3D-consistent representations.

These findings indicate that for complex structural constraints like 3D geometry, **direct preference optimization is superior to conventional sequential pipelines.** The imitative nature of SFT lacks the discriminative power necessary for stable geometric grounding, making it less effective—and even counter-productive—compared to the direct distillation of geometric priors via VideoGPA.

*Table 8.* **Ablation on Optimization Strategies.** We evaluate whether SFT aids or hinders geometric alignment. (1) *DPO vs. SFT*: On identical winner samples, DPO significantly outperforms SFT. (2) *Sequential Interference*: Direct DPO training from the base model surpasses the SFT (10K steps on real videos) $\to$ DPO pipeline. These results suggest that **SFT stages may harm geometric priors** by prematurely constraining the policy, making it less effective than direct preference optimization for structural consistency.

| Config | PSNR $\uparrow$ | SSIM $\uparrow$ | LPIPS $\downarrow$ | MVCS $\uparrow$ | 3DCS $\downarrow$ | Epipolar $\downarrow$ | OVL $\uparrow$ |
|---|---|---|---|---|---|---|---|
| CogVideoX-I2V-5B | 22.854 | 0.786 | 0.476 | 0.945 | 0.485 | 0.585 | – |
| SFT-1K (Winners) | 21.285 | 0.772 | 0.489 | **0.957** | 0.498 | 0.589 | 39.0% |
| SFT (10K) $\to$ DPO (1K) | **23.115** | 0.811 | 0.458 | 0.951 | 0.464 | 0.561 | 63.0% |
| **DPO-1K** | 23.045 | **0.831** | **0.438** | 0.954 | **0.445** | **0.509** | **66.0%** |

Reprojection based metrics are calculated with DA3-Large backbone.

### E.2. Foundation Model Backbone Robustness

We further investigate whether the choice of the geometric foundation model backbone influences the preference signals. Specifically, we evaluate the consensus between *VGGT* and alternative backbones, *DUSt3R* (Wang et al., 2024a) and *Depth Anything V3 (DA3)* (Lin et al., 2026), across over 3,000 I2V groups. To quantify this, we define three hierarchical metrics:

- **Agreement Rate (AR):** The percentage of pairs where the backbone agrees with VGGT on the binary winner/loser relationship.

- **Top-1 Consistency (T1):** The probability that the best-performing sample identified by VGGT is also ranked first by the alternative backbone within a triplet.

- **Full Ranking Agreement (FR):** The rate at which the entire ordinal ranking of a group remains identical across backbones.

As shown in Table 9, DA3 achieves 100% consensus with VGGT across all 3,000+ groups. This high degree of alignment suggests that as geometric foundation models evolve, their preference signals converge toward a consistent 3D geometric interpretation, validating the reliability of our supervision source.

*Table 9.* **Preference Agreement Analysis.** We evaluate the consistency of alternative backbones against VGGT's preference pairs across 3,000+ samples. DA3 demonstrates 100% agreement.

| Backbone | AR | T1 | FR |
|---|---|---|---|
| DUSt3R-ViTLarge | 80.8% | 62.5% | 48.5% |
| DA3-Large | 100.0% | 100.0% | 100.0% |

Furthermore, to test the robustness of our DPO-based training against potential supervision noise, we conducted a stress test by randomly flipping up to $20\%$ of the preference labels for 1,000 steps. As reported in Table 10, the model maintains strong performance even with a $20\%$ noise injection, with only a marginal decline in overall preference (OVL). This demonstrates that our pipeline is highly resilient to occasional labeling errors, ensuring stable geometric alignment in practical scenarios.

*Table 10.* **Ablation on Geometric Signal Robustness.** Performance of VideoGPA under different levels of label noise. The consistent gains across all metrics, even with 20% flipped labels, confirm the pipeline's tolerance to potential inaccuracies in geometric supervision.

| Config | PSNR ↑ | SSIM ↑ | LPIPS ↓ | MVCS ↑ | 3DCS ↓ | Epipolar ↓ | OVL ↑ |
|---|---|---|---|---|---|---|---|
| CogVideoX-I2V-5B | 22.854 | 0.786 | 0.476 | 0.945 | 0.485 | 0.585 | - |
| 00% flipped | 23.045 | 0.831 | 0.438 | **0.954** | 0.445 | **0.509** | **66.0%** |
| 10% flipped | **23.380** | **0.834** | **0.436** | 0.952 | **0.443** | 0.516 | **66.0%** |
| 20% flipped | 23.253 | 0.832 | 0.439 | 0.951 | 0.446 | 0.523 | 63.0% |

Reprojection based metrics are calculated with DA3-Large backbone.

# F. Generalization on Out-of-Distribution Datasets

To evaluate the zero-shot generalization capability of VideoGPA, we conduct experiments on two distinct out-of-distribution (OOD) datasets: *WebVid* (Bain et al., 2021) and *Panda-70M* (Chen et al., 2024). These datasets are disjoint from our training set and feature significantly different visual distributions and complex motion dynamics.

## F.1. Quantitative Performance on WebVid

We evaluate 100 randomly sampled videos from WebVid (Bain et al., 2021), comparing VideoGPA against the baseline, SFT, and Epipolar-DPO. As shown in Table 11, VideoGPA achieves the best performance in *LPIPS*, *3DCS*, and *Epipolar* error. These results confirm that our geometric consistency improvements effectively generalize to OOD web-scale content.

*Table 11.* **Quantitative evaluation on WebVid.** We report 3D reconstruction and geometric consistency metrics. VideoGPA achieves the best LPIPS, 3DCS, and Epipolar scores, confirming that geometric consistency improvements generalize effectively across datasets.

| Config | PSNR ↑ | SSIM ↑ | LPIPS ↓ | MVCS ↑ | 3DCS ↓ | Epipolar ↓ |
|---|---|---|---|---|---|---|
| Baseline-I2V | 17.434 | 0.598 | 0.542 | 0.966 | 0.582 | 1.282 |
| SFT | **20.453** | 0.689 | 0.470 | **0.975** | 0.486 | 0.748 |
| Epipolar-DPO | 19.445 | **0.710** | 0.443 | 0.972 | 0.462 | 0.699 |
| **VideoGPA (Ours)** | 18.767 | 0.693 | **0.435** | 0.972 | **0.460** | **0.602** |

Reprojection based metrics are calculated with DA3-Large backbone.

## F.2. Generalization to Motion Coherence (Panda-70M)

A critical challenge in video generation is maintaining **motion coherence** in highly dynamic scenes. We tested VideoGPA on 100 dynamic videos from *Panda-70M* (Chen et al., 2024), which contain complex object movements and camera transitions absent from our relatively static-scene training data.

As summarized in Table 12, VideoGPA significantly outperforms the baseline across all VideoReward metrics, including a substantial gain in *Overall Preference* (OVL: 64.0%). This improvement on OOD dynamic sequences suggests that VideoGPA does not merely memorize static geometric patterns; instead, it enforces a **temporally coherent geometric prior** that stabilizes motion even in highly dynamic contexts. We will further explore in Appendix H.

*Table 12.* **VideoReward on Panda-70M.** Evaluated on 100 dynamic videos. VideoGPA significantly outperforms the baseline in zero-shot settings. This improvement in dynamic scenes (absent from static-scene training) supports the hypothesis that the model learns generalizable geometric priors.

| Metric | VQ | MQ | TA | OVL |
|---|---|---|---|---|
| Baseline-I2V | – | – | – | – |
| **VideoGPA (Ours)** | **61.0%** | **61.0%** | **51.0%** | **64.0%** |

# G. Additional Evaluation on Wan Family

To examine whether VideoGPA generalizes to other video diffusion architectures, we evaluate it on **Wan2.2-TI2V-5B** (Team Wan, 2025). We follow the same protocol used for our CogVideoX experiments: preference pairs are constructed from videos generated under scripted static-scene camera-motion prompts (Sec. A).

**Static-scene camera-motion evaluation.** The picture changes once we evaluate Wan2.2 on the same scripted static-scene prompts used during preference data construction, a setting that explicitly stresses 3D geometric consistency under camera motion. As shown in the top half of Table 13, the base Wan2.2 model degrades noticeably in this regime, and VideoGPA recovers substantial performance: PSNR improves from 19.40 to 24.41, LPIPS drops from 0.480 to 0.394, and Epipolar error reduces from 0.594 to 0.499, alongside consistent gains across all 3D consistency metrics. These geometric improvements also translate to stronger human-aligned quality, with VideoGPA achieving VideoReward win rates of 52.00% in Visual Quality (VQ), 65.00% in Motion Quality (MQ), and 57.00% in Overall preference (OVL).

**VLM generated caption evaluation.** As shown in the bottom half of Table 13, under natural generated video captions where the base model is already strong, VideoGPA still yields modest improvements in reconstruction quality and most consistency metrics, including PSNR ($24.06 \rightarrow 24.41$), SSIM ($0.815 \rightarrow 0.823$), LPIPS ($0.412 \rightarrow 0.401$), MVCS ($0.941 \rightarrow 0.945$), and 3DCS ($0.424 \rightarrow 0.411$), though Epipolar error shows a slight increase ($0.588 \rightarrow 0.611$). In terms of human-aligned quality, VideoGPA achieves VideoReward win rates of 47.00% in Visual Quality (VQ), 67.00% in Motion Quality (MQ), and 56.00% in Overall preference (OVL). Together, these results suggest that VideoGPA provides the most pronounced geometric benefits in the camera-motion regime where modern video models remain weakest, while consistently improving performance under unconstrained natural language prompts.

*Table 13.* Quantitative evaluation of VideoGPA on **Wan2.2-TI2V-5B** under two prompt regimes: (top) scripted static-scene prompts with explicit camera-motion descriptions, which directly stress 3D geometric consistency; and (bottom) natural video captions generated by CogVLM2, on which the base model is already strong. VideoGPA substantially improves the base model under scripted camera-motion prompts and yields consistent, albeit modest, improvements under natural captions.

| Method | 3D Reconstruction Error | | | 3D Consistency | | | VideoReward (Win Rate %) | | | |
|---|---|---|---|---|---|---|---|---|---|---|
| | PSNR ↑ | SSIM ↑ | LPIPS ↓ | MVCS ↑ | 3DCS ↓ | Epipolar ↓ | VQ | MQ | TA | OVL |
| *Scripted static-scene camera-motion prompts* | | | | | | | | | | |
| Baseline (Wan2.2) | 19.40 | 0.723 | 0.480 | 0.898 | 0.506 | 0.594 | - | - | - | - |
| **VideoGPA (Ours)** | **24.41** | **0.842** | **0.394** | **0.944** | **0.451** | **0.499** | **52.00** | **65.00** | 35.00 | **57.00** |
| *Natural CogVLM2-generated video captions* | | | | | | | | | | |
| Baseline (Wan2.2) | 24.06 | 0.815 | 0.412 | 0.941 | 0.424 | **0.588** | - | - | - | - |
| **VideoGPA (Ours)** | **24.41** | **0.823** | **0.401** | **0.945** | **0.411** | 0.611 | 47.00 | **67.00** | 45.00 | **56.00** |

Reprojection based metrics are calculated with DA3-Large backbone.

## H. Emergent Consistency in Dynamic Scenes

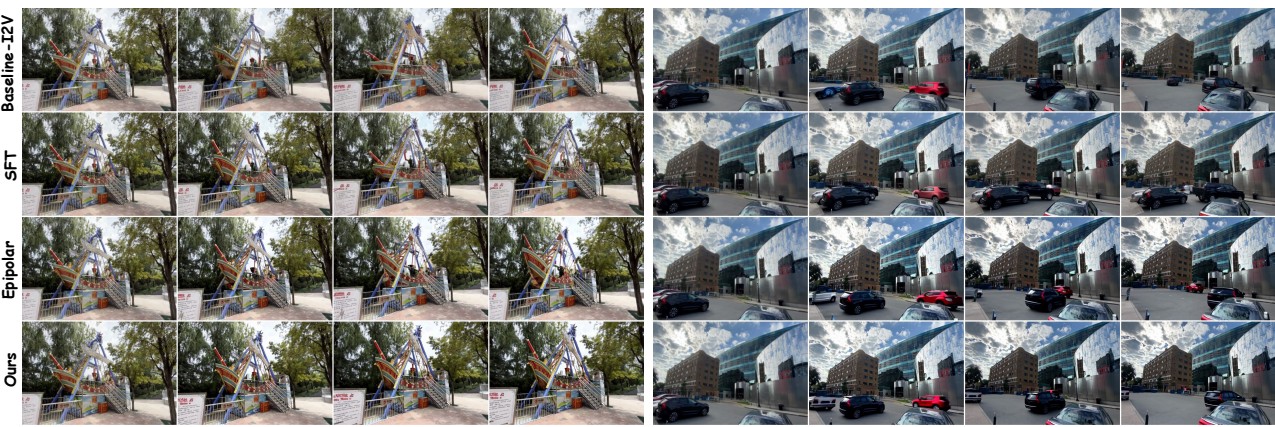

(a) Colorful pirate rides in a serene and empty outdoor park.    (b) Modern glass building in urban setting surrounded by vehicles.

*Figure 7.* **Examples of dynamic object generation in unconstrained scenes.** When prompted with dynamic object without static scene constraints, (a) VideoGPA is the only approach that preserves the rigid-body integrity of the pirate ride during spinning. (b) Our method successfully maintains the physical plausibility of the vehicle throughout its trajectory and ensures color consistency for the red car after occlusion occurs, whereas baseline models exhibit color drift or structural distortion.

In our validation tests, we observe that when the static-scene constraint is relaxed, the model occasionally generates moving

objects based on their natural semantics, such as a spinning pirate ride or moving vehicles. In these cases, VideoGPA significantly improves the coherence of object motion, even in complex scenarios that differ substantially from the static scenes used during training. As illustrated in Fig. 7, when prompted with dynamic subjects, the model exhibits superior **temporal stability** and **physical plausibility**.

**The Disentanglement Hypothesis** Initially, we hypothesized that these improvements might arise from a trivial background-foreground disentanglement effect. We posited that by enforcing background rigidity, the model was essentially "faking" coherent motion by constraining foreground objects to satisfy local multi-view consistency against a stabilized background.

**Experimental Discovery.** However, our extensive experiments in both Image-to-Video (I2V) and Text-to-Video (T2V) settings provided results that overturned this initial intuition. As demonstrated in Sec. H.1, the observed improvements represent a motion-coherent level advancement rather than a simple static multi-view consistency fix. Remarkably, even in the presence of highly articulated or non-rigid motions, such as animal limb movements, ear morphing, VideoGPA effectively suppresses chronic artifacts like "geometry collapse" and "semantic metamorphosis." These findings provide compelling empirical evidence for our **Video Motion Manifold** hypothesis (Sec. 5.2). By regularizing the model toward a 3D-consistent manifold, we inherently facilitate more physically-grounded generative behaviors. Even historically "hard" cases, such as intricate limb inconsistency or objects splitting and merging, are rectified. This proves that our geometric alignment effectively reshapes the underlying motion manifold of the video diffusion model.

### Further Discussion: Physical Plausibility Improvement.

Additionally, we observe that VideoGPA significantly enhances the **physical plausibility** of the generated videos. We hypothesize that this improvement is a beneficial side effect of distilling knowledge from the 3D reconstruction model. Since VGGT (Wang et al., 2025) is pre-trained on large-scale 3D datasets and DINO (Caron et al., 2021; Oquab et al., 2024) possesses strong visual understanding and segmentation capabilities, the reconstruction model exhibits a strong geometric prior. This bias encourages the generation of point clouds that align with physically plausible object structures. Consequently, leveraging this geometric signal as a ranking criterion guides the model toward producing objects with more realistic physical properties, a capability fundamentally absent in frame-level constraint such as Epipolar (Kupyn et al., 2025) or Vision-Language-Model based metrics (Li et al., 2025c;b).

### H.1. Qualitative Analysis of Complex Motion Coherence

The following qualitative comparisons evaluate motion coherence in animal movements across our I2V and T2V experimental settings. These scenarios are widely considered the most challenging benchmarks for both closed-source and open-source video generation models. Note that we exclude GeoVideo as its released checkpoint is specialized for static environments and does not yield plausible dynamic scene generation. Additional high-resolution results can be found on the anonymous project webpage, with the link provided as a separate file in the supplementary package.

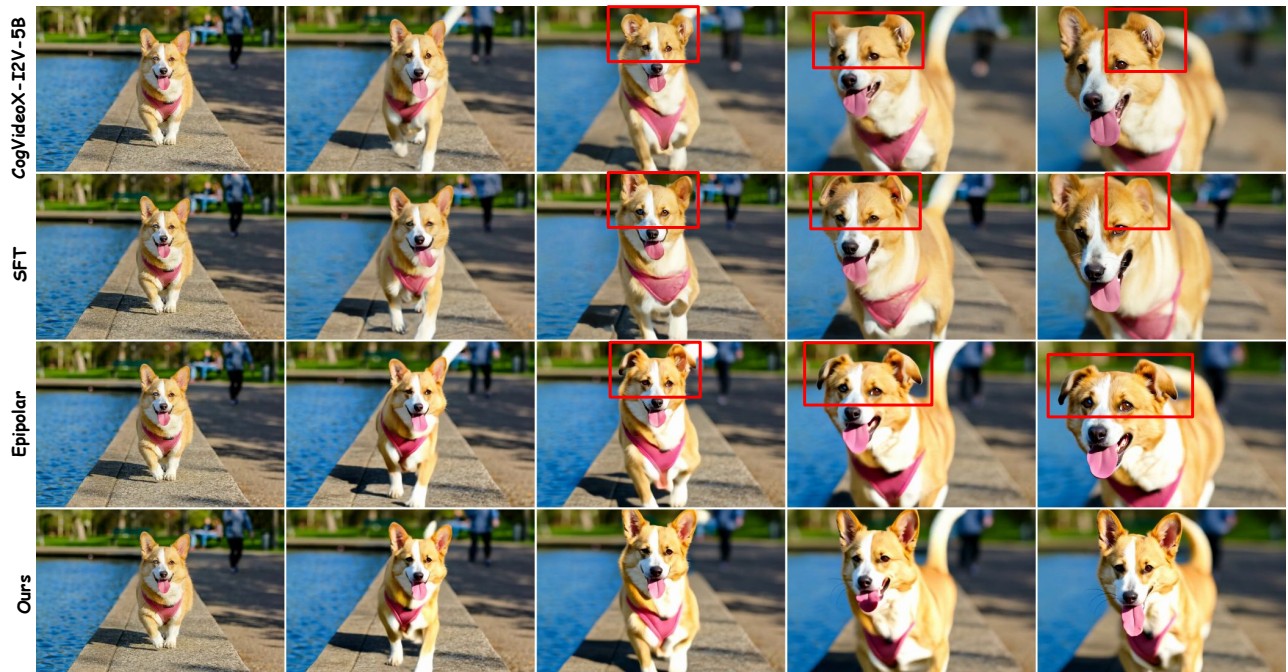

The video shows a dog walking directly toward the camera with a calm and steady gait. Its eyes are fixed forward, and its tail moves slightly, suggesting curiosity or familiarity. The setting is softly lit, with the dog's approach gradually filling the frame and creating a sense of intimacy and engagement.

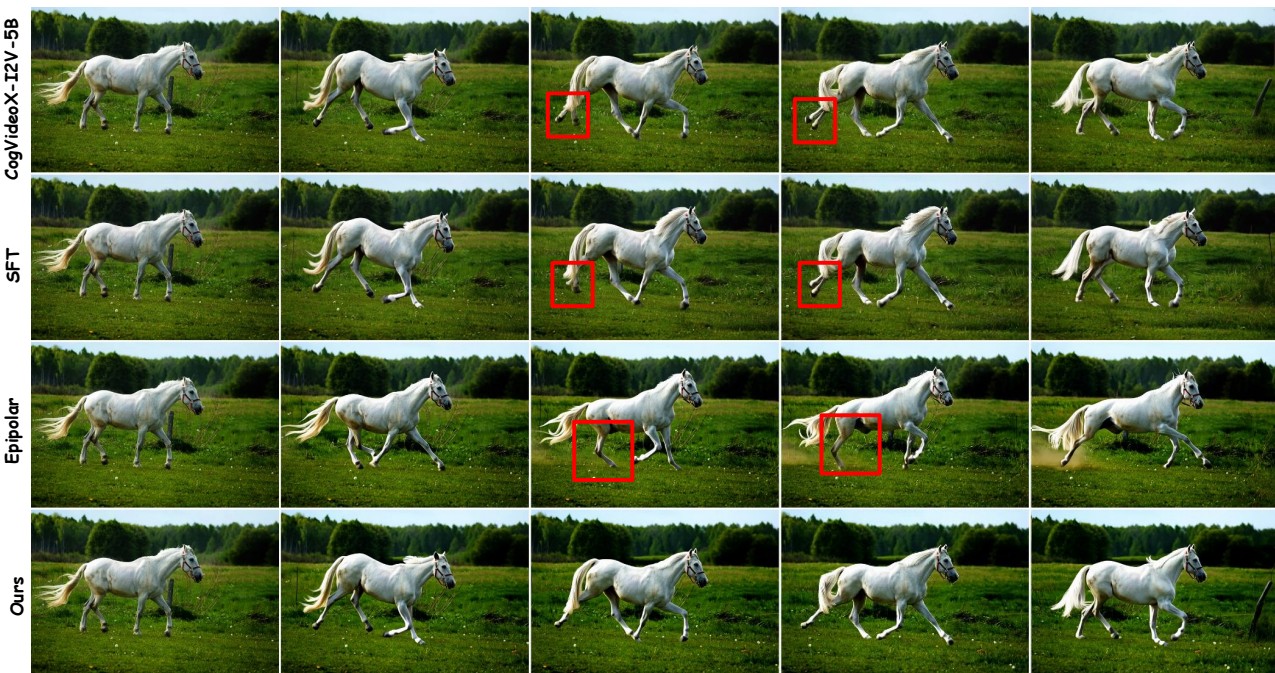

The video captures a white horse in mid-run, its hooves kicking up dust as it moves across a sun-drenched field. The flowing mane and tail add to the sense of movement, while the soft lighting and open landscape highlight the horse's form and motion.

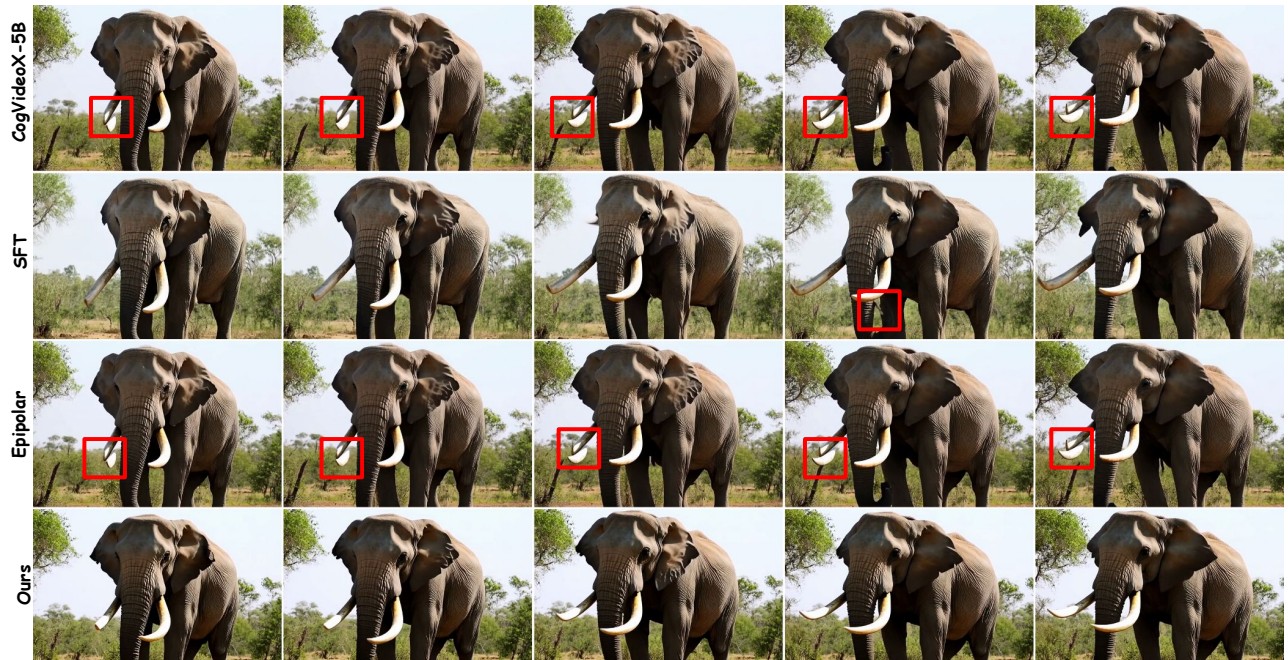

A majestic elephant with large tusks and wrinkled grey skin is captured in a natural setting, surrounded by sparse vegetation and a few trees. The elephant's massive size and textured skin are highlighted against the backdrop of a clear sky, suggesting a warm, sunny day. As the elephant moves slowly, its trunk sways gently, and its ears flap occasionally, showcasing its calm demeanor. The scene conveys a sense of tranquility and the elephant's harmonious existence within its environment. The overall atmosphere is one of peacefulness and respect for this gentle giant of the wild.

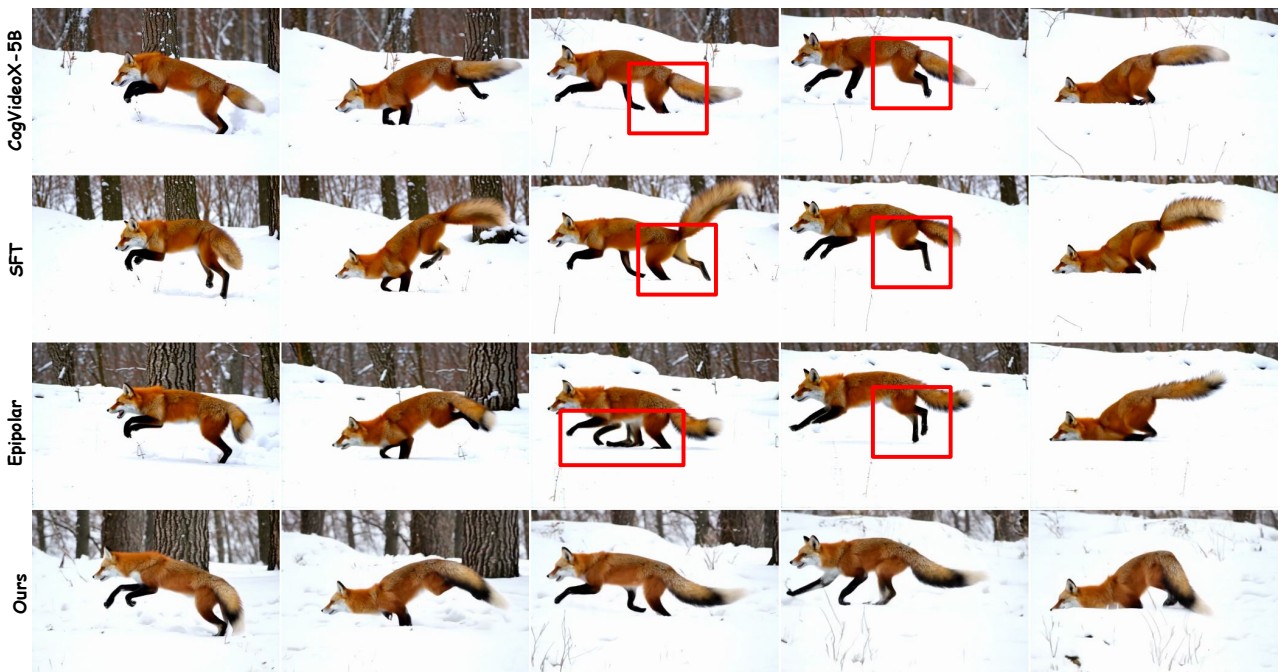

A vibrant red fox with a bushy tail and sharp features is captured mid-leap in a snowy forest. The fox's fur contrasts strikingly against the white snow, while the surrounding trees, bare of leaves, create a stark and wintry backdrop. As the fox jumps gracefully, its agile movements are highlighted, showcasing its natural athleticism and adaptability to the cold environment. The scene conveys a sense of energy and vitality, with the fox's focused expression and dynamic pose emphasizing its role as a skilled hunter in the wilderness. The overall atmosphere is one of crispness and liveliness amidst the winter landscape.

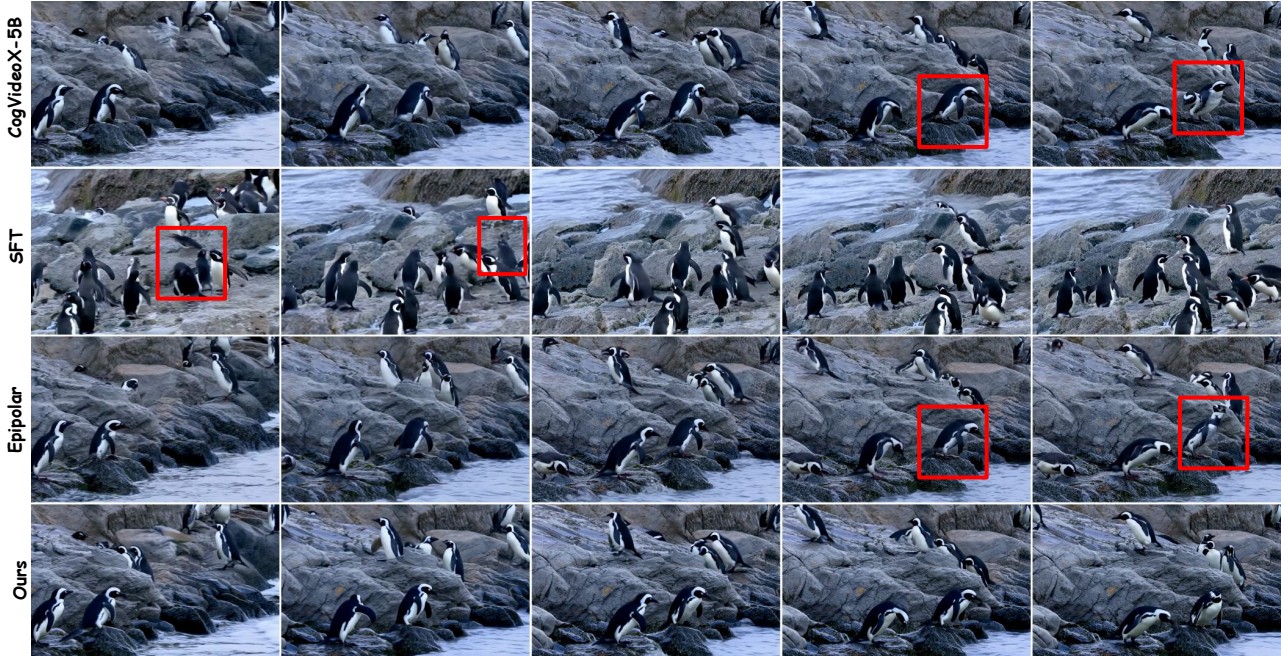

A group of penguins is seen navigating a rocky shoreline, engaging in various activities such as standing, walking, and interacting with each other. The penguins' black and white plumage contrasts against the rugged, grey rocks and the soft blue hues of the surrounding water. The scene captures the penguins' social behavior and their adaptation to the harsh coastal environment. As they move about, the lighting suggests a cool, overcast day, enhancing the natural ambiance of their habitat. The overall mood is one of camaraderie and resilience among these seabirds in their challenging surroundings.",

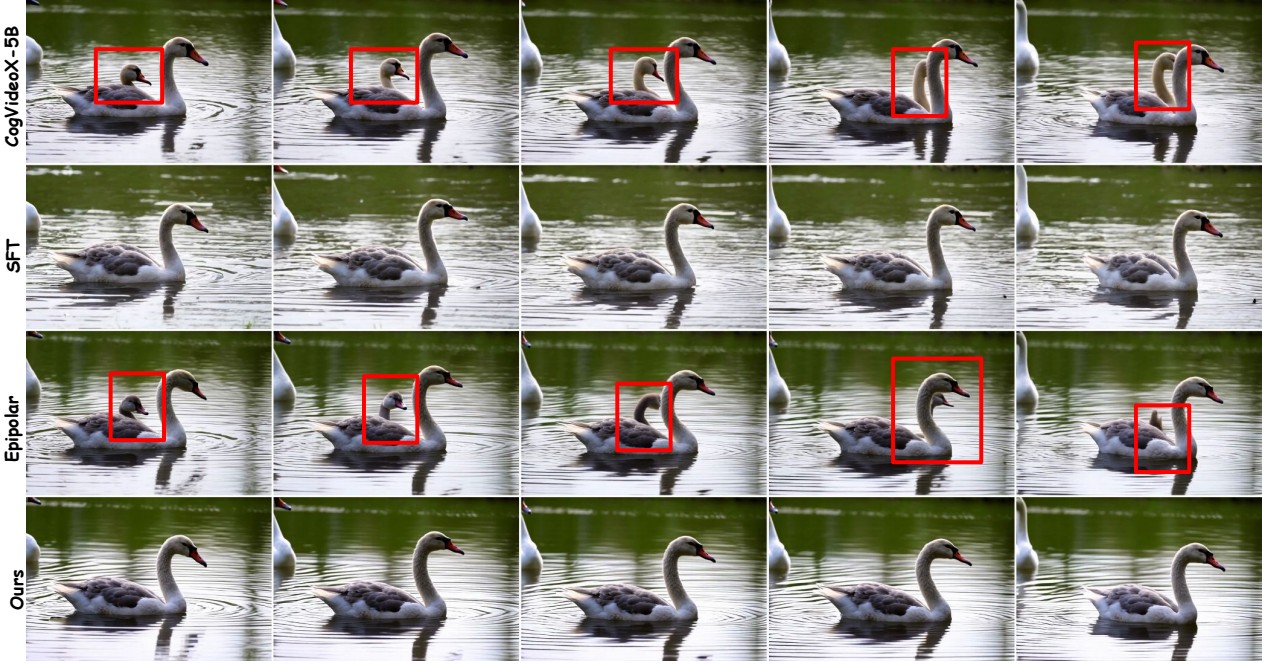

A juvenile swan with grey and white plumage is seen engaging in playful behavior in a tranquil pond, repeatedly dipping its beak and feet into the water. The swan's actions are highlighted by the gentle ripples on the water's surface and the soft lighting, which enhances the serene atmosphere. As time passes, another swan, possibly an adult, appears in the background, observing the young swan's antics. Eventually, a group of juvenile swans is shown, with one engaging in playful behavior while another watches, against a backdrop of lush greenery, suggesting a peaceful, natural habitat.

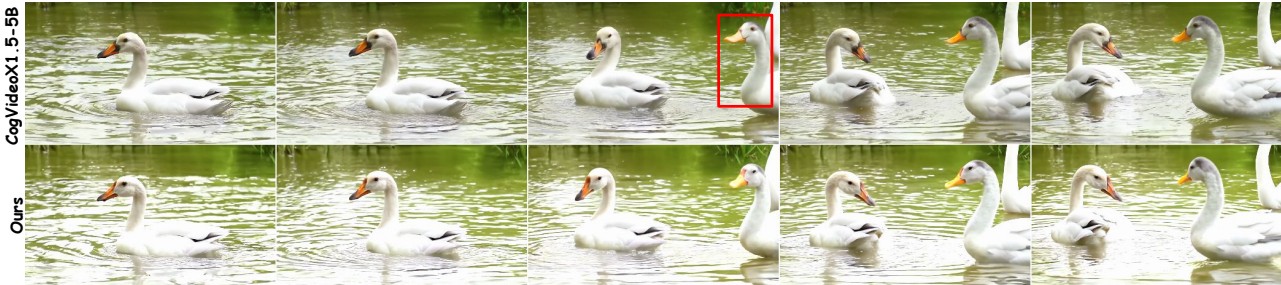

Two seagulls are observed on a tranquil beach, engaging in various activities such as standing, walking, and foraging in the shallow waters. The scene is serene, with the seagulls' white and grey feathers contrasting against the muted beige sands and soft blue-green hues of the sea. The overcast sky enhances the calm atmosphere, contributing to the absence of human presence. Gentle waves wash over the shore, and the seagulls' relaxed postures suggest a peaceful coexistence within their natural coastal habitat. The lighting indicates it might be early morning or late afternoon, adding to the quietude of the setting.",

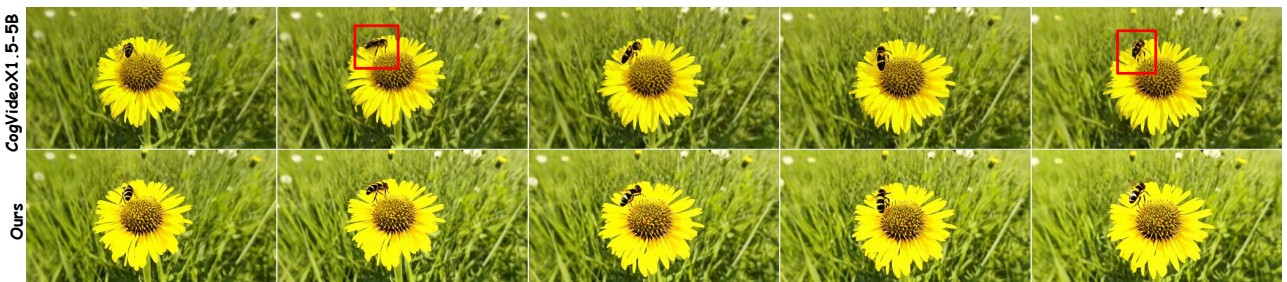

A juvenile swan with grey and white plumage is seen engaging in playful behavior in a tranquil pond, repeatedly dipping its beak and feet into the water. The swan's actions are highlighted by the gentle ripples on the water's surface and the soft lighting, which enhances the serene atmosphere. As time passes, another swan, possibly an adult, appears in the background, observing the young swan's antics. Eventually, a group of juvenile swans is shown, with one engaging in playful behavior while another watches, against a backdrop of lush greenery, suggesting a peaceful, natural habitat.",

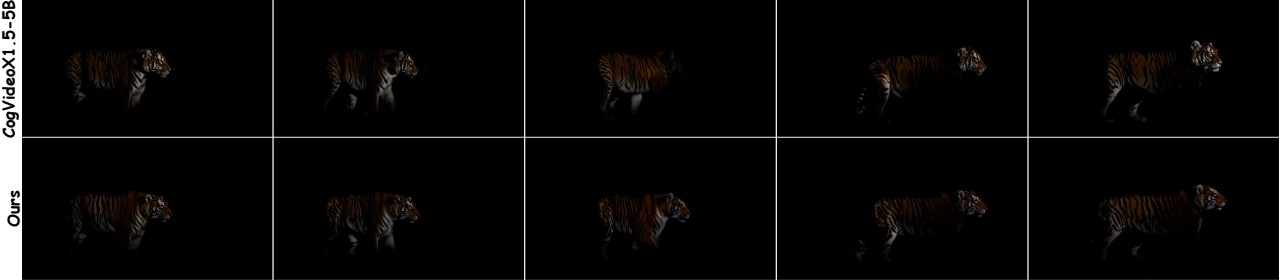

A wasp with yellow and black markings is seen pollinating a dandelion flower amidst a lush meadow. The scene unfolds under the bright midday sun, highlighting the vivid colors and intricate details of the wasp and flower. As time passes, the wasp continues its pollination efforts, with the background of green grass and other dandelions softly blurred, emphasizing the serene and dynamic interaction between the insect and its floral host. The video captures the essence of nature's daily ballet, showcasing the wasp's delicate movements and the tranquil beauty of the meadow.

A tiger stands in stark contrast against the darkness, its silhouette and striped fur highlighted by a soft light that casts deep shadows, creating an atmosphere of mystery and solitude. The tiger's gaze is consistently directed off-camera, suggesting a silent vigilance and a serene yet powerful presence. The scene remains devoid of any other elements, focusing solely on the tiger's majestic form and the dramatic interplay of light and shadow. The tiger's poised stance and the subtle interplay of light and shadow accentuate its solitary figure, evoking a sense of quiet anticipation and the wild's enigmatic allure.",

# I. Additional Qualitative Results

### I.1. Image-to-Video

We present additional image-to-video results in this subsection. Compared to the base model, SFT, and Epipolar-DPO, VideoGPA produces more stable geometry under camera motion, with reduced spatial drift and texture flickering.

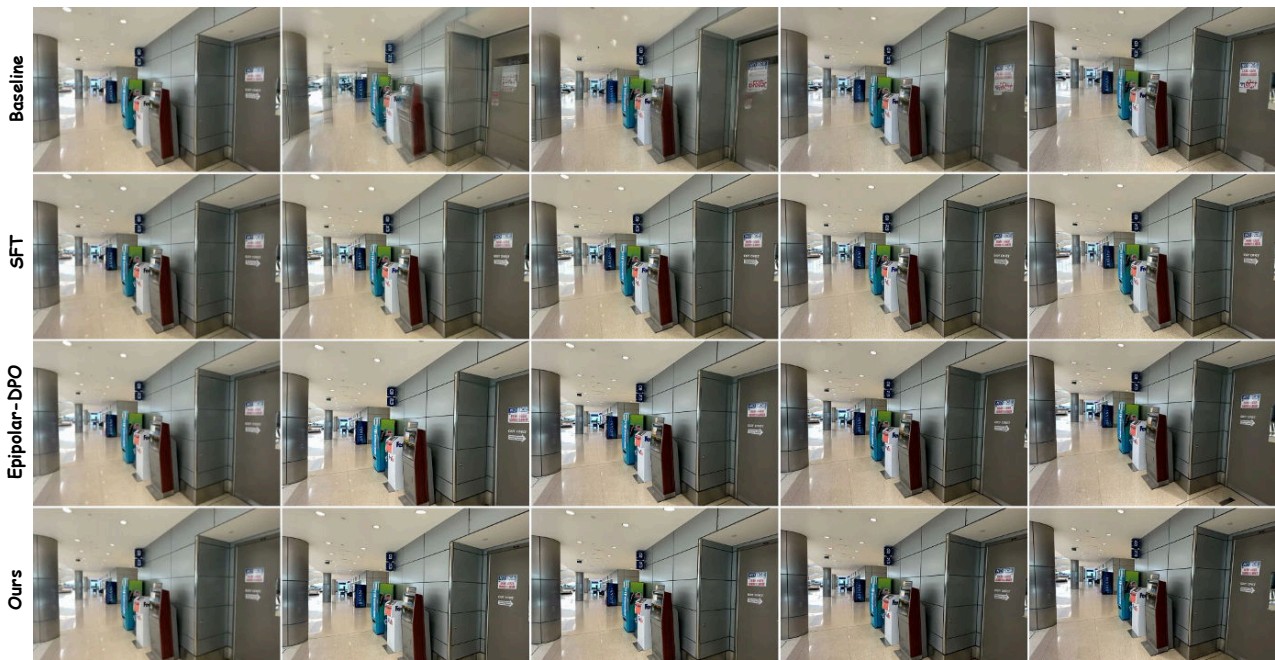

The video features a series of vending machines in an airport terminal, including a colorful lottery machine, a FedEx self-service kiosk, and various other machines like a Pepsi vending machine and a UPS store. The machines are set against a backdrop of modern architecture with reflective floors, bright lighting, and signs indicating accessible restrooms and elevators. The area is quiet and well-lit, with no visible activity, suggesting a moment of stillness in a bustling transit space. The video captures the essence of a busy yet orderly airport environment.

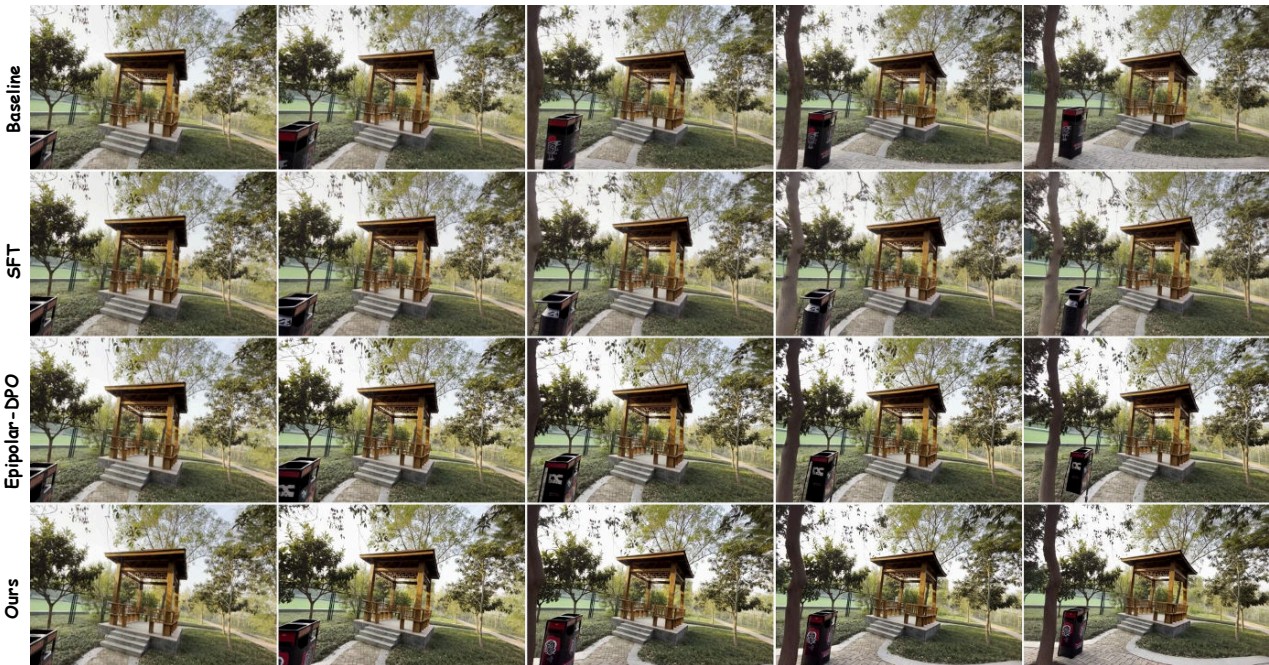

A traditional Chinese-style wooden gazebo with a curved roof and ornate latticework is situated in a well-maintained park, surrounded by lush greenery and a winding brick pathway. The scene is tranquil, with a tennis court visible in the background, suggesting a recreational setting. As time passes, the gazebo remains a focal point, with the surrounding trees and the pathway leading to it. The lighting indicates it might be early morning or late afternoon. A modern trash bin with a red and black design appears, indicating an emphasis on cleanliness. The park is serene, with no people or animals present.

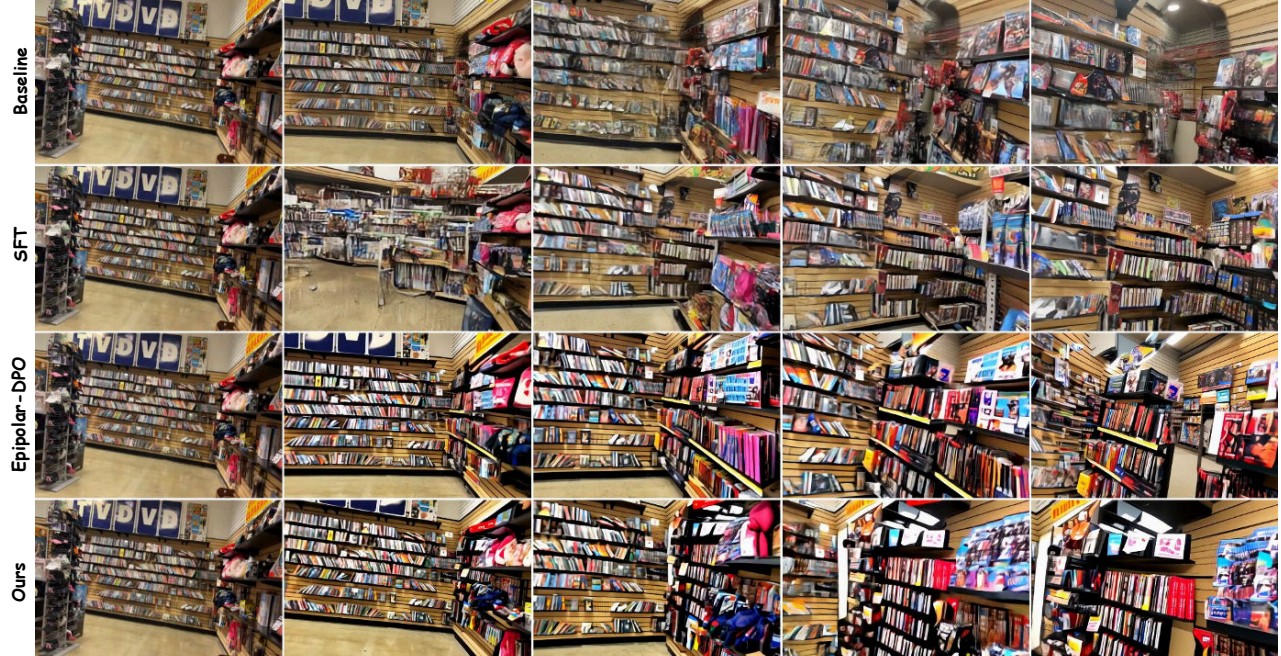

The video takes us through a vibrant video game store, starting with a view of the store's interior, showcasing DVDs, video games, and collectibles. As we move through the store, we see a variety of items including baseball caps, plush toys, and action figures, with clearance signs indicating sales. The store features a range of posters, including 'TRENDPOSTERS' and 'SUPERSTAR POSTERS', and a section dedicated to gaming accessories. The shelves are well-organized, displaying DVDs, Blu-rays, and video games, with a focus on the entertainment industry. The store's atmosphere is inviting, with a warm color palette and a sense of excitement for customers.

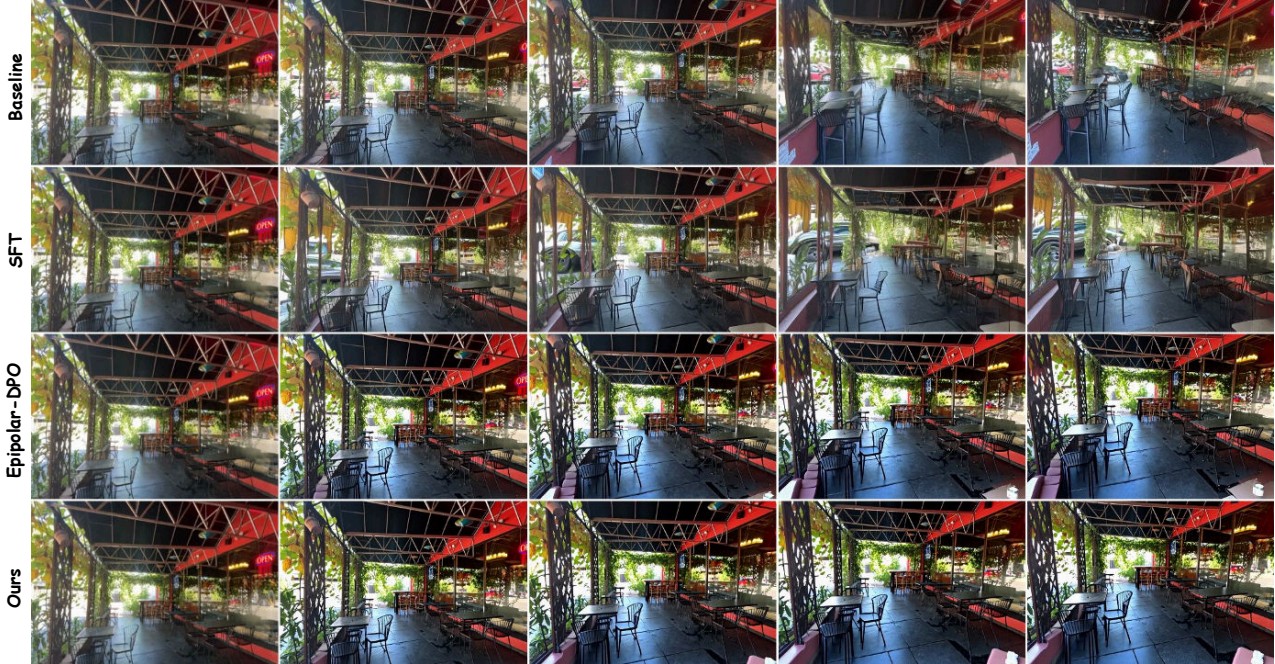

The video features an outdoor dining area with a black metal truss canopy and greenery, set against a red wall with large windows reflecting the interior. Initially, the area is empty, with a red pickup truck parked outside. As time passes, the scene includes a red bench, a black table with chairs, and a high-top table with red cushioned chairs. The area is well-lit by pendant lights and natural light, with a 'NO ENTRY' sign and a 'CLOSED' sign indicating the establishment's status. The setting is tranquil, with a red pickup truck and a black car parked nearby, and a 'PARKING' sign suggests restrictions.

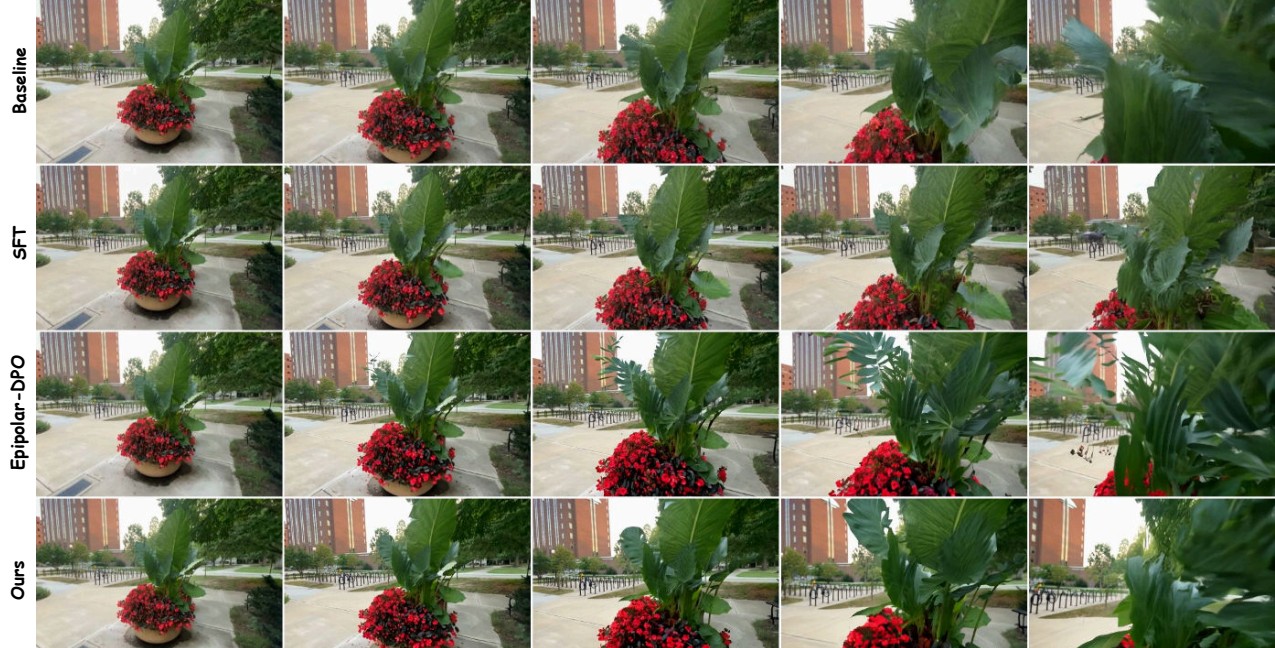

The video features the Monument to the People's Heroes in Beijing, China, under a clear blue sky. The monument, with its grey stone base and golden Chinese characters, is surrounded by a white railing and lush greenery. As the sun casts a soft glow, the scene remains tranquil and devoid of people, emphasizing the monument's solemnity and historical significance. The inscription on the monument's base changes slightly, indicating different eras or aspects of the monument's history. The serene atmosphere is maintained throughout, with the monument's grandeur highlighted by the sunlight.

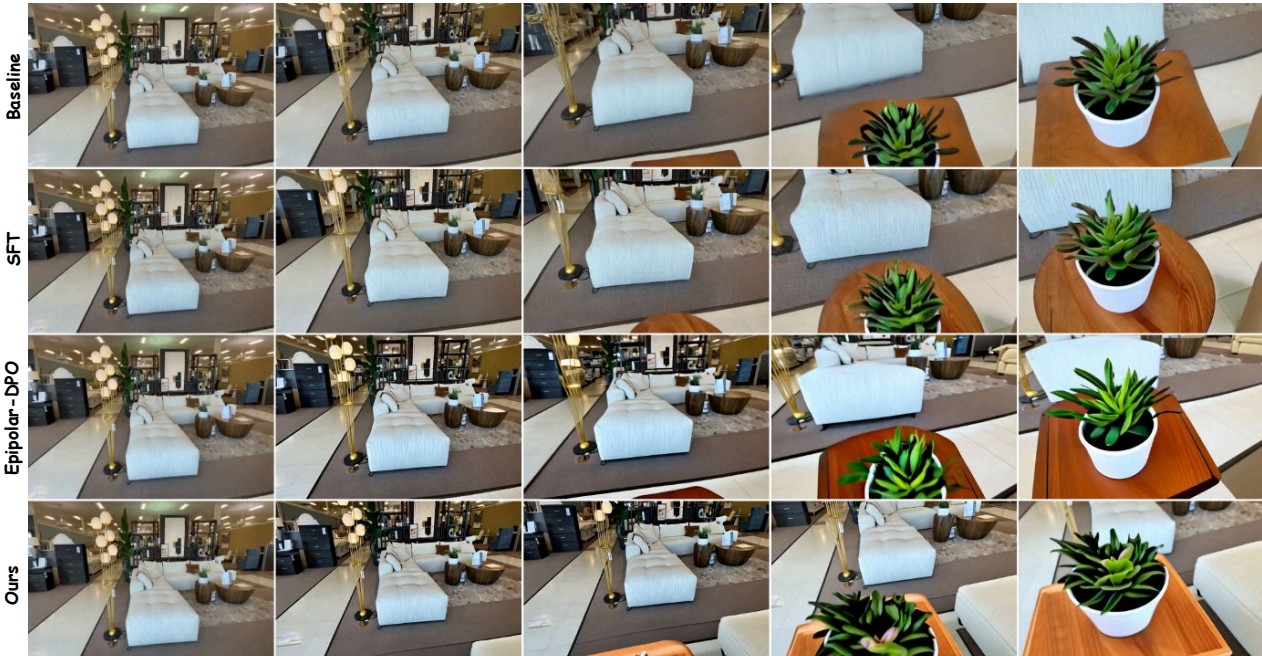

The video takes viewers through a furniture store, showcasing a variety of modern and stylish furniture pieces. Initially, a white sectional sofa with plush cushions is highlighted, surrounded by a dark wood coffee table and a shelving unit with decorative items. As the video continues, different sections of the store are featured, including a cream-colored sofa with a chaise lounge, a round wooden coffee table, and a shelving unit with vases and books. The store's ambiance is warm and inviting, with a focus on contemporary furniture and decorative elements like potted plants and abstract art. The video concludes with a display of a white ceramic pot with green succulents on a wooden stool, surrounded by other furniture pieces.

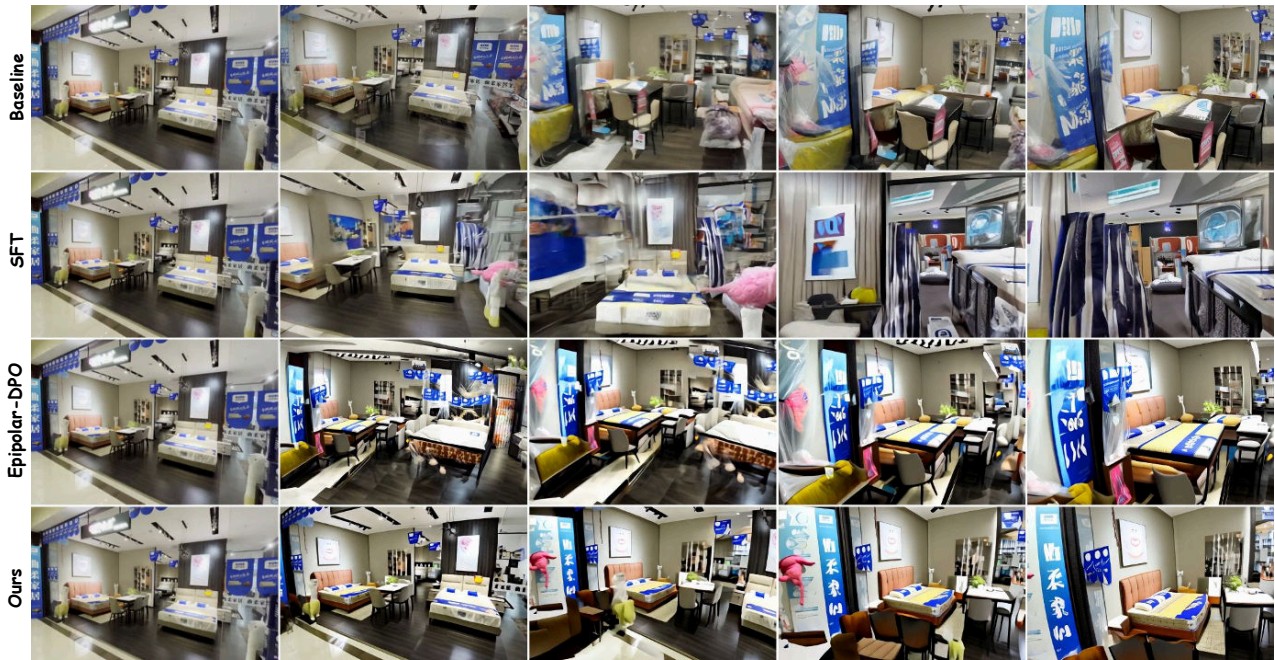

The video takes place in a modern furniture store, showcasing a variety of beds with different headboards and mattresses, some wrapped in plastic. The store features a dining set, a kitchenette, and a living area, all under warm lighting. Decorative elements like abstract art, a pink flamingo sculpture, and a plush ostrich toy add whimsy. The store displays beds with blue and white branding, including 'OYO' and 'QUBO', and offers promotional materials. The ambiance is contemporary, with a color scheme of neutral tones and warm lighting, creating an inviting atmosphere for customers.

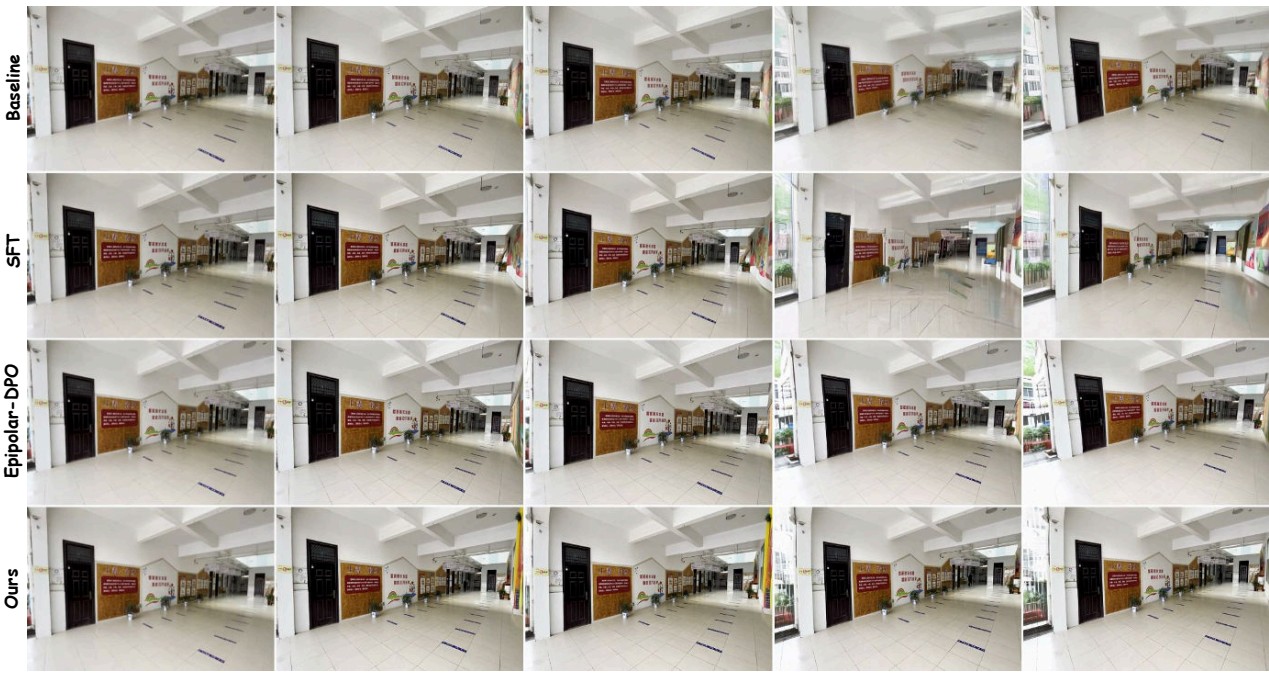

The video takes place in a well-lit indoor corridor with a polished floor and high ceilings, adorned with potted plants and educational posters. A red banner with Chinese characters is visible, along with a bulletin board displaying information and photographs. The corridor is empty, with blue social distancing stickers on the floor and a sign indicating the entrance to a building. As the video continues, the corridor is shown with a large window, a door with a red sign, and a staircase. The walls feature colorful murals and a 'Welcome' sign, with a few individuals seen in the distance. The setting suggests a school or community space with a focus on safety and orderliness.

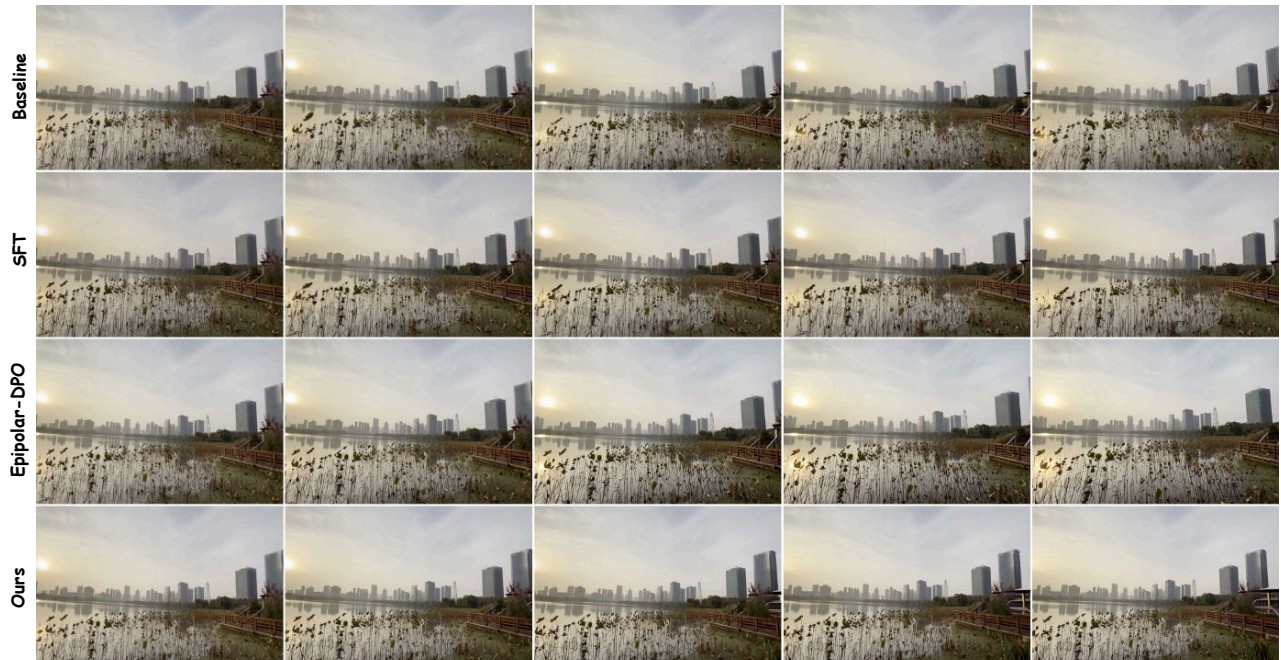

The video features a tranquil lakeside scene with a wooden boardwalk leading to a gazebo, surrounded by lotus flowers and aquatic plants. The calm waters reflect the soft glow of the sun, either rising or setting, and the hazy skyline of a city with modern skyscrapers. The scene is serene, with no people or wildlife, and the lighting suggests it's either early morning or late afternoon. As time passes, the sky transitions from pale blue to warm orange, and the city's silhouette becomes more pronounced against the natural backdrop. The video captures the peaceful coexistence of nature and urban life.

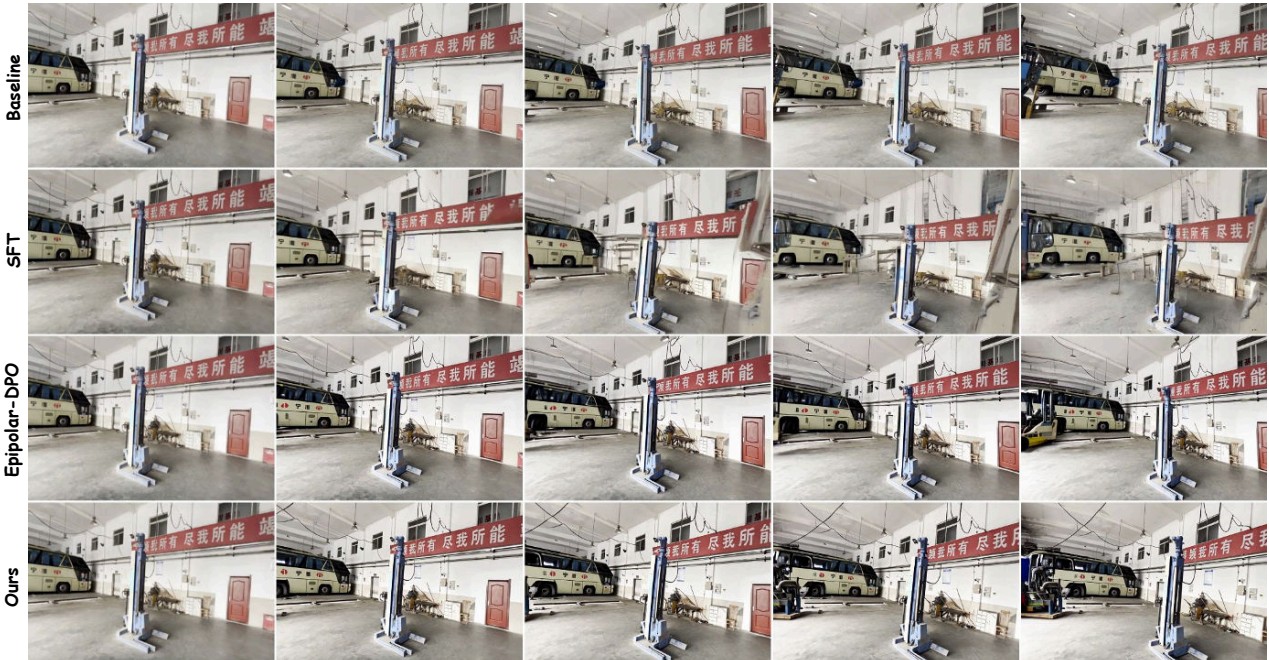

The video takes place in a well-lit industrial workshop with a focus on bus maintenance. Initially, a blue and white hydraulic lift is central, with a red banner and a bus under maintenance. The scene shifts to show the lift with a control panel, a red banner with Chinese characters, and a bus with a damaged front end. Various equipment like a manual pallet jack, a red fire extinguisher, and a wooden bench appear, indicating ongoing work. A technician is seen working on a bus, with tools and parts scattered around. The workshop is equipped with multiple lifts, a forklift, and a bus undergoing repair, with a red banner and a sign with Chinese characters visible. The video concludes with a still scene of the workshop, highlighting the bus and the mechanical equipment.

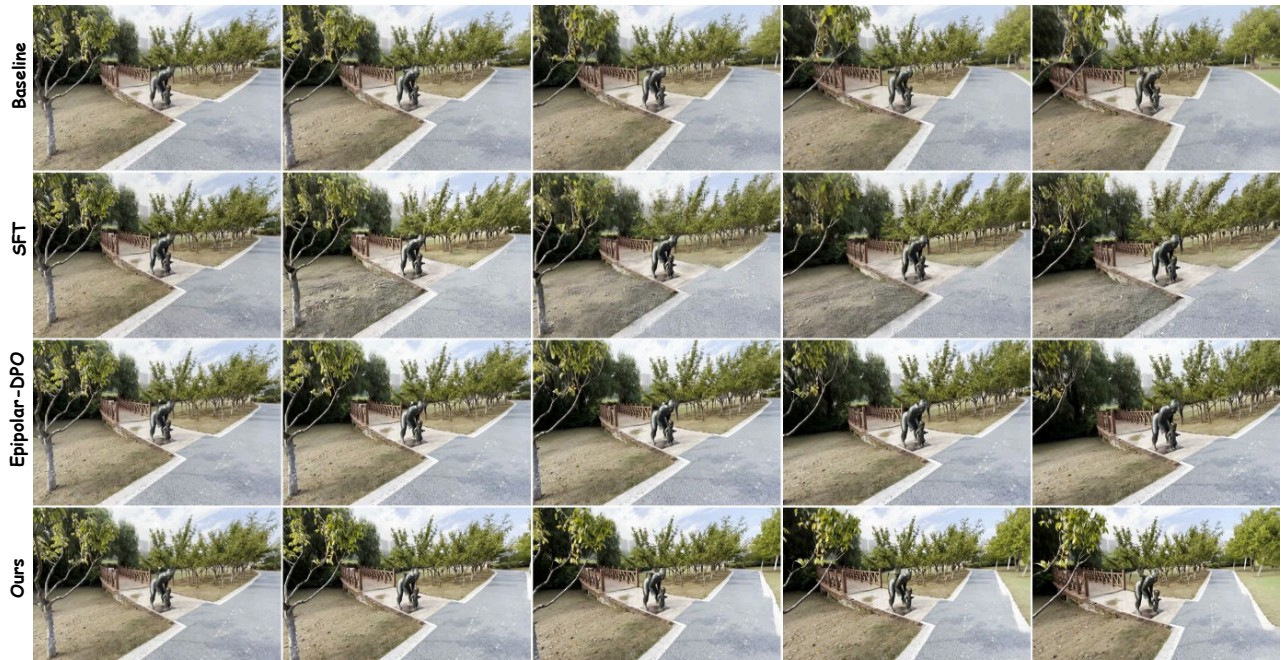

The video features a series of bronze statues in a park, each depicting an adult and a child in various intimate moments, suggesting a theme of guardianship and learning. The statues are set against a backdrop of a wooden bridge, lush greenery, and a clear sky, with a pathway leading to the bridge. The scenes are tranquil, with no people present, and the soft lighting indicates it might be early morning or late afternoon. The presence of a logo in some frames hints at an association with a media outlet or organization.

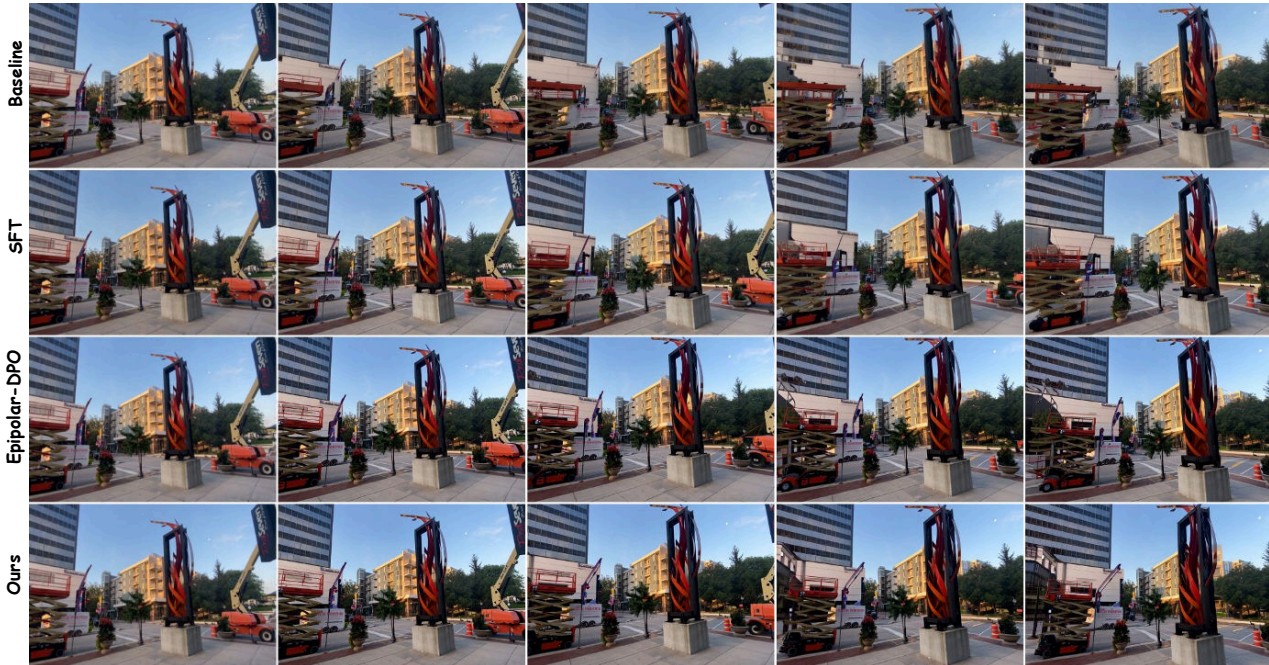

The video features a serene urban setting with a red and black abstract sculpture on a concrete base, surrounded by modern buildings and a clear blue sky. As time passes, the scene includes a cherry picker, a 'Parkway' sign, and a 'Downtown' sign, suggesting maintenance or installation activities. The sculpture's dynamic design contrasts with the tranquil environment. A cherry picker lifts a street sign, and a 'Parkway' sign is visible. The cityscape is quiet, with no people, and a 'Chase' building is seen in the background. The video concludes with a 'PARK' sign and a 'Chase' building, with a cherry picker indicating maintenance work.uare with maintenance cherry picker.

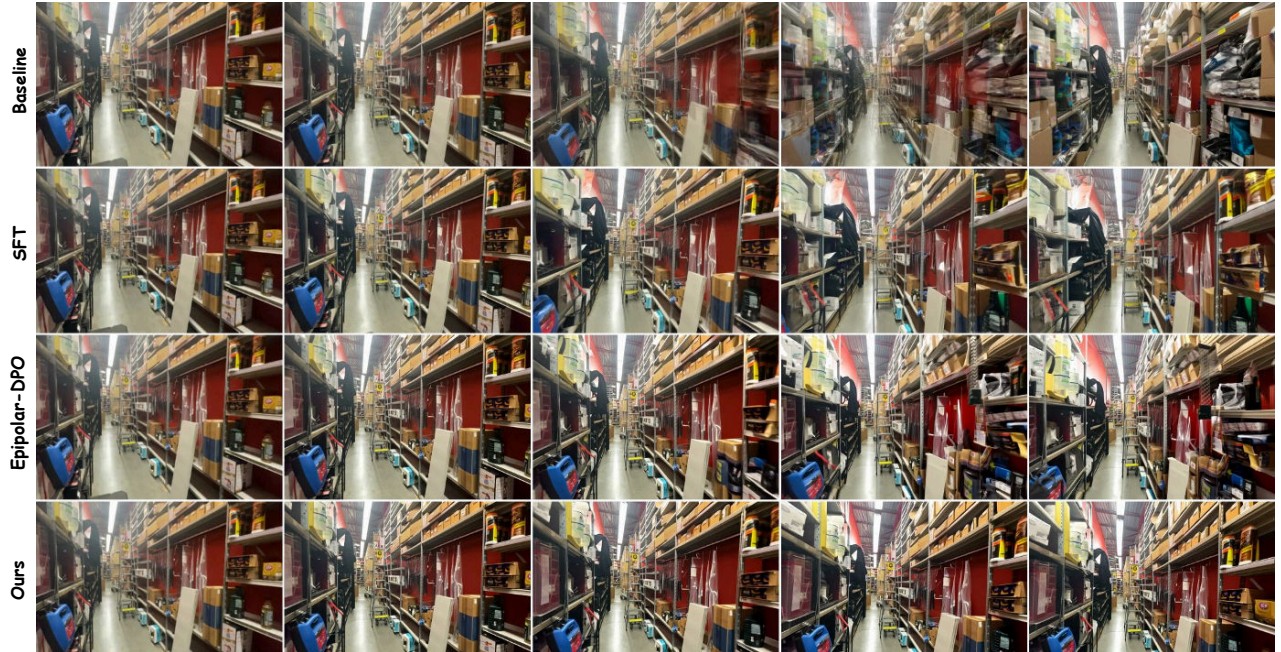

The video takes us through a well-lit warehouse filled with shelves stocked with various items. Initially, we see a red wall with metal shelves holding boxes, bags, and a ladder. As we move forward, the shelves display an assortment of goods including kitchen appliances, cleaning supplies, and automotive parts. The items are neatly organized, with some boxes labeled for specific products. A hand truck is visible, suggesting recent or upcoming activity. The warehouse is brightly lit, with fluorescent lights enhancing the visibility of the goods, which range from new to used, and are methodically arranged for easy access and identification.

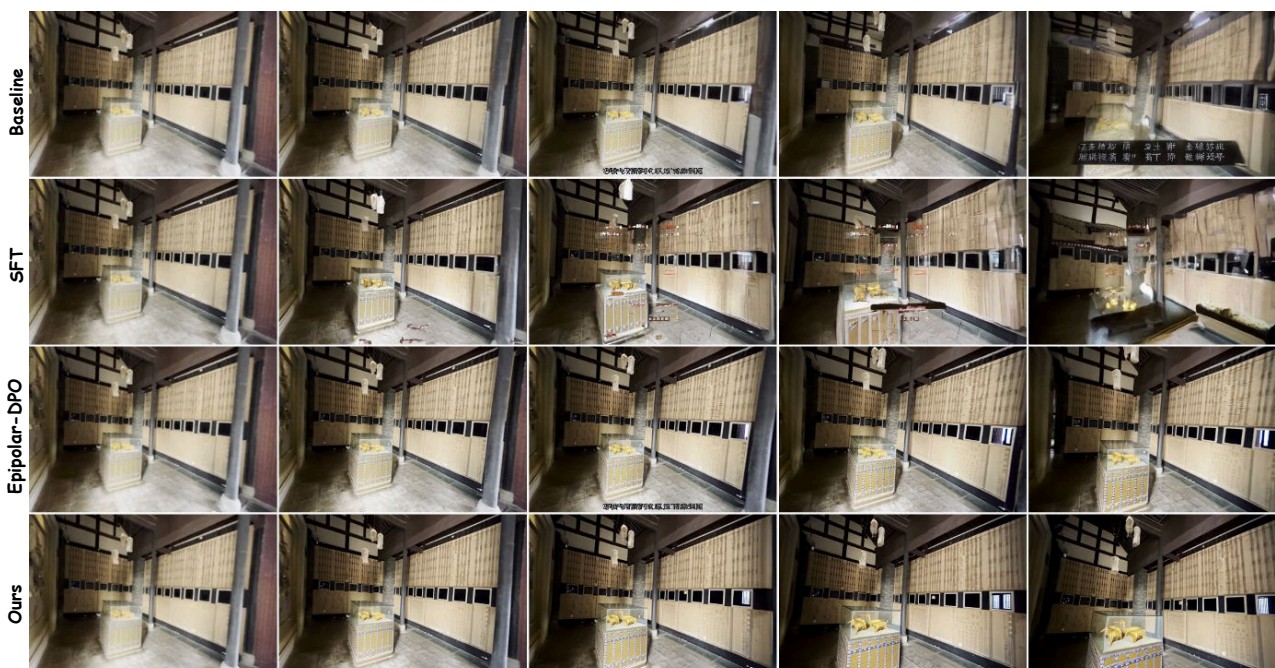

The video explores a traditional Chinese museum, starting with a display case of golden figurines and moving through various historical exhibits, including a lion statue, a model of an ancient Chinese city, and a human skeleton with tools. The museum features a high ceiling with wooden beams, stone columns, and bamboo wall panels, with soft lighting enhancing the cultural ambiance. Textual information in Chinese is visible, suggesting an educational purpose. The scenes transition from a serene, well-preserved interior to a more somber, reflective atmosphere with a display case of golden artifacts and a model of a historical battle scene, all under a warm glow.

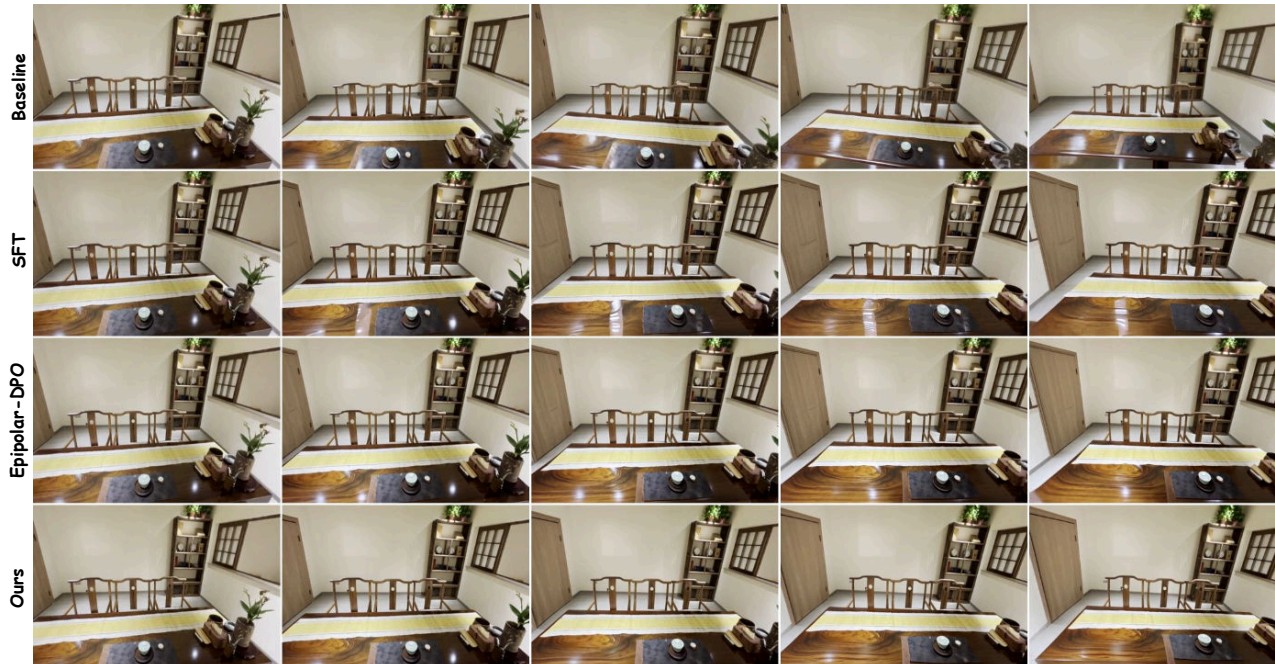

The video features a serene and culturally rich setting, showcasing a traditional Chinese tea ceremony room with a polished wooden table, elegant chairs, and a shelving unit with ceramic vases and books. As time passes, the room is shown with various angles, highlighting the tea set, the bamboo runner, and the soft lighting that creates a tranquil atmosphere. Decorative elements like a potted plant, a small statue, and framed pictures add to the ambiance. The room's design, with its minimalist aesthetic and soft lighting, suggests a space for contemplation and cultural appreciation.

## I.2. Text-to-Video on CogVideoX-5B

We show additional text-to-video results in this subsection. VideoGPA better preserves object structure and appearance over time, reducing deformation and semantic drift across diverse, visually complex scenes.

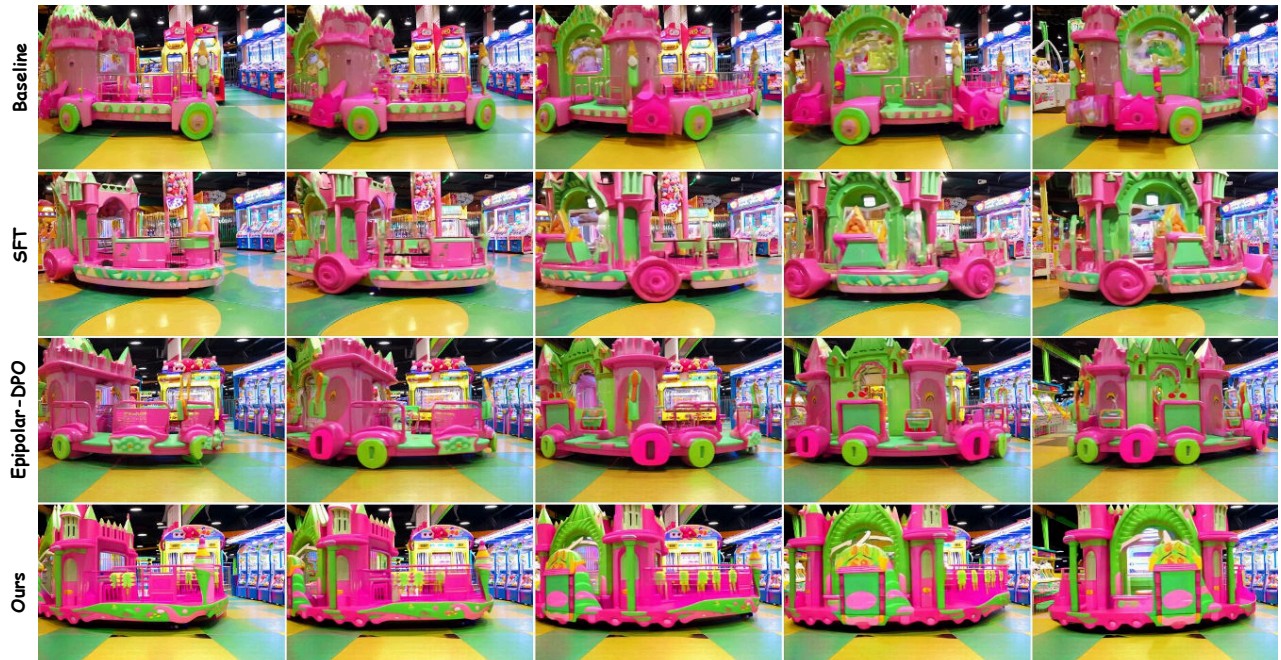

The video features an indoor playground with a variety of amusement rides, including a castle-themed ride with a pink and green color scheme, and a train ride with a similar design. The playground is brightly lit and decorated with a checkered floor pattern. As the video progresses, the playground appears empty, with no people present, and the scene is still. The playground is filled with colorful arcade machines, plush toys, and a claw machine, all under bright lighting. The video concludes with a focus on a pink and green train-shaped ride with a castle-like structure, surrounded by claw machines and other amusement rides.

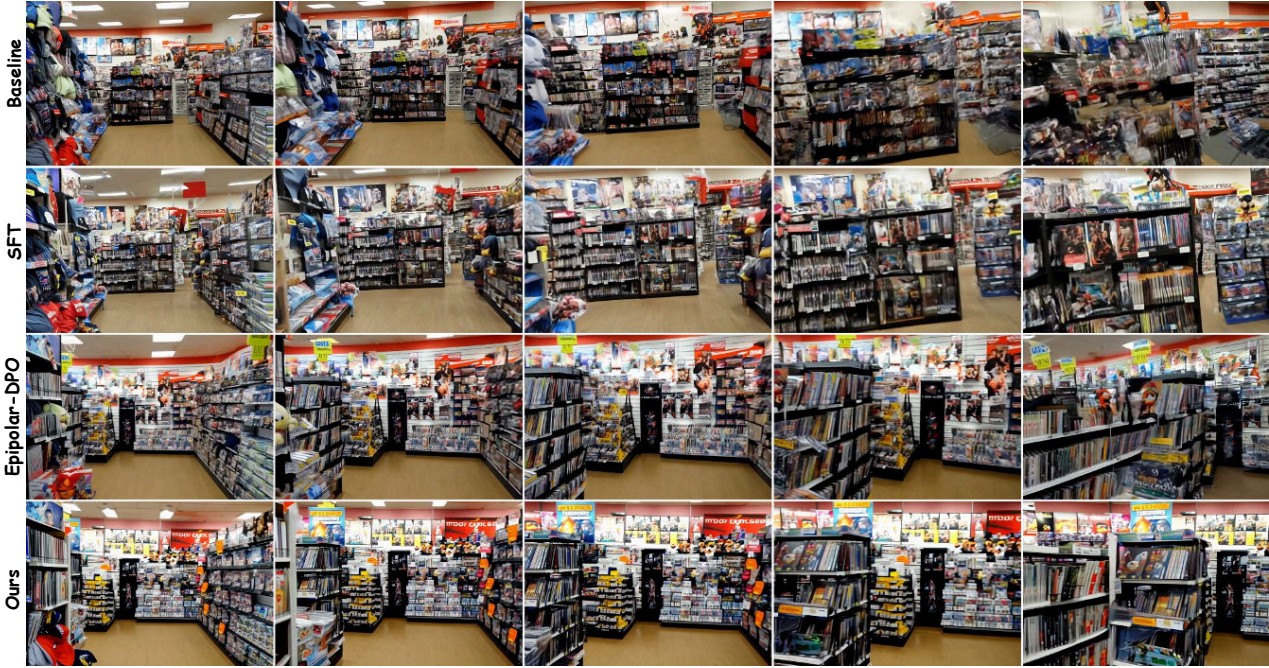

The video takes us through a vibrant video game store, starting with a view of the store's interior, showcasing DVDs, video games, and collectibles. As we move through the store, we see a variety of items including baseball caps, plush toys, and action figures, with clearance signs indicating sales. The store features a range of posters, including 'TRENDPOSTERS' and 'SUPERSTAR POSTERS', and a section dedicated to gaming accessories. The shelves are well-organized, displaying DVDs, Blu-rays, and video games, with a focus on the entertainment industry. The store's atmosphere is inviting, with a warm color palette and a sense of excitement for customers.

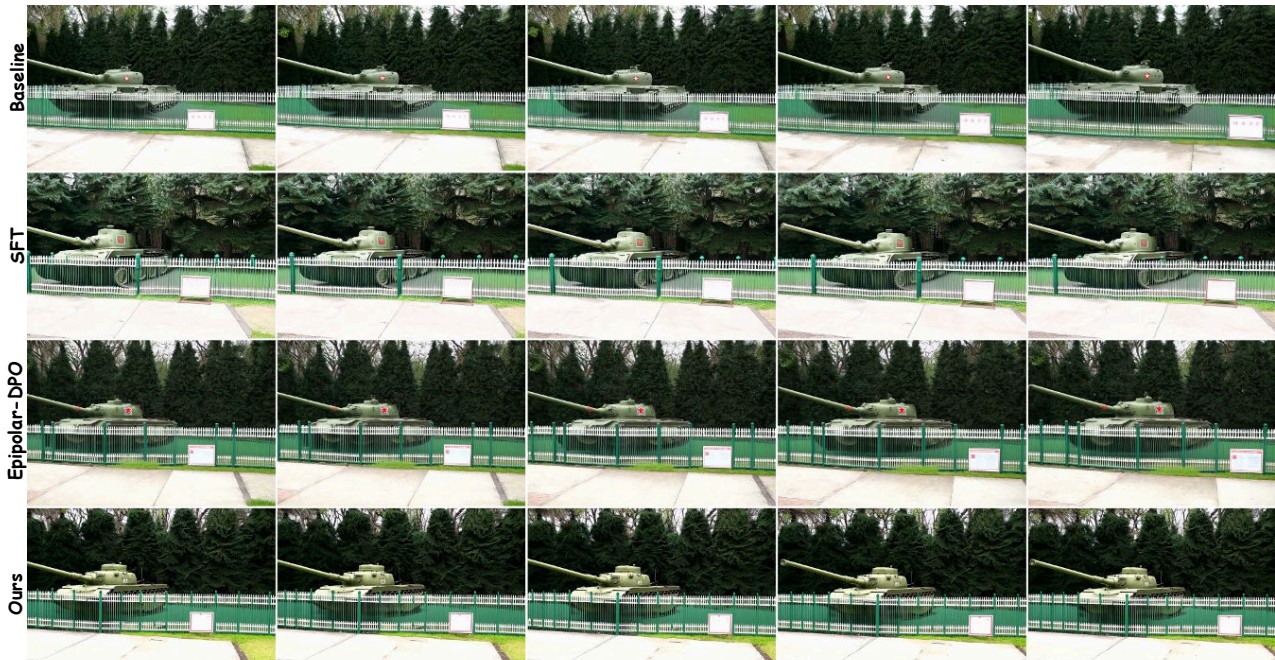

A green and white metal fence encloses a concrete platform with a historical military tank, featuring a long barrel and a red star emblem, suggesting its Chinese origin. The tank is surrounded by dense evergreen trees and a well-maintained lawn, with a sign providing historical context. Over time, the scene includes a green-painted metal fence with white picket tops, a concrete pathway, and a sign with Chinese characters, indicating the tank's location in a park or memorial area. The tank, possibly a M-48 or M-60, is displayed in a tranquil, reflective setting, with a sign indicating its historical significance.

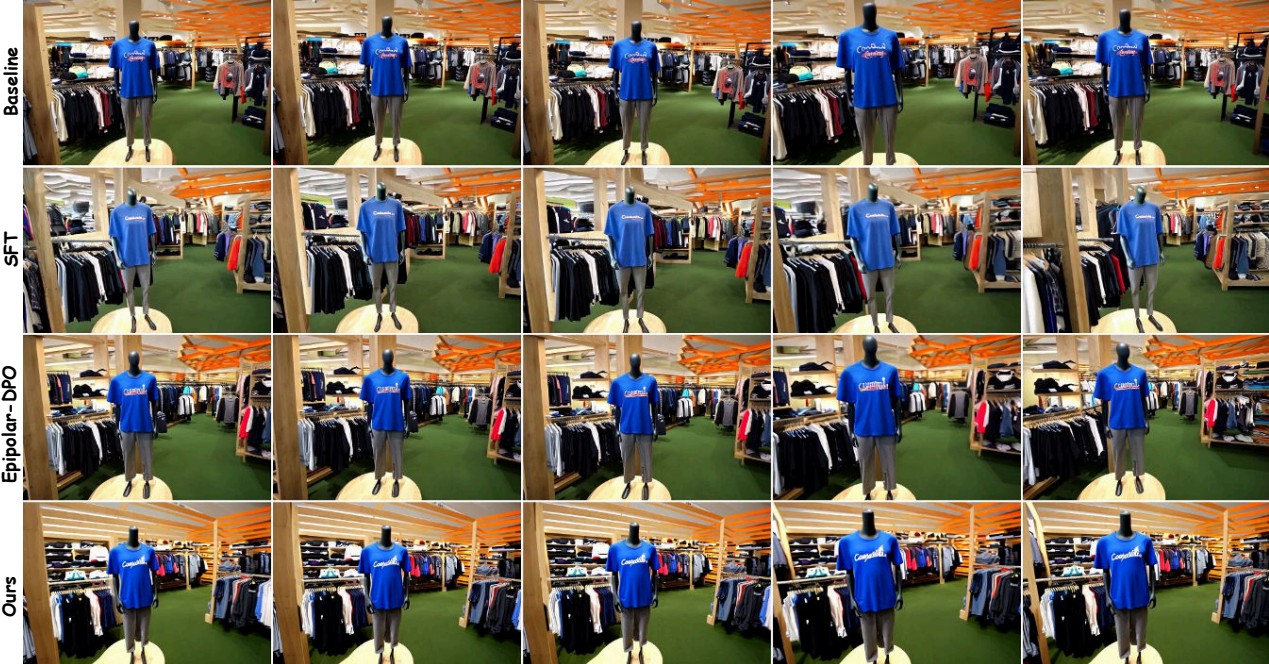

The video takes place in a clothing store, starting with a mannequin in a blue 'Columbia' t-shirt and grey pants on a wooden platform, surrounded by sportswear. As time passes, the mannequin changes outfits, including a 'Columbia' logo t-shirt and a 'Cabin' branded t-shirt, while the store's interior showcases various sportswear, casual clothes, and accessories. The store features a green carpet, wooden shelves, and a modern, minimalist design with a geometric wooden structure and vibrant orange stripes. The atmosphere is casual and inviting, with a focus on sportswear and casual wear, and a clear emphasis on the store's contemporary style.

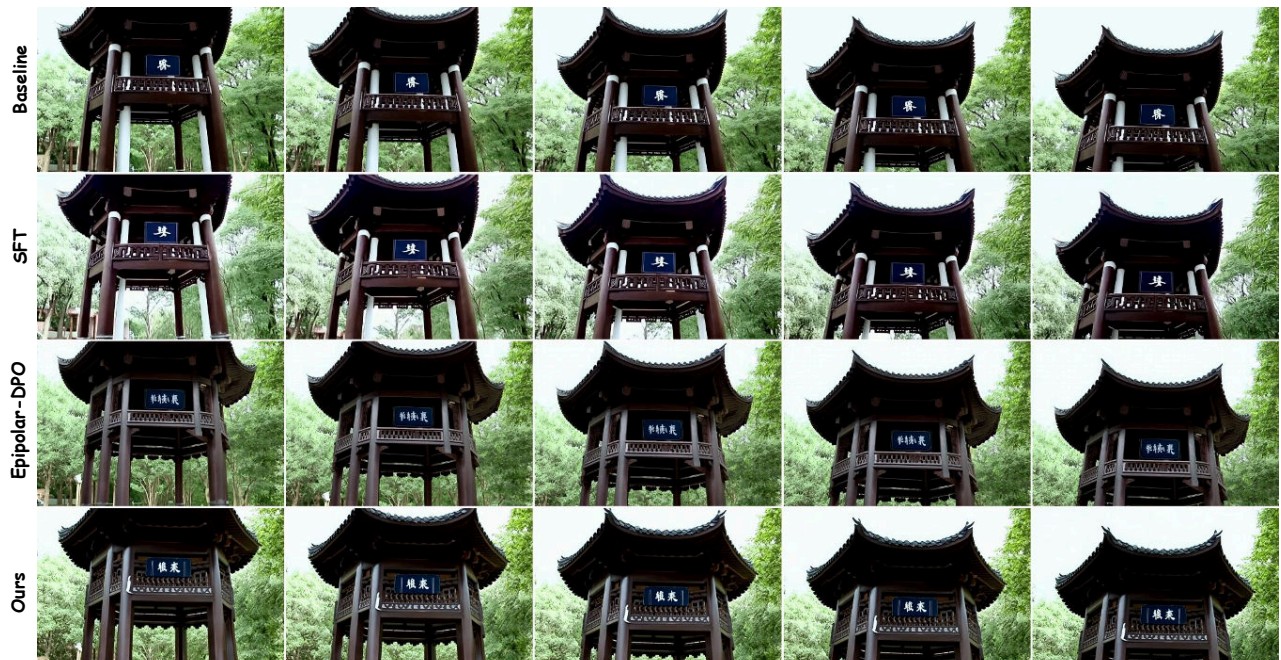

The video features a traditional Chinese-style gazebo with a multi-tiered, pagoda-like structure, set in a park with lush greenery. The gazebo, with its dark brown wooden beams, white columns, and grey tiled roof, is adorned with a blue sign and a black sign with Chinese characters. As the video progresses, the gazebo is shown from various angles, highlighting its architectural details and the surrounding greenery. The structure's design, including a curved roofline and ornate railings, suggests a tranquil and culturally rich environment, possibly linked to a media platform or event.

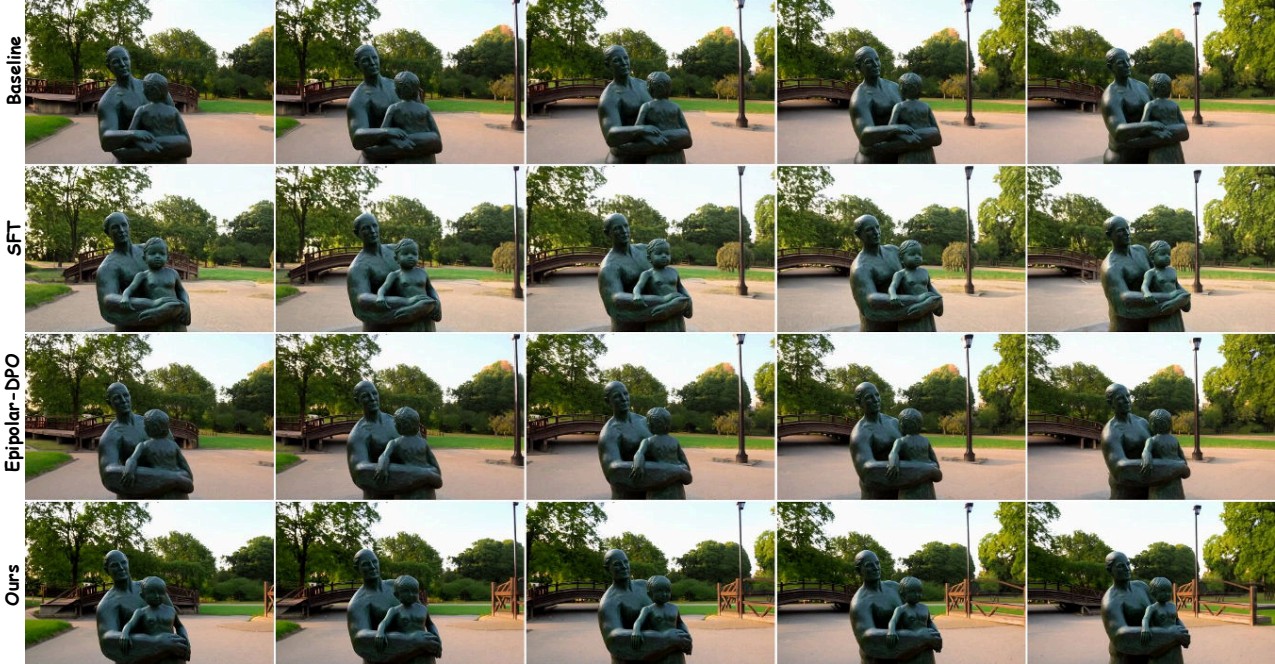

The video features a series of bronze statues in a park, each depicting an adult and a child in various intimate moments, suggesting a theme of guardianship and learning. The statues are set against a backdrop of a wooden bridge, lush greenery, and a clear sky, with a pathway leading to the bridge. The scenes are tranquil, with no people present, and the soft lighting indicates it might be early morning or late afternoon. The presence of a logo in some frames hints at an association with a media outlet or organization.

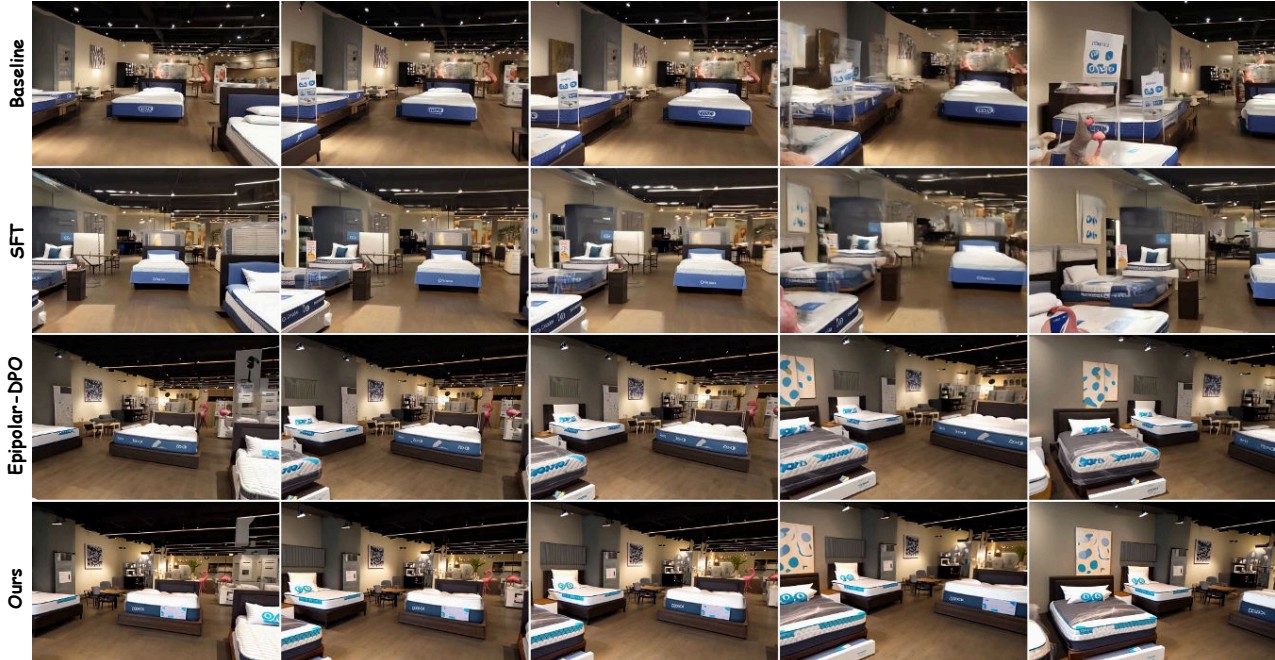

The video takes place in a modern furniture store, showcasing a variety of beds with different headboards and mattresses, some wrapped in plastic. The store features a dining set, a kitchenette, and a living area, all under warm lighting. Decorative elements like abstract art, a pink flamingo sculpture, and a plush ostrich toy add whimsy. The store displays beds with blue and white branding, including 'OYO' and 'QUBO', and offers promotional materials. The ambiance is contemporary, with a color scheme of neutral tones and warm lighting, creating an inviting atmosphere for customers.

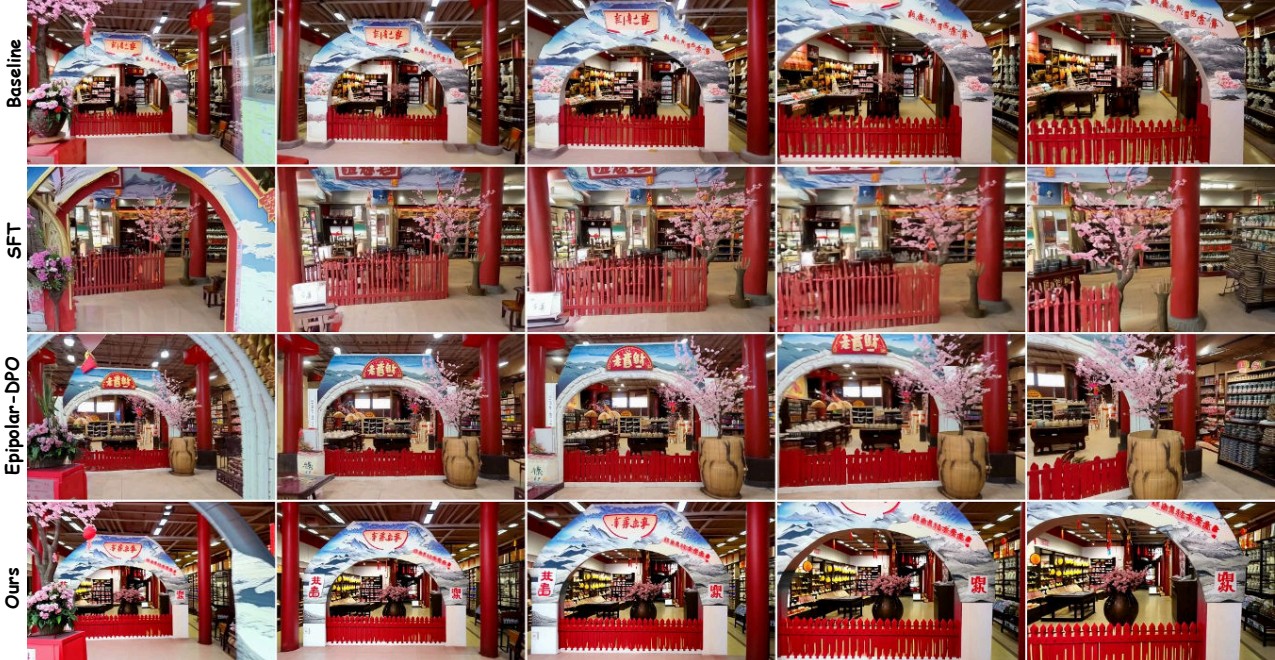

The video takes us through a traditional Chinese cultural store, starting with a view of a cherry blossom tree, a guqin, and a moon gate, all under red and gold decorations. As we move through the store, we see a traditional Chinese archway, a bamboo vase with flowers, and shelves stocked with tea sets and umbrellas. The ambiance is warm, with red columns, wooden furniture, and a mountainous landscape on the archway. A red and white sign with Chinese characters is visible, and the store features a cherry blossom tree, a red and white picket fence, and a display of tea sets and umbrellas, creating a festive atmosphere.

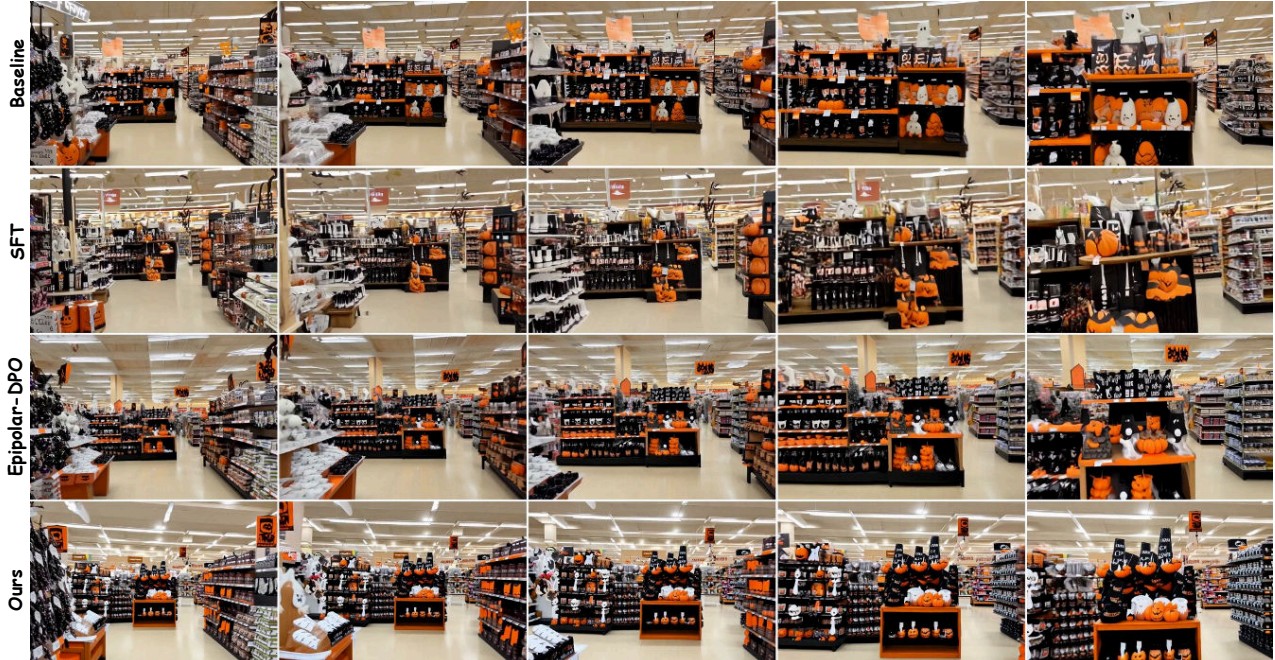

The video takes us through a department store's Halloween section, showcasing a variety of festive decorations. Initially, the store is empty, but as we move through the aisles, we see Halloween-themed items like candles, ornaments, and kitchenware. The decorations are arranged on wooden tables and shelves, with a color scheme of black, orange, and white. Some items feature playful phrases like 'BOO', 'BOOZE', and 'HOME SWEET HALLOWEEN'. The store is well-lit, with a warm ambiance, and the decorations include ghost figures, pumpkins, and bats, creating a cozy, celebratory atmosphere.

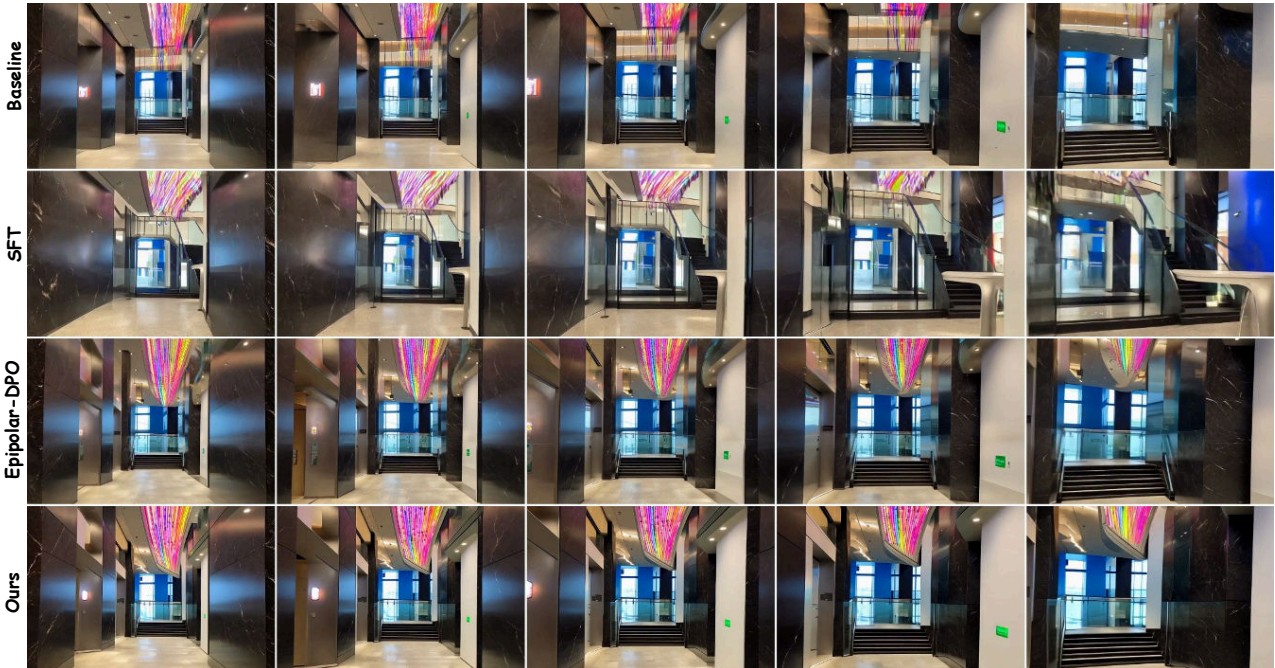

The video explores a modern building's interior, starting with a view of a staircase with dark marble walls and a colorful light installation. As the camera moves, it reveals various angles of the staircase, the surrounding walls, and the building's contemporary design, including a glass balustrade and a 'Science and Technology' sign. The focus shifts to a hallway with a 'Transforming Lives Through Chemistry' display, a white table, and a blue wall with a green sign. The video concludes with a view of a grand staircase leading to an upper level, with a 'EXIT' sign and a 'Parking Garage' sign visible

