# OpenReview forum: "VideoGPA: Distilling Geometry Priors for 3D-Consistent Video Generation"
_ICML.cc/2026/Conference — ICML 2026 regular_

### Official Review · Reviewer_jd1T · 2026-03-09

**Soundness:** 3
**Presentation:** 3
**Significance:** 3
**Originality:** 3
**Overall Recommendation:** 4
**Confidence:** 4

**Summary:**

This paper introduces VideoGPA (Video Geometric Preference Alignment), a data-efficient framework designed to address the 3D structural inconsistency and spatial drift found in video diffusion models. To achieve this, the authors: 1) utilize a geometry foundation model (VGGT) to reconstruct 3D structures and camera poses from generated videos; 2) design a 3D consistency score based on the reconstructed 3D geometry to rank video quality and construct preference pairs for Direct Preference Optimization (DPO); 3) use DPO to fine-tune the video diffusion model based on the constructed preference pairs.

**Compliance With Llm Reviewing Policy:**

Affirmed.

**Final Justification:**

The authors have resolved my concerns. I raise my score to 4 (Weak Accept).

**Key Questions For Authors:**

See "Weaknesses".

**Limitations:**

yes

**Strengths And Weaknesses:**

Strengths:
1. Clarity of Presentation: The paper is well-written and the proposed pipeline is described in a clear and organized manner.
2. Comprehensive Experiments: The authors provide a detailed quantitative evaluation across various metrics, including 3D reconstruction error, geometric consistency, and human-aligned scores, covering both I2V and T2V tasks. The experimental section also includes a variety of visual comparisons and frame-by-frame analyses that help illustrate the differences between the proposed method and the selected baselines.

Weaknesses:
1. Motivation and Scope of 3D Inconsistency: I question whether 3D geometric inconsistency is a real problem for video diffusion models. According to my empirical observations of models such as Wan, Veo, or Kling, they already maintain high geometric stability without explicit geometric constraints. Therefore, it is unclear whether the artifacts shown in Figure 1 reflect a general failure of current VDM technology or are merely limitations specific to the CogVideoX base model.
2. Reliability of VGGT Model: The use of VGGT as the foundation for preference modeling raises concerns regarding its robustness. Specifically: 1) While VGGT is proficient at 3D reconstruction for static environments, it suffers from poor 4D reconstruction capabilities required for dynamic scenes, which constitute a significant portion of video generation. 2) Even in static settings, the VGGT is susceptible to errors. The manuscript does not sufficiently address how to prevent the VDM from aligning with inaccurate geometric priors introduced by the VGGT's own reconstruction failures.
3. Marginal Quantitative Improvements: When compared to existing baselines, the performance gains of VideoGPA in terms of 3D Reconstruction Error and 3D Consistency metrics appear relatively marginal.

---

> ### Author Rebuttal · Authors · 2026-03-27
>
> We sincerely appreciate your time and effort in reviewing our paper and providing valuable comments. We provide explanations to your questions point-by-point in the following.
>
> ### **${\color{#557A46}\text{R1 to W1: Is 3D inconsistency a real problem?}}$**
>
> We appreciate this question. Since Veo and Kling are not open-sourced, we cannot evaluate or apply post-training alignment to them. To address this concern, we applied VideoGPA to **Wan2.2-TI2V-5B**, one of the strongest available open-source video generation models, using the same training setup with no modification. VideoGPA yields clear improvements across all geometric metrics and achieves **57**% OVL on VideoReward. Since Wan has a fundamentally different architecture from CogVideoX, this confirms that 3D inconsistency is not model-specific and that geometric alignment provides complementary gains on top of scaling. Please see $\color{#7E5CAD}\text{9zXy R4}$ for the full results table.
>
> ### **${\color{#557A46}\text{R2 to W2: Reliability of VGGT model}}$**
> We thank the reviewer for raising this important point.
> i) Regarding the 4D concern, we *intentionally* use static-scene prompts (Sec. 3.4) because VGGT excels at static 3D reconstruction. This is a deliberate design choice, and we are encouraged to find that geometric priors learned from static scenes generalize to dynamic scenes (Sec. 5.2, App. E).
> ii) Regarding error susceptibility and how we prevent the model from aligning with inaccurate priors: we use *relative ranking* between candidates for the same prompt rather than absolute scores, so systematic errors in VGGT affect all candidates similarly and cancel out in pairwise comparisons. Multi-stage filtering (App. A) further removes pairs where the geometric signal is unreliable.
> iii) To quantify this robustness, we conducted a label corruption analysis where preference labels are randomly flipped, showing graceful degradation even at 20% noise (see $\color{#536493}\text{7R1i R1}$). We also found that Depth Anything V3 achieves **100% agreement** with VGGT’s preference ranking across 3,000+ pairs, confirming that the supervision signal is reliable and not model-specific (see $\color{#536493}\text{7R1i R1}$ for details).
> iv) As geometric foundation models for dynamic scenes emerge, our pipeline can naturally extend by substituting the backbone.
>
>
> ### **${\color{#557A46}\text{R3 to W3: Marginal Quantitative Improvements}}$**
>
> We respectfully offer additional context:
>   - **Human study**: 53.5% win rate vs 22.4% for the next best method, a **2.4×** gap that participants clearly perceive.
>   - **VideoReward** (independent human-preference model): VideoGPA achieves the highest OVL win rates (**76.0**% I2V, **60.3**% T2V).
>   - **PSNR/SSIM**: Show smaller gains, which is consistent with the known insensitivity of pixel-level metrics to structural improvements. Our cross-model evaluation with DA3 shows the same pattern (see $\color{#7E5CAD}\text{9zXy R1}$), confirming this reflects metric behavior rather than weak improvements.
>   - **Data efficiency**: Achieving these gains with ~2,500 auto-generated pairs and ~1% trainable parameters makes the cost-to-benefit ratio highly favorable.

---

> > ### Author Rebuttal · Reviewer_jd1T · 2026-04-03
> >
> > My concerns regarding W1 and W3 are resolved. For W2, Could the authors provide a comparison of the consistency scores for "win" vs. "lose" cases generated under the same text/image conditions? This would effectively validate that:
> >
> > 1) 3D inconsistency is indeed a real problem in current base models;
> >
> > 2) Base models can generate both "win" and "lose" cases suitable for DPO training;
> >
> > 3) The proposed score is sensitive enough to distinguish between these cases.
> > ----
> > (update)
> >
> > Thanks for the authors' additional results. My concern regarding W2 is resolved. I will raise my score to 4 (Weak Accept).

---

> > > ### Author Response · Authors · 2026-04-05
> > >
> > > We thank the reviewer for acknowledging that W1 and W3 are resolved, and for the constructive follow-up on W2.
> > > We provide visual comparisons of "win" vs "lose" cases generated under the same conditioning input at https://anonymous43qx2.github.io/videogpa_rebuttal/, covering both I2V and T2V settings. The quality difference between winners and losers is visually apparent, confirming that:
> > >
> > > (1) the base model generates videos with varying levels of 3D consistency under identical conditions;
> > >
> > > (2) the variation is sufficient to construct meaningful winner/loser pairs for DPO; and
> > >
> > > (3) our scoring pipeline captures these differences reliably.
> > >
> > > We hope this resolves the remaining concern. We would greatly appreciate a score update if so, and are happy to address any further questions.

---

### Official Review · Reviewer_7R1i · 2026-03-11

**Soundness:** 4
**Presentation:** 4
**Significance:** 4
**Originality:** 3
**Overall Recommendation:** 5
**Confidence:** 4

**Summary:**

The paper observes that Video Diffusion Models (VDMs) produce geometrically inconsistent outputs despite massive training data, and attributes this to the denoising objective lacking geometric regularization. It proposes VideoGPA: given a generated video, reconstruct 3D via a geometry foundation model (VGGT), measure reprojection error as a 3D consistency score, rank samples into preference pairs, and align via DPO. With ~2,500 preference pairs and LoRA on ~1% of parameters, VideoGPA improves geometric consistency across I2V and T2V settings on CogVideoX, outperforming SFT, Epipolar-DPO, and GeoVideo.

**Compliance With Llm Reviewing Policy:**

Affirmed.

**Final Justification:**

Authors provided substantial additional experiments. I keep my accept rating.

**Key Questions For Authors:**

Questions for Authors

1. VGGT robustness. How sensitive is VideoGPA to the quality of the geometric foundation model? Have you tested with a weaker model, or analyzed cases where VGGT produces incorrect geometry?

2. Dataset generalization. Can you provide quantitative results on conditioning inputs from a source other than DL3DV to show that the improvements hold across visual distributions?

3. Motion manifold hypothesis. Have you considered any way to measure the regularization effect more directly? Not a requirement, but it could strengthen an already interesting finding.

4. SFT + DPO. Have you tried combining them? The paper positions these as alternatives, but they might be complementary.

**Limitations:**

yes

**Strengths And Weaknesses:**

What Works:

- The idea is simple, and the components are all standard (VGGT, DPO, LoRA), but the way they're composed is effective. The pipeline is easy to describe in one sentence, which I think speaks well for the method.
- From personal experience, VDMs do struggle with geometric consistency, so the motivation resonates. The results back it up: improvements are consistent across I2V and T2V, two base model versions, and all metric categories. The comparison against GeoVideo is interesting: explicit geometric supervision on 10K videos actually yields worse perceptual quality (VideoReward OVL: 18% vs 58%), suggesting that preference alignment might be a better strategy than direct supervision for this kind of problem.
- The data efficiency (~2,500 pairs, ~1% of parameters, competitive at 1,000 steps per Table 6) is somewhat expected given what we know about LoRA and DPO, but still impressive in practice. The human preference study (53.5% win rate vs. 22.4% for the next-best) confirms that the gains are real for humans, not just for metrics.
- What elevates this paper for me is the analysis. Section 5.1 makes a concrete case for scene-level over local constraints; the false-positive example in Fig. 5, where epipolar metrics pass a corrupted video but the 3D consistency score catches it, is compelling. Section 5.2 is where it gets genuinely interesting: training on static scenes with scripted camera motion somehow improves coherence in dynamic scenes. The authors had a hypothesis (background-foreground disentanglement), falsified it, and proposed an alternative (motion manifold regularization). I appreciate seeing that kind of intellectual process in a paper.
- The math seems to be correct.

Core Concerns:

- VGGT as an oracle assumption. The pipeline assumes VGGT is a reliable geometric supervisor. But what if it has systematic failure modes for certain scene types (transparent objects, reflections, highly dynamic scenes)? Those biases would propagate into the preference signal, and the model would learn to satisfy VGGT rather than achieve true 3D consistency. The filtering strategy provides some tolerance to noise, but no robustness analysis is provided. I think this connects to a broader open question in ML: when you use a foundation model as a reward signal, you inherit its biases. Even a simple analysis, corrupting some percentage of preference labels and measuring degradation, would help.
- Single-dataset quantitative evaluation. All numbers come from DL3DV-10K. The qualitative results are diverse, and the emergent dynamic consistency results (Appendix E) are convincing, but a quantitative evaluation on a second dataset would strengthen generalization claims.

Some more comments:

- The motion manifold hypothesis in Section 5.2 is one of the more interesting parts of the paper. The argument is informal, but the fact that the authors proposed it, tested their initial intuition (disentanglement), falsified it, and offered an alternative is already above the bar. Formalizing this would be a natural next step, perhaps for future work.
- The paper attributes inconsistency primarily to the objective function. I think it's probably a mix of objective, architecture, and data, some inductive bias in the architecture could also help. But addressing all of that is beyond the scope of this paper.
- There is a broader pattern emerging in the ML literature: use a foundation model as a reward signal to align another model (RLHF, RLAIF, and now geometric alignment). I think this approach has promise, but the jury is still out on its ceiling. When the teacher model improves, does the student automatically follow? The paper could position itself more explicitly in this trend.
- Is there an emerging consensus on SFT vs RL for alignment? This paper provides evidence that preference optimization beats direct supervision for geometric consistency and adds to this discussion in a meaningful way.

---

> ### Author Rebuttal · Authors · 2026-03-27
>
> We thank Reviewer 7R1i for the thorough review and for recognizing soundness, presentation, and significance as "excellent." We especially appreciate the engagement with the motion manifold hypothesis and the foundation-model-as-reward paradigm.
>
> ### **${\color{#536493}\text{R1 to W1+Q1: VGGT robustness}}$**
> We agree foundation model signals inherit biases. Our pipeline ensures **robustness** via:
> (i) Relative comparisons that cancel systematic bias.
> (ii) Multi-stage filtering that prunes low-margin/static pairs (App. A).
> (iii) Independent evaluation via VideoReward and human study, both decoupled from VGGT.
>
> Following the reviewer’s suggestion, we trained with 0–20% flipped preference labels for 1k steps. All results use DA3 as an independent geometric backbone:
>
> | Config | PSNR ↑ | SSIM ↑ | LPIPS ↓ | MVCS ↑ | 3DCS ↓ | Epipolar ↓ | OVL ↑ |
> | :--- | :---: | :---: | :---: | :---: | :---: | :---: | :---: |
> | CogVideoX-I2V-5B | 22.854 | 0.786 | 0.476 | 0.945 | 0.485 | 0.585 | - |
> | 0% flipped | 23.045 | 0.831 | 0.438 | **0.954** | 0.445 | **0.509** | **66.0%** |
> | 10% flipped | **23.380** | **0.834** | **0.436** | 0.952 | **0.443** | 0.516 | **66.0%** |
> | 20% flipped | 23.253 | 0.832 | 0.439 | 0.951 | 0.446 | 0.523 | 63.0% |
>
> Degradation is gradual and consistent (OVL: 66%→66%→63%), confirming the pipeline’s tolerance to noisy geometric supervision.
>
> We also measured whether alternative backbones agree with VGGT's preference pairs across 3K+ pairs:
> | Backbone | Agreement Rate | Top-1 Ranking | Full Ranking |
> | :--- | :---: | :---: | :---: |
> | DUSt3R | 80.8% | 62.5% | 48.5% |
> | DA3 | **100.0%** | **100.0%** | **100.0%**|
>
> DA3 achieves full consensus with VGGT, suggesting that as geometric supervisors improve, preference signals converge. (Agreement Rate: winner/loser agreement; Top-1: best-of-3 agreement; Full Ranking: complete ordering agreement.)
>
> ### **${\color{#536493}\text{R2 to W2+Q2: Dataset generalization}}$**
> We evaluated on 100 WebVid videos, **disjoint** from DL3DV-10K:
> | Config | PSNR ↑ | SSIM ↑ | LPIPS ↓ | MVCS ↑ | 3DCS ↓ | Epipolar ↓ |
> | :--- | :---: | :---: | :---: | :---: | :---: | :---: |
> | Baseline-I2V | 17.434 | 0.598 | 0.542 | 0.966 | 0.582 | 1.282 |
> | SFT | **20.453** | 0.689 | 0.470 | **0.975** | 0.486 | 0.748 |
> | Epi-DPO | 19.445 | **0.710** | 0.443 | 0.972 | 0.462 | 0.699 |
> | VideoGPA | 18.767 | 0.693 | **0.435** | 0.972 | **0.460** | **0.602** |
>
> VideoGPA achieves the best LPIPS, 3DCS, and Epipolar scores, confirming geometric consistency improvements generalize across datasets.
>
> ### **${\color{#536493}\text{R3 to Comment 1+Q3: Motion manifold hypothesis}}$**
> On 100 Panda-70M dynamic videos, VideoGPA outperforms the baseline across all VideoReward metrics (OVL 64.0%). Since these dynamic scenes are absent from our static-scene training data, this OOD improvement supports the manifold regularization hypothesis: the model learns generalizable geometric priors rather than memorizing scene-specific patterns. Formalizing this through latent distribution analysis is a promising future direction.
>
> | Metric | VQ | MQ | TA | OVL |
> | :--- | :---: | :---: | :---: | :---: |
> | Baseline-I2V | -- | – | – | – |
> | VideoGPA | **61.0%** | **61.0%** | **51.0%**| **64.0%**|
>
> ### **${\color{#536493}\text{R4 to Comment 2: Outcome attributions}}$**
> We agree that 3D consistency is likely shaped by objective, architecture, and data jointly. Our work addresses the objective function, but architecture and data could provide complementary gains. We will soften the framing in the revision.
>
> ### **${\color{#536493}\text{R5 to Comment 3: Foundation model as reward signal}}$**
> Our cross-backbone analysis (R1) provides initial evidence that student quality tracks teacher quality: DA3 (stronger) achieves 100% agreement with VGGT's preferences while DUSt3R (weaker) agrees at 80.8%. This suggests that as geometry models improve, our pipeline directly benefits by regenerating preference pairs with stronger supervisors, with no change to the framework. We will discuss this trend in Sec. 6.
>
> ### **${\color{#536493}\text{R6 to Comment 4+Q4: SFT vs RL}}$**
> Our controlled SFT-1K (Winner) baseline (see $\color{#E16A54}\text{n36s R1}$) confirms the DPO objective drives gains beyond direct supervision.
>
> Following the reviewer’s suggestion, we started from the SFT checkpoint (10K steps) and continued with 1K DPO:
>
> | Config | PSNR ↑ | SSIM ↑ | LPIPS ↓ | MVCS ↑ | 3DCS ↓ | Epipolar ↓ | OVL ↑ |
> | :--- | :---: | :---: | :---: | :---: | :---: | :---: | :---: |
> | CogVideoX-I2V-5B | 22.854 | 0.786 | 0.476 | 0.945 | 0.485 | 0.585 | - |
> | SFT(10K)→DPO(1K) | **23.115** | 0.811 | 0.458 | 0.951 | 0.464 | 0.561 | 63.0% |
> | DPO only (1K) | 23.045 | **0.831** | **0.438** | **0.954** | **0.445** | **0.509** | **66.0%** |
>
>
> DPO-only outperforms SFT→DPO on most metrics and overall preference (**66**% vs 63%). The SFT stage appears to interfere with subsequent DPO optimization. We leave deeper investigation for future work.

---

> > ### Author Rebuttal · Reviewer_7R1i · 2026-04-04
> >
> > Thank you for a thorough rebuttal that directly addresses both concerns. The flipped label experiment and cross-backbone agreement analysis provide exactly the robustness evidence I was looking for, and the WebVid and Panda-70M results confirm that the improvements generalize beyond DL3DV. The finding that DPO-only outperforms SFT followed by DPO is interesting and somewhat surprising.
> > One request: please ensure that the flipped-label results, cross-backbone analysis, and WebVid generalization numbers appear in the camera-ready version, either in the main paper or in the appendix, rather than only in the rebuttal.

---

> > > ### Author Response · Authors · 2026-04-05
> > >
> > > We thank the reviewer for the constructive feedback that directly shaped several of our strongest new results, particularly the flipped-label analysis and cross-backbone agreement study, which originated from the reviewer's suggestions. As requested, we will include these along with the WebVid generalization and Panda-70M OOD evaluation in the camera-ready version. We would be grateful if the reviewer could consider whether the strengthened evaluation warrants a higher score. Thank you for the thoughtful engagement throughout the review process.

---

### Official Review · Reviewer_9zXy · 2026-03-12

**Soundness:** 2
**Presentation:** 3
**Significance:** 3
**Originality:** 3
**Overall Recommendation:** 4
**Confidence:** 4

**Summary:**

The authors propose **VideoGPA**, a self-supervised framework that leverages a geometry foundation model to automatically generate geometric preference signals and guide training via Direct Preference Optimization (DPO). The method improves temporal stability, physical plausibility, and motion coherence, outperforming state-of-the-art approaches without requiring human annotations.

**Compliance With Llm Reviewing Policy:**

Affirmed.

**Key Questions For Authors:**

**Q1. Evaluation Bias:** The evaluation of 3D reconstruction closely mirrors the training process and relies on highly correlated metrics. Could the authors clarify how they ensure the evaluation is truly independent of the training preferences? Are there alternative metrics or approaches that could more robustly assess 3D reconstruction quality?

**Q2. Scene Complexity Preservation:** While VideoReward suggests perceptual quality is maintained, it is unclear whether the model **simplifies complex scenes** to improve 3D consistency. Have the authors analyzed scene complexity, or can they provide evidence that it is preserved?

**Q3. Generalizability to Other Models:** Experiments are restricted to the CogVideoX series. Can VideoGPA be effectively applied to other video generation architectures? If not, how does it compare against other state-of-the-art video generation models?

**Limitations:**

Yes

**Strengths And Weaknesses:**

**Strengths**:

1. This work leverages the geometric priors provided by Geometry Foundation Models (GFMs), distilling reconstruction-based 3D knowledge into video diffusion models. This approach aligns video generation with physically consistent 3D geometry without the need for training from scratch or relying on human annotations.
2. With just 2,500 preference pairs and LoRA fine-tuning on 1% of parameters, VideoGPA boosts geometric coherence and temporal stability while keeping the base model’s visual quality and motion realism, demonstrating a lightweight and efficient approach.

**Weaknesses**:

1. The evaluation of *3D Reconstruction Error* closely mirrors the reconstruction process used to generate the training preferences. Since both rely on similar signals (LPIPS/MSE for training, PSNR/SSIM/LPIPS for evaluation), this raises a potential bias: the method is effectively optimized and assessed on highly correlated metrics, limiting the independence of the evaluation. Moreover, the modest improvements observed in PSNR/SSIM seem to further support this concern.

2. The evaluation primarily measures 3D consistency and uses VideoReward to verify perceptual quality, but it remains unclear whether the model simplifies complex scenes to achieve higher 3D consistency. The authors should assess scene complexity to confirm that this aspect of quality is not compromised.

3. There appears to be a typo in the manuscript. On Line 315, it states “*33.33% for Epipolar-DPO*”, which does not match the value in the table (48.67%).

4. The proposed improvements are primarily demonstrated on the CogVideoX series models. It is unclear whether VideoGPA can be effectively applied to other models with similar architectures. If not, the authors should provide comparisons with other state-of-the-art video generation models.

5. The manuscript frequently mentions “*physical plausibility*” and “*physical laws*”, but in practice, the work only enforces **3D geometric consistency** and does not address actual physical properties or dynamics.

---

> ### Author Rebuttal · Authors · 2026-03-27
>
> We sincerely appreciate your time and effort in reviewing our paper and providing valuable comments. We provide explanations to your questions point-by-point in the following.
>
> ### **${\color{#7E5CAD}\text{R1 to W1+Q1: 3D reconstruction metric concern}}$**
> We share this concern. The reconstruction metrics are indeed computed through the same VGGT pipeline used during training. However, our two strongest evaluation signals are *fully independent*: VideoReward (OVL 76.0% I2V, 60.3% T2V) and the blind human study (53.5% vs 22.4% for the next best).
>
> The reviewer observes that metrics closer to our training signal (LPIPS, 3DCS) show larger gains while less related metrics (PSNR, SSIM) show modest gains, and suggests this pattern supports the circularity concern. To test this directly, we reevaluated all 3D reconstruction-based metrics using Depth Anything V3, a geometry model with completely different architecture and training data from VGGT. If the gains were artifacts of circular evaluation, they would *diminish* under an independent backbone. Instead, VideoGPA achieves consistent improvements across LPIPS, MVCS, 3DCS, and Epipolar under DA3. The same modest PSNR/SSIM pattern appears under DA3, confirming this reflects a known limitation of pixel-level metrics in capturing structural improvements, not evidence of circularity.
>
> #### **I2V Configuration**
> | Config | PSNR ↑ | SSIM ↑ | LPIPS ↓ | MVCS ↑ | 3DCS ↓ | Epipolar ↓ |
> | :--- | :---: | :---: | :---: | :---: | :---: | :---: |
> | Baseline-I2V | **22.85** | **0.786** | 0.476 | 0.945 | 0.579| 0.706 |
> | SFT | 21.58 | 0.749 | 0.513 | 0.947 | 0.651 | 0.628 |
> | Epipolar-DPO | 21.38 | 0.773 | 0.475 | 0.944 | 0.558 | 0.571 |
> | **VideoGPA** | 21.24 | 0.779 | **0.473** | **0.950** | **0.551** | **0.564** |
>
> #### **T2V Configuration**
> | Config | PSNR ↑ | SSIM ↑ | LPIPS ↓ | MVCS ↑ | 3DCS ↓ | Epipolar ↓ |
> | :--- | :---: | :---: | :---: | :---: | :---: | :---: |
> | Baseline-T2V | 21.47 | 0.784 | 0.435 | 0.944 | 0.445 | 0.584 |
> | SFT | 19.99 | 0.721 | 0.496 | 0.937 | 0.510 | 0.719 |
> | Epipolar-DPO | **21.58** | 0.791 | 0.434 | **0.953** | 0.443 | 0.579 |
> | **VideoGPA** | 21.24 | **0.803** | **0.411** | **0.953** | **0.422** | **0.548** |
>
> ### **${\color{#7E5CAD}\text{R2 to W2+Q2: Scene complexity concern}}$**
>
> Visually, we observe no degradation in scene complexity. During training, DPO incorporates KL regularization to prevent the model from deviating excessively from the original policy. To provide quantitative evidence, we measured four spatio-temporal metrics: Laplacian variance (texture sharpness), FFT high-frequency ratio (fine-grained detail), edge density (structural complexity), and optical flow magnitude (motion intensity). As shown below, all four metrics are *preserved or improved* after VideoGPA fine-tuning in both I2V and T2V settings, confirming that geometric alignment does not come at the cost of scene complexity.
>
> | Model | LV ↑ | FHR ↑ | ED ↑ | OFM ↑ |
> | :--- | :---: | :---: | :---: | :---: |
> | Baseline-I2V | 299.560 ± 217.910 | 0.506 ± 0.054 | 0.117 ± 0.053 | 10.250 ± 4.770 |
> | **VideoGPA-I2V** | **839.260 ± 687.600** | **0.558 ± 0.058** | **0.136 ± 0.059** | **11.170 ± 4.750** |
> | Baseline-T2V | 711.130 ± 374.300 | 0.585 ± 0.058 | 0.113 ± 0.049 | 11.460 ± 5.420 |
> | **VideoGPA-T2V** | **917.120 ± 512.530** | **0.594 ± 0.056** | **0.115 ± 0.049** | **11.550 ± 5.610** |
>
> ### **${\color{#7E5CAD}\text{R3 to W3: Typo}}$**
>
> We thank the reviewer for catching this. The value on Line 315 is a typo and will be corrected in the revision.
>
> ### **${\color{#7E5CAD}\text{R4 to W4+Q3: Generalizability on other model family}}$**
> Our pipeline is *architecture-agnostic*. We applied VideoGPA to Wan2.2-TI2V-5B, which fundamentally has a different architecture from CogVideoX, using the same training setup with no modification. VideoGPA improves across all geometric metrics and achieves 57% OVL on VideoReward. We will include training details, evaluation, and visualizations for Wan in the revised paper.
>
> | Method | PSNR ↑ | SSIM ↑ | LPIPS ↓ | MVCS ↑ | 3DCS ↓ | Epipolar ↓ | OVL ↑ |
>   | :--- | :---: | :---: | :---: | :---: | :---: | :---: | :---: |
>   | Wan2.2-TI2V-5B | 19.406 | 0.723 | 0.480 | 0.898 | 0.506 | 0.594 | - |
>   | VideoGPA (Ours) | **24.411** | **0.842** | **0.394** | **0.944** | **0.451** | **0.499** | **57.0%** |
>
> ### **${\color{#7E5CAD}\text{R5: Concern of physical plausibility claim}}$**
> We agree with the reviewer. Our method enforces 3D geometric consistency, and we should not overclaim beyond that. We have revised the terminology to "geometric plausibility" and "geometric constraints." That said, we did observe emergent improvements in motion coherence beyond static geometry, which we explore in the Discussion section and Appendix E. We acknowledge that geometric consistency is a necessary but not sufficient condition for full physical realism (e.g., gravity, collisions), and will add this clarification to Sec. 6.

---

> > ### Author Rebuttal · Reviewer_9zXy · 2026-04-04
> >
> > Thank the authors for the detailed rebuttal. My major concerns regarding the metrics, scene complexity, and generalization to other model backbones have been adequately addressed. I will maintain my original borderline acceptance rating.

---

> > > ### Author Response · Authors · 2026-04-05
> > >
> > > We thank the reviewer for confirming that the concerns are adequately addressed. We will incorporate all new experiments into the camera-ready version. We would greatly appreciate it if the reviewer could consider a score update given that the evaluation concerns, particularly regarding metric independence, have been fully resolved with the DA3 cross-model validation. Of course, we are happy to address any remaining questions.

---

### Official Review · Reviewer_n36s · 2026-03-12

**Soundness:** 3
**Presentation:** 3
**Significance:** 3
**Originality:** 3
**Overall Recommendation:** 4
**Confidence:** 4

**Summary:**

This paper tackles the issue in video diffusion models, the struggle to maintain 3D structural consistency. To address this, the authors propose VideoGPA, a data-efficient post-training pipeline. Instead of relying on human feedback or local, pairwise geometric constraints (like epipolar lines), the method uses the VGGT as an automated reward model. Specifically, it extracts depth and camera poses from generated video frames, unprojects them into a global 3D point cloud, and then measures the reprojection error back onto the 2D frames. This reprojection error serves as a global 3D consistency score. The authors use this score to automatically rank candidate videos, filter out poor-quality samples, and construct a compact dataset of about 2,500 preference pairs. Finally, they fine-tune base models (CogVideoX) using DPO via a lightweight LoRA.

The method achieves noticeable improvements in temporal stability and dynamic motion coherence with just ~1% trainable parameters and a few thousand auto-generated preference pairs, which is an attractive recipe for the community. However, there are several experimental details and baseline comparisons that need to be tightened up.

**Compliance With Llm Reviewing Policy:**

Affirmed.

**Final Justification:**

The paper presents a practical and data-efficient pipeline for improving 3D consistency in video diffusion models. The authors provided a good rebuttal that directly resolved my initial concerns by adding a fair SFT ablation, confirming the absence of motion suppression via optical flow metrics, and demonstrating that the method generalizes to a stronger base model (Wan2.2). Thus, I maintain my recommendation of a Weak Accept.

**Key Questions For Authors:**

- The SFT baseline in the paper is trained on ~20k general clips, while VideoGPA uses a highly curated set of ~2,500 pairs filtered for motion salience and geometric margin. To truly isolate the benefit of the DPO objective, can the authors provide an SFT baseline trained only on the ~2,500 winning videos from your curated dataset?
- I read the "emergent dynamic consistency" discussion, but I have a lingering worry about motion suppression. Can you report a metric like average optical flow magnitude or camera translation distance for the generated videos before and after VideoGPA?
- LPIPS is sensitive to holes and splatting noise, how noisy is this reward signal in practice?
- Given how fast this field moves, models like Wan have stronger native 3D priors out of the box than CogVideoX. Since your post-training pipeline is lightweight (~1% parameters, 2.5k pairs), is it possible to run a quick test on one of these stronger open-weight models during the rebuttal?

**Limitations:**

On the technical side, it would be great if they explicitly acknowledged that their current training pipeline heavily relies on static-scene prompts as a limitation, as this constraint might restrict the model from learning more complex, multi-object interactions during the alignment phase.

**Strengths And Weaknesses:**

## Strengths
- The pipeline is cheap to run. Achieving this level of visual improvement using only ~2,500 automatically generated preference pairs and a LoRA fine-tune makes the method highly accessible.

## Weaknesses
- The comparison against the SFT baseline is not an apples-to-apples evaluation. The SFT model is trained on ~20k general clips, while VideoGPA uses DPO on ~2.5k highly curated preference pairs (which were strictly filtered for motion salience, visual quality, and geometric margin). Because of this, it is hard to tell if the performance boost actually comes from the DPO objective, or if the authors simply built a highly effective data curation filter.
- The core 3D Consistency Score relies on unprojecting pixels into a 3D point cloud and rendering them back to 2D using a basic painter's algorithm. Reprojecting sparse point clouds inherently creates holes, splatting artifacts, and quantization errors. LPIPS is famously sensitive to these kinds of artifacts.
- A common loophole when training video models on geometric metrics is that the model learns to generate very conservative, slow-moving, or nearly static videos to minimize alignment errors. While the paper reports improved Motion Quality scores from a VLM judge, VLM judges may evaluate whether motion looks natural, not the quantity of the motion. The paper doesn't provide numerical evidence to prove that the fine-tuned model hasn't just slowed down its generation dynamics to "play it safe".
- In Table 1, the authors report their own 3DCS as a key evaluation metric. Since the model was explicitly optimized to minimize this exact score via DPO during training, using it to evaluate the model's success is circular.

---

> ### Author Rebuttal · Authors · 2026-03-27
>
> We sincerely appreciate your time and effort in reviewing our paper and providing valuable comments. We provide explanations to your questions point-by-point in the following.
> ### **${\color{#E16A54}\text{R1 to W1+Q1: SFT baseline comparison clarification}}$**
>
> The SFT baseline is trained on real-world videos from the DL3DV-10K with consistently high visual quality and physically plausible motion. Real videos are inherently 3D-consistent, so SFT has the advantage in both data quality and quantity. Yet VideoGPA still outperforms SFT indicating that *DPO objective itself*, not data curation, is the source of the gain.
>
> To further isolate the DPO objective, we trained CogVideoX-I2V-5B on the same 2.5K winning videos for SFT and DPO with1K steps, respectively, evaluated on 100 videos with DA3 as the geometric backbone to address circularity concerns.
>
> | Config | PSNR ↑ | SSIM ↑ | LPIPS ↓ | MVCS ↑ | 3DCS ↓ | Epipolar ↓ | OVL ↑ |
> | :--- | :---: | :---: | :---: | :---: | :---: | :---: | :---: |
> | Baseline-I2V | 22.854 | 0.786 | 0.476 | 0.945 | 0.485 | 0.585 | - |
> | SFT-1K (Winners) | 21.285 | 0.772 | 0.489 | **0.957** | 0.498 | 0.589 | 39.0% |
> | VideoGPA-1K (Ours) | **23.045** | **0.831** | **0.438** | 0.954 | **0.445** | **0.509** | **66.0%** |
>
> SFT-1K underperforms VideoGPA across most metrics, especially on VideoReward (39% vs 66%), confirming the DPO objective drives the gain.
>
> ### **${\color{#E16A54}\text{R2 to W2+Q3: 3D consistency score artifacts}}$**
>
> VGGT uses pointmap where each pixel corresponds to a point in the 3D point cloud. Since we preserve all points, every pixel maps to at least one reprojected point, thereby no holes in practice. Splatting artifacts typically occur when the input frames are too sparse. We use 10 frames per video, which keeps a good balance between reconstruction density and inference efficiency. Despite the small amount of rendering noise, during training, DPO only sees the *relative ranking* between candidates for the same prompt, not absolute scores. Any rendering noise shared across candidates from the same scene cancels out in pairwise comparisons and does not affect preference pair quality.
>
> ### **${\color{#E16A54}\text{R3 to W3+Q2: Motion suppression loophole}}$**
>
> We thank the reviewer for raising this concern and for suggesting optical flow magnitude (OFM) as a direct measurement. Visually, we do not observe any motion degeneration in our generated videos. We also note that static videos are filtered from both training and evaluation (Sec. 4.1.3).
>
> Following the reviewer’s suggestion, we computed OFM (Farneback, 10 sampled frames) across our full evaluation set.
>
> ### Average Optical Flow Magnitude
> | Model | Mean ↑ | Std |
> | :--- | :---: | :---: |
> | Baseline-I2V | 10.250  | 4.770 |
> | VideoGPA-I2V | **11.170**  | 4.750 |
> | Baseline-T2V | 11.460 | 5.420 |
> | VideoGPA-T2V | **11.550** | 5.610 |
>
> As shown above, OFM confirmed no motion degeneration. We also measured scene complexity metrics to show VideoGPA does not simplify scenes to achieve consistency gain (Please see $\color{#7E5CAD}\text{9zXy R2}$). We appreciate this great suggestion and will include these measurements in the final version.
>
> ### **${\color{#E16A54}\text{R4 to W4: 3DCS circular concern}}$**
>
> We acknowledge that 3DCS is computed through the same VGGT pipeline used during training. However, our strongest evaluation signals are *fully independent*: VideoReward (OVL **76.0**% I2V, **60.3**% T2V) and the blind human study (**53.5**% vs 22.4%). To further address this, we evaluated all methods using Depth Anything V3 as an independent geometric backbone, confirming improvements are not VGGT-specific. Please see $\color{#7E5CAD}\text{9zXy R1}$ for the full DA3 table and discussion.
>
> ### **${\color{#E16A54}\text{R5 to Q4: Test on stronger models}}$**
> We applied VideoGPA to Wan2.2-TI2V-5B using the same training setup. Results show consistent improvements.
>
> | Method | PSNR ↑ | SSIM ↑ | LPIPS ↓ | MVCS ↑ | 3DCS ↓ | Epipolar ↓ | OVL ↑ |
>   | :--- | :---: | :---: | :---: | :---: | :---: | :---: | :---: |
>   | Wan2.2-TI2V-5B | 19.406 | 0.723 | 0.480 | 0.898 | 0.506 | 0.594 | - |
>   | VideoGPA (Ours) | **24.411** | **0.842** | **0.394** | **0.944** | **0.451** | **0.499** | **57.0%** |
>
> ### **${\color{#E16A54}\text{R6 to Limitation}}$**
> We acknowledge that our training pipeline relies on static-scene prompts. This is an *intentional design choice* as VGGT excels at static 3D reconstruction. Despite this, we observe that geometric priors learned from static scenes generalize to dynamic scenes with moving animals, vehicles, and articulated motion (Sec. 5.2, Appendix E), suggesting the alignment reshapes the model's latent geometry broadly rather than memorizing scene-specific patterns. As geometric foundation models capable of handling dynamic scenes continue to emerge, our pipeline can naturally extend to dynamic settings by substituting the geometry backbone. We will add this discussion to the final version.

---

> > ### Author Rebuttal · Reviewer_n36s · 2026-04-04
> >
> > Thanks for the detailed rebuttal, which directly addressed my main concerns.
> >
> > Running the SFT baseline on the exact same 2.5k winning pairs clears up the confounding variable issue: it’s good to see proof that the DPO objective itself is driving the gains, rather than just the data filtering. The Optical Flow Magnitude numbers also effectively resolve my concern about motion suppression.
> >
> > I also appreciate the extra effort to test the pipeline on Wan2.2 and use DA3 for the evaluation. Showing that the method scales to a stronger, more recent base model and isn't just overfitting to the VGGT metric makes the overall evaluation stronger.
> >
> > I am keeping my score as a weak accept. It's a practical paper. Please make sure all these rebuttal experiments are included in the camera-ready version.

---

> > > ### Author Response · Authors · 2026-04-05
> > >
> > > We sincerely thank the reviewer for the thorough evaluation and for confirming that our rebuttal experiments address the concerns. We will include all new results in the camera-ready version as promised. We would greatly appreciate a score increase if we have successfully resolved your concerns. Otherwise, we would be happy to address any further questions.

---

### Decision · Program_Chairs · 2026-04-30

**Decision:**

Accept (regular)

**Comment:**

The paper received all acceptance ratings after rebuttal and discussion, in which one reviewer upgraded the rating. Initially, reviewers had concerns about some technical clarity, baseline comparisons (e.g., dataset), and evaluation protocol (e.g., via VGGT). After the rebuttal, all the reviewers are satisfied with the feedback. After taking a closer look at the paper, reviews, and rebuttal, AC agrees with the reviewers' assessment and hence recommends the acceptance rating, while highly encouraging the authors to improve the current version accordingly and release the code for reproducibility.